# A Theoretical Analysis of Fine-tuning with Linear Teachers

**Gal Shachaf**
Blavatnik School of Computer Science,
Tel Aviv University, Israel

**Alon Brutzkus**
Blavatnik School of Computer Science,
Tel Aviv University, Israel

**Amir Globerson**
Blavatnik School of Computer Science,
Tel Aviv University, Israel
and Google Research

## Abstract

Fine-tuning is a common practice in deep learning, achieving excellent generalization results on downstream tasks using relatively little training data. Although widely used in practice, it is lacking strong theoretical understanding. Here we analyze the sample complexity of this scheme for regression with linear teachers in several architectures. Intuitively, the success of fine-tuning depends on the similarity between the source tasks and the target task, however measuring this similarity is non trivial. We show that generalization is related to a measure that considers the relation between the source task, target task and covariance structure of the target data. In the setting of linear regression, we show that under realistic settings a substantial sample complexity reduction is plausible when the above measure is low. For deep linear regression, we present a novel result regarding the inductive bias of gradient-based training when the network is initialized with pretrained weights. Using this result we show that the similarity measure for this setting is also affected by the depth of the network. We further present results on shallow ReLU models, and analyze the dependence of sample complexity on source and target tasks in this setting.

## 1 Introduction

In recent years fine-tuning has emerged as an effective approach to learning tasks with relatively little labeled data. In this setting, a model is first trained on a source task where much data is available (e.g., masked language modeling for BERT), and then it is further tuned using gradient descent methods on labeled data of a target task [1, 2, 3, 4]. Furthermore, it has been observed that fine-tuning can outperform the strategy of fixing the representation learned on the source task, mainly in natural language processing [1, 5]. Despite its empirical success, fine-tuning is poorly understood from a theoretical perspective. One apparent conundrum is that fine-tuned models can be much larger than the number of target training points, resulting in a heavily overparameterized model that is prone to overfitting and poor generalization. Thus, the answer must lie in the fact that fine-tuning is performed with gradient descent and not an arbitrary algorithm that could potentially "ignore" the source task [6]. Here we set out to formalize this problem and understand the factors that determine whether fine-tuning will succeed. We note that this question can be viewed as part of the general quest to understand the implicit bias of gradient based methods [6, 7, 8, 9, 10, 11, 12, 13], but in the particular context of fine-tuning.

35th Conference on Neural Information Processing Systems (NeurIPS 2021).

We begin by highlighting the obvious link between fine-tuning and initialization. Namely, the only difference between "standard" training of a target task and fine-tuning on it, is the initial value of the model weights before beginning the gradient updates. Our goal is to understand the interplay between the model parameters at initialization (namely the source task), the target distribution, and the accuracy of the fine-tuned model. A natural hypothesis is that the distance between the pretrained and fine-tuned model weights is what governs the success of fine-tuning. Indeed, some argue that this is both the key to bound the generalization error of a model and the implicit regularization of gradient-based methods [14, 15, 16, 17]. However, this approach has been discouraged both by empirical testing of the generalization bounds inspired by it [18] and by theoretical works showing this cannot be the inductive bias in deep neural networks [19]. Our results further establish the hypothesis that the success of fine-tuning is affected by other factors.

In this paper we focus on the case in which both source and target regression tasks are linear functions of the input. We start by considering one layer linear networks, and derive novel sample complexity results for fine-tuning. We then proceed to the more complex case of deep linear networks, and prove a novel result characterizing the fine-tuned model as a function of both the weights after pretraining and the depth of the network, and use it to derive corresponding generalization results.

Our results provide several surprising insights. First, we show that the covariance structure of the target data has a significant effect on the success of fine-tuning. In particular, sample complexity is affected by the degree of alignment between the source-target weight difference and the eigenvectors of the target covariance. Second, we find a strong connection between the depth of the network and the results of the fine-tuning process, since deeper networks will serve to cancel the effect of scale differences between source and target tasks. Our results are corroborated by empirical evaluations.

We conclude with results on ReLU networks, providing the first sample complexity result for fine-tuning. For the case of linear teachers, this asserts a simple connection between the source and target models and the test error of fine-tuning.

Taken together, our results demonstrate that fine-tuning is affected not only by some notion of distance between the source and target tasks, but also by the target covariance and the architecture of the model. These results can potentially lead to improved accuracy in this setting via appropriate design of the tasks used for pretraining and the choice of the model architecture.

## 2   Related work

Empirical work [20] has shown that two instances of models initialized from pre-trained weights are more similar in features space than those initialized randomly. Other works [21, 22, 23] have shown that fine-tuned models generalize well when the representation used by the target task is similar to the one used by the source tasks.

In linear regression, [24] showed that gradient descent finds the solution with minimal distance to the initial weights. More recently, attention has turned towards the phenomenon of "benign overfitting" [25, 26] in high dimensional linear regression, where despite fitting noise in training data, population risk may be low. Theoretical analysis of this setting [25] studied how it is affected by the data covariance structure. Benign overfitting was also recently analyzed in the context of ridge-regression [27] and online stochastic gradient descent [28]. Our work continues this line of work on high dimensional regression, but differs from the above papers as we start from a source task, then train on a fixed training set from a target task and consider the global optimum of the this training loss (unlike online SGD). Furthermore, we go beyond the linear regression framework, and obtain surprising characteristics of fine-tuning in deep linear networks.

For linear regression with deep linear models, [29] have recently shown an implicit bias for a two-layer network with deterministic initialization, and [30] have shown an implicit bias for a network with arbitrary depth and near-zero random initialization. Our work generalizes the inductive bias found by [29] to a network of arbitrary depth, and analyses the generalization error of such networks for infinite depth. For linear regression with shallow linear networks [31] have shown a generalization bound that depends only on the norm of the target task, which we use in Section 6.

# 3 Preliminaries and settings

**Notations** Let $\|\cdot\|$ be the $L^2$ norm for vectors and the spectral norm for matrices. For a vector $\boldsymbol{v}$ we denote $\hat{\boldsymbol{v}} \triangleq \frac{\boldsymbol{v}}{\|\boldsymbol{v}\|}$. For a matrix $\mathbf{M} \in \mathbb{R}^{d \times d}$ and some $0 \leq m \leq d$, we define $\mathbf{M}_{\leq m} \in \mathbb{R}^{d \times m}$ to be the matrix containing the first $m$ columns of $\mathbf{M}$. Similarly, we let $\mathbf{M}_{>m}$ denote the matrix containing the columns from $m+1$ to $d$ in $\mathbf{M}$.

Let $\mathcal{D}$ be a distribution over $\mathbb{R}^d$. Let $\boldsymbol{\Sigma}$ be the covariance matrix of $\mathcal{D}$ and let $\mathbf{V} \boldsymbol{\Lambda} \mathbf{V}^\top$ be its eigenvalue decomposition such that $\lambda_1 \geq \ldots \geq \lambda_d$. We define the projection matrices:

$$\mathbf{P}_{\leq k} \triangleq \mathbf{V}_{\leq k} \mathbf{V}_{\leq k}^\top; \quad \mathbf{P}_{>k} \triangleq \mathbf{V}_{>k} \mathbf{V}_{>k}^\top,$$

projecting onto the span of the top $k$ eigenvectors of $\boldsymbol{\Sigma}$, onto the span of the $d-k$ bottom eigenvectors of $\boldsymbol{\Sigma}$, respectively. We will refer to the former as the "top-$k$ span" of $\boldsymbol{\Sigma}$, and to the latter as the "bottom-$k$ span" of $\boldsymbol{\Sigma}$.

Let $\mathbf{X} \in \mathbb{R}^{n \times d}$ be the row matrix of $n < d$ samples drawn from $\mathcal{D}$, and denote the empirical covariance matrix $\frac{1}{n}\mathbf{X}^T\mathbf{X}$ by $\tilde{\boldsymbol{\Sigma}}$. Define $\mathbf{P}_\|$ to be the projection matrix into the row space of $\mathbf{X}$, and $\mathbf{P}_\perp$ to be the projection matrix into its orthogonal complement, i.e.:

$$\mathbf{P}_\| \triangleq \mathbf{X}^\top (\mathbf{X}\mathbf{X}^\top)^{-1}\mathbf{X}, \quad \mathbf{P}_\perp \triangleq \mathbf{I} - \mathbf{P}_\|.$$

Consider a set of parameters $\boldsymbol{\Theta}$, and let $\boldsymbol{\Theta}(t)$ denote the set of parameters at time $t$. We denote the output of a model whose weights are $\boldsymbol{\Theta}(t)$ on a vector $\mathbf{x}$ by $f(\mathbf{x}; \boldsymbol{\Theta}(t)) \in \mathbb{R}$. In the different sections of this work we will overload $f$ with different architectures.

We consider the problem of fine-tuning based transfer learning in regression tasks with linear teachers. Let $\boldsymbol{\theta}_T \in \mathbb{R}^d$ be the ground-truth parameters of the target task, i.e. the linear teacher which we wish to learn, and $\mathbf{y} \in \mathbb{R}^n$ be the target labels of $\mathbf{X}$, s.t. $\mathbf{y} = \mathbf{X}\boldsymbol{\theta}_T$.

We define $L(\boldsymbol{\Theta})$ to be the empirical MSE loss on $\mathbf{X}, \mathbf{y}$ and define $R(\boldsymbol{\Theta})$ as the $\mathcal{D}$ population loss:

$$L(\boldsymbol{\Theta}) \triangleq \frac{1}{n} \|f(\mathbf{X}, \boldsymbol{\Theta}) - \mathbf{y}\|_2^2, \quad R(\boldsymbol{\Theta}) \triangleq \mathbb{E}_{\mathbf{x} \sim \mathcal{D}}\left[\left(\mathbf{x}^\top \boldsymbol{\theta}_T - f(\mathbf{x}, \boldsymbol{\Theta})\right)^2\right].$$

We separate the training procedure into two parts. In the first "pretraining" part, we train a model on $n_S$ pretraining samples $\mathbf{X}_S \in \mathbb{R}^{n_S \times d}$ labeled by a linear teacher $\boldsymbol{\theta}_S$ (i.e., $\mathbf{y}_S = \mathbf{X}_S \boldsymbol{\theta}_S \in \mathbb{R}^{n_S}$), resulting in the set of model weights $\boldsymbol{\Theta}_S$. In the second part, which we call fine-tuning, we initialize a model with the pretrained weights $\boldsymbol{\Theta}(0) = \boldsymbol{\Theta}_S$ and learn the target task by optimizing $L(\boldsymbol{\Theta}(t))$.

Optimization is done by either gradient descent (GD) or gradient flow (GF). Let $\boldsymbol{\theta}(t)$ be some weight vector or weight matrix in $\boldsymbol{\Theta}(t)$. The dynamics for gradient descent optimization with some learning rate $\eta > 0$ are $\boldsymbol{\theta}(t+1) = \boldsymbol{\theta}(t) - \eta \frac{\partial L(\boldsymbol{\Theta}(t))}{\partial \boldsymbol{\theta}(t)}$, and the dynamics for gradient flow are $\dot{\boldsymbol{\theta}}(t) = -\frac{\partial L(\boldsymbol{\Theta}(t))}{\partial \boldsymbol{\theta}(t)}$. Next we state several assumptions about our setup.

**Assumption 3.1.** $\mathbf{X}\mathbf{X}^\mathbf{T}$ *is non-singular. i.e. the rows of* $\mathbf{X}$ *are linearly-independent.*

This assumption holds with high probability for, e.g., a continuous distribution with support over a non-zero measure set. This assumption is only used for simplicity, as the high probability can be incorporated into the analysis.

**Assumption 3.2** (Perfect pretraining). *The pretraining optimization process learns the linear teacher perfectly, e.g. for linear regression we assume that* $f(\mathbf{x}, \boldsymbol{\Theta}_S) = \mathbf{x}^\top \boldsymbol{\theta}_S$, *for* $\mathbf{x} \sim \mathcal{D}$.

Notice that for linear and deep linear models, perfect pretraining can be achieved when $n_S \geq d$. Our results can be easily extended to the case where the equality $f(\mathbf{x}, \boldsymbol{\Theta}_S) = \mathbf{x}^\top \boldsymbol{\theta}_S$ holds approximately and with high probability, but for simplicity we assume equality.

**Assumption 3.3** (Zero train loss). *The fine-tuning converges, i.e.* $\lim_{t \to \infty} L(\boldsymbol{\Theta}(t)) = 0$.

We note that when $f$ is standard linear regression, arbitrarily small train loss can be obtained via gradient descent. For deep linear networks, it can be shown [32] that under suitable initialization a global optimum can be reached, and thus Assumption 3.3 holds for this framework as well.

# 4 Analyzing fine-tuning in linear regression

In this section we analyze fine-tuning for the case of linear teachers for linear regression when using gradient descent for optimization. We define $\boldsymbol{\Theta}(t) = \mathbf{w}(t) \in \mathbb{R}^d$ and overload $f(\mathbf{x}, \boldsymbol{\Theta}(t)) \triangleq \mathbf{x}^\top \mathbf{w}(t)$. In what follows we denote the parameter learned in the fine-tuning process by $\boldsymbol{\gamma} \triangleq \lim_{t \to \infty} \mathbf{w}(t)$.

## 4.1 Results

The following known results (e.g., [24, 25, 10]) show the inductive bias of gradient descent with non-zero initialization in under-determined linear regression and the corresponding population loss.

**Theorem 4.1.** *[24, 25, 10] When $f(\mathbf{x}, \boldsymbol{\Theta})$ is a linear function, fine-tuning with GD under Assumption 3.1, Assumption 3.2 and Assumption 3.3 results in the following model:*

$$\boldsymbol{\gamma} = \mathbf{P}_\perp \boldsymbol{\theta}_S + \mathbf{P}_\| \boldsymbol{\theta}_T, \tag{1}$$

*and*

$$R(\boldsymbol{\gamma}) = \left\| \boldsymbol{\Sigma}^{1/2} \mathbf{P}_\perp (\boldsymbol{\theta}_T - \boldsymbol{\theta}_S) \right\|^2. \tag{2}$$

Theorem 4.1 provides two interesting observations: the first is that $\boldsymbol{\gamma}$ consists of two parts, one which is the projection of the initial weights $\boldsymbol{\theta}_S$ into the null space of $\mathbf{X}$, and the other which is the projection of $\boldsymbol{\theta}_T$ into the span of $\mathbf{X}$. The second observation is that the population risk depends solely on the difference $\boldsymbol{\theta}_T - \boldsymbol{\theta}_S$ that is projected to the null space of the data. For completeness, the proof of Theorem 4.1 is given in the supplementary.

Theorem 4.1 depends on the data matrix $\mathbf{X}$ (via $\mathbf{P}_\|, \mathbf{P}_\perp$). However, to better understand the properties of fine-tuning, a high probability bound on $R$ that does not depend on $\mathbf{X}$ is desirable. We provide such a bound, highlighting the dependence of the population risk on the source and target tasks, and the target covariance $\Sigma$.

**Theorem 4.2.** *Assume the conditions of Theorem 4.1 hold, and assume that the rows of $\mathbf{X}$ are i.i.d. subgaussian centered random vectors. Then, there exists a constant $c > 0$, such that, for all $\delta \geq 1$, and for all $1 \leq m \leq d$ such that $\lambda_m > 0$, with probability at least $1 - e^{-\delta}$ over $\mathbf{X}$, the population risk $R(\boldsymbol{\gamma})$ is bounded by:*

$$2g(\boldsymbol{\lambda}, \delta, n)^3 \frac{\|\mathbf{P}_{\leq m}(\boldsymbol{\theta}_T - \boldsymbol{\theta}_S)\|^2}{\lambda_m^2} + 2g(\boldsymbol{\lambda}, \delta, n)\|\mathbf{P}_{>m}(\boldsymbol{\theta}_T - \boldsymbol{\theta}_S)\|^2, \tag{3}$$

*where $g(\boldsymbol{\lambda}, \delta, n) = c\lambda_1 \max\{\sqrt{\frac{\sum_i \lambda_i}{n\lambda_1}}, \frac{\sum_i \lambda_i}{n\lambda_1}, \sqrt{\frac{\delta}{n}}, \frac{\delta}{n}\}$ and $\left\| \tilde{\boldsymbol{\Sigma}} - \boldsymbol{\Sigma} \right\| \leq g(\boldsymbol{\lambda}, \delta, n)$.*

In the proof, we address the randomness of $\mathbf{P}_\perp (\boldsymbol{\theta}_T - \boldsymbol{\theta}_S)$ in (2), by decomposing $\boldsymbol{\theta}_T - \boldsymbol{\theta}_S$ into its top-$k$ span and bottom-$k$ span components, and then applying the Davis-Kahan $\sin(\Theta)$ theorem [33] to bound the norm of the projection of the former to the null space of the data. The full proof is given in the supp.

The bound in Theorem 4.2 has two key components. The first is the function $g(\boldsymbol{\lambda}, \delta, n)$ that captures how well the covariance $\boldsymbol{\Sigma}$ is estimated, and shows the dependence of the bound on the number of train samples used (as it depends on $n^{-0.5}$). The second relates to the two matrix norms of $\boldsymbol{\theta}_T - \boldsymbol{\theta}_S$ with respect to different parts of the covariance $\boldsymbol{\Sigma}$. Notice that the term relating to the *top-k* span decreases like $n^{-1.5}$, while the term relating to *bottom-k* span decreases like $n^{-0.5}$.

This theorem highlights the conditions under which fine-tuning is expected to perform well. For small enough $n$ s.t. $g(\boldsymbol{\lambda}, \delta, n) > 1$, the bound mainly depends on $\|\mathbf{P}_{\leq m}(\boldsymbol{\theta}_T - \boldsymbol{\theta}_S)\|$. In this case, the bound will be low if $\boldsymbol{\theta}_T$ and $\boldsymbol{\theta}_S$ are close in the span of the *top* eigenvectors of the target distribution. On the other hand, for large enough $n$ s.t. $g(\boldsymbol{\lambda}, \delta, n) < 1$, the bound mainly depends on $\|\mathbf{P}_{>m}(\boldsymbol{\theta}_T - \boldsymbol{\theta}_S)\|$. Thus, the bound will be low if $\boldsymbol{\theta}_T$ and $\boldsymbol{\theta}_S$ are close in the span of the *bottom* eigenvectors of the target distribution.

We conclude with a remark regarding the integer $m$ appearing in the bound, in the case where $g(\boldsymbol{\lambda}, \delta, n) < 1$. While finding the exact $m$ that minimizes the bound is not straightforward, the trade-off in selecting it suggests taking the largest $m$ which holds $\lambda_{m+1} \approx \lambda_m$. This will "cover" more of $\mathbf{P}_{>m}(\boldsymbol{\theta}_T - \boldsymbol{\theta}_S)$ without greatly increasing the left part of (3).

Table 1: Correlation coefficient $R^2$ between the accuracy on different transfer tasks in MNIST and various population risk upper bounds. Each value is a mean over 10 calculations of $R^2$ with different initialization, and each $R^2$ is calculated from 20 points, each one representing a mean accuracy value of 25 random samples.

| Number of Samples | 10 | 15 | 20 | 25 | 30 |
|---|---|---|---|---|---|
| $\|\boldsymbol{\theta}_T - \boldsymbol{\theta}_S\|^2$ | $0.69 \pm 0.03$ | $0.68 \pm 0.04$ | $0.66 \pm 0.04$ | $0.64 \pm 0.03$ | $0.62 \pm 0.02$ |
| Bound from [25] | $0.73 \pm 0.03$ | $0.75 \pm 0.03$ | $0.74 \pm 0.03$ | $0.71 \pm 0.02$ | $0.67 \pm 0.02$ |
| Ours for $m = 2$ | $\mathbf{0.86 \pm 0.02}$ | $\mathbf{0.89 \pm 0.02}$ | $\mathbf{0.84 \pm 0.02}$ | $\mathbf{0.75 \pm 0.01}$ | $\mathbf{0.69 \pm 0.02}$ |

## 4.2 Experiments

In Figure 1 we empirically verify the conclusions from the bound in (3). We set $d = 1000$ and design the target covariance $\boldsymbol{\Sigma}$ s.t. the first $m = 50$ eigenvalues are significantly larger than the rest (1.5 vs. 0.3). We then consider two settings for $\boldsymbol{\theta}_T - \boldsymbol{\theta}_S$. In the first, which we call "Top Eigen Align", we select $\boldsymbol{\theta}_T$ and $\boldsymbol{\theta}_S$ such that $\mathbf{P}_{\leq m}(\boldsymbol{\theta}_T - \boldsymbol{\theta}_S) = 0$. In the second which we call "Bottom Eigen Align" we set $\mathbf{P}_{>m}(\boldsymbol{\theta}_T - \boldsymbol{\theta}_S) = 0$. In both settings we use the same norm $\|\boldsymbol{\theta}_T - \boldsymbol{\theta}_S\|_2$, to show that the bound is not affected by this norm.

As discussed above, our bound suggests better generalization performance of "Bottom Eigen Align" for large $n$ and better performance of "Top Eigen Align" for small $n$. Indeed, we see that while for very few samples "Top Eigen Align" has a lower population loss than "Bottom Eigen Align", the population loss of "Bottom Eigen Align" drops significantly as $n$ grows, and drops to zero well before $n = d$.

We next evaluate the bound on fine-tuning tasks taken from the MNIST dataset [34], and compare it to alternative bounds. Specifically, since we do not expect bounds to be numerically accurate, we calculate the correlation between the actual risk in the experiment and the risk predicted by the bounds. The task we consider (both source and target) is binary classification, which we model as regression to outputs $\{-1, +1\}$. We generate $K$ source-target task pairs (e.g., source task is label 2 vs label 3 and target tasks is label 5 vs label 6). For each such pair we perform source training followed by fine-tuning to target. We then record both the 0-1 error on an independent test set and the value predicted by the bounds. This way we obtain $K$ pairs of points (i.e., actual error vs bound), and calculate the $R^2$ for these pairs, indicating the level to which the bound agrees with the actual error. In addition to our bound in (3), we consider the following: the norm of source-target difference $\|\boldsymbol{\theta}_T - \boldsymbol{\theta}_S\|^2$ and a bound adapted from [25] to the case of fine-tuning.[1] The results in Table 1 show that there is a strong correlation between our bound and the actual error, and the correlation is weaker for the other bounds.

## 5 Analyzing fine-tuning in deep linear networks

In this section we focus on the setting of overparameterized deep linear networks. Although the resulting function is linear in its inputs, like in the previous section, we shall see that the effect of fine-tuning is markedly different. Previous works (e.g. [35, 36]) have shown that linear networks exhibit many interesting properties which make them a good study case towards more complex non-linear networks.

We consider networks with $L$ layers, given by the following matrices: $\boldsymbol{\Theta}(t) = \{\mathbf{W}_1(t), \cdots, \mathbf{W}_L(t)\}$ s.t. $\mathbf{W}_j(t) \in \mathbb{R}^{d_{j-1} \times d_j}$, $d_0 = d$, $d_L = 1$ and for $1 \leq j \leq L - 1 : d_j \geq d$. We also define:

$$\boldsymbol{\beta}(t) = \mathbf{W}_1(t) \cdot \mathbf{W}_2(t) \cdots \mathbf{W}_L(t),$$

such that $f(\mathbf{x}; \boldsymbol{\Theta}(t))(t) = \mathbf{x}^\top \boldsymbol{\beta}(t)$. From Assumption 3.2, we have that $\boldsymbol{\beta}(0) = \boldsymbol{\theta}_S$.

We recall the condition of perfect balancedness (or 0-balancedness) [32]:

**Definition 5.1.** *The weights of a depth $L$ deep linear network at time $t$ are called 0-balanced if:*

$$\mathbf{W}_j(t)^\top \mathbf{W}_j(t) = \mathbf{W}_{j+1}(t)\mathbf{W}_{j+1}(t)^\top \quad for \quad j \in [L-1]. \tag{4}$$

---

[1]The adaptation is straightforward: since the population loss for non-random initialization depends on $\boldsymbol{\theta}_T - \boldsymbol{\theta}_S$ instead of $\boldsymbol{\theta}_T$, we can replace the ground-truth expression $\boldsymbol{\theta}^\star$ in Theorem 4 from [25] with $\boldsymbol{\theta}_T - \boldsymbol{\theta}_S$.

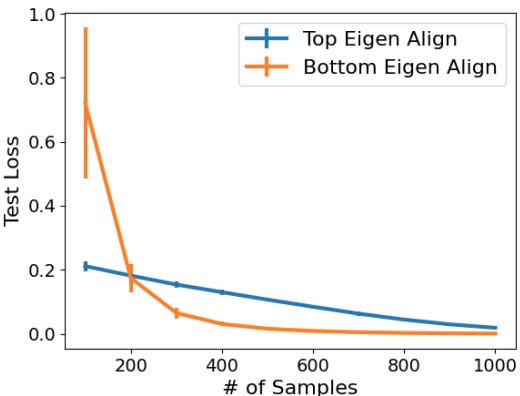

Figure 1: Comparison between different $\boldsymbol{\theta}_T - \boldsymbol{\theta}_S$. "Top Eigen Align" is the linear predictor initialized with $\mathbf{P}_{\leq m}(\boldsymbol{\theta}_T - \boldsymbol{\theta}_S) = 0$ and "Bottom Eigen Align" is the linear predictor initialized with $\mathbf{P}_{>m}(\boldsymbol{\theta}_T - \boldsymbol{\theta}_S) = 0$, for $m$=50. The top $m$ eigenvalues have the value 1.5, compared to the rest which have the value 0.3.

Our analysis requires the initial random initialization (prior to pretraining) to be 0-balanced, which can be achieved with a near zero random initialization, as discussed in [32]. We provide three results on the effect of fine-tuning in this setting. The first result shows the inductive bias of fine-tuning a depth $L$ deep linear network (Theorem 5.2), which holds for arbitrary $L$ and generalizes known results for $L = 1$ (Theorem 4.1) and $L = 2$ [29]. The second result analyzes the population risk of such a predictor when $L \to \infty$ for certain settings (Theorem 5.3 and Theorem 5.4). The third result shows why fixing the first layer (or any set of layers containing the first layer) after pretraining can harm fine-tuning (Theorem 5.5).

The next theorem characterizes the model learned by fine-tuning in the above setting (it can thus be viewed as the deep-linear version of the $L = 1$ result in Theorem 4.1):

**Theorem 5.2.** *Assume that before pretraining, the weights of the model were 0-balanced and that Assumption 3.1, Assumption 3.2 and Assumption 3.3 hold. Then:*

$$\lim_{t \to \infty} \boldsymbol{\beta}(t) = \left( \frac{\| \lim_{t \to \infty} \boldsymbol{\beta}(t) \|}{\| \boldsymbol{\theta}_S \|} \right)^{\frac{L-1}{L}} \mathbf{P}_\perp \boldsymbol{\theta}_S + \mathbf{P}_\| \boldsymbol{\theta}_T \tag{5}$$

*and:*

$$\lim_{L \to \infty} \lim_{t \to \infty} \boldsymbol{\beta}(t) = \frac{\| \mathbf{P}_\| \boldsymbol{\theta}_T \|}{\| \mathbf{P}_\| \boldsymbol{\theta}_S \|} \mathbf{P}_\perp \boldsymbol{\theta}_S + \mathbf{P}_\| \boldsymbol{\theta}_T. \tag{6}$$

To prove this, we focus on $\mathbf{W}_1$, and notice that the gradients $\dot{\mathbf{W}}_1(t)$ are in the span of $\mathbf{X}$, and hence $\mathbf{P}_\perp \mathbf{W}_1(0)$ and its norm remain static during the GF optimization ([30]). We then analyze the norm of the fine-tuned model by using the 0-balancedness property of the weights and the min-norm solution to the equivalent linear regression problem, and achieve (5). (6) is achieved by calculating the limit w.r.t. $L$. The proof of Theorem 5.2 is given in the supplementary.

Although the expression in (5) is not a closed form expression for $\lim_{t \to \infty} \boldsymbol{\beta}(t)$ (because $\| \lim_{t \to \infty} \boldsymbol{\beta}(t) \|$ appears on the RHS), taking $L$ to infinity (6) does result in a closed form expression and demonstrates the effect of increasing model depth. As in (1), we see that the end-to-end equivalent has two components: one which is parallel to the data and one which is orthogonal to it. However, while in (1) the orthogonal component has the original norm of the orthogonal projection of $\boldsymbol{\theta}_S$, the expression in (6) offers a re-scaling of the norm of this component by some ratio that also depends on $\boldsymbol{\theta}_T$. Presenting this phenomenon for the infinity depth limit might look impractical, but the empirical results given in this section show that the effect of depth is apparent even for models of relatively small depth.

## 5.1 When Does Depth Help Fine-Tuning?

In this subsection we wish to understand the effect of depth on the population risk of the fine-tuned model. For simplicity we focus on the limit in (6), and denote $\boldsymbol{\beta} = \lim_{L\to\infty} \lim_{t\to\infty} \boldsymbol{\beta}(t)$.

Since the linear network is a linear function of $\boldsymbol{x}$, we can derive an expression for the population risk of the network, similar to (2):

$$R(\boldsymbol{\beta}) = \left\| \boldsymbol{\Sigma}^{\frac{1}{2}} \mathbf{P}_{\perp} \left( \boldsymbol{\theta}_T - \frac{\|\mathbf{P}_{\|}\boldsymbol{\theta}_T\|}{\|\mathbf{P}_{\|}\boldsymbol{\theta}_S\|} \boldsymbol{\theta}_S \right) \right\|^2. \tag{7}$$

However, since $\mathbf{P}_{\|}$ depends on the random matrix $\mathbf{X}$, without further assumptions this expression by itself is not enough to understand the behaviour of $R(\boldsymbol{\beta})$. Theorem 5.3 and Theorem 5.4 analyze cases for which a bound on (7) can be achieved, showing that it depends on $\|\boldsymbol{\theta}_T\| (\hat{\boldsymbol{\theta}}_T - \hat{\boldsymbol{\theta}}_S)$, i.e. the product of the norm of $\boldsymbol{\theta}_T$ and the difference of the *normalized* $\boldsymbol{\theta}_T$ and $\boldsymbol{\theta}_S$, compared to (2) which depends on the difference between the un-normalized vectors. This observation further highlights the fact that the distance between source and target vectors is not a good predictor of fine-tuning accuracy for some architectures, as fine-tuning can still succeed even if the source and target are very far as long as they are aligned.

We formalize this in the following result, where $\boldsymbol{\theta}_T$ is identical to $\boldsymbol{\theta}_S$ in direction, but not in norm.

**Theorem 5.3.** *Assume that the conditions of Theorem 5.2 hold, and that $\hat{\boldsymbol{\theta}}_T = \hat{\boldsymbol{\theta}}_S$. Namely:*

$$\boldsymbol{\theta}_T = \alpha \boldsymbol{\theta}_S, \quad for \quad \alpha > 0,$$

*then for $L \to \infty$ the risk of the end-to-end solution $\boldsymbol{\beta}$ is*

$$R(\boldsymbol{\beta}) = 0,$$

*while for the $L = 1$ solution $\boldsymbol{\gamma}$, the risk is:*

$$R(\boldsymbol{\gamma}) = \left( \frac{\alpha - 1}{\alpha} \right)^2 \|\boldsymbol{\Sigma}^{1/2}\mathbf{P}_{\perp}\boldsymbol{\theta}_T\|^2 \neq 0 \quad for \quad \alpha \neq 1, \alpha > 0. \tag{8}$$

This setting highlights our conclusion on the role of alignment in deep linear models: if the tasks are aligned, the deep linear predictor achieves zero generalization even with a single sample, while the population risk of the $L = 1$ predictor still depends on $n$.

Another example for this behaviour can be seen when $\mathbf{X}$ is i.i.d Gaussian (i.e., $\mathcal{D} = \mathcal{N}(0, 1)^d$).

**Theorem 5.4.** *Assume that the conditions of Theorem 5.2 hold, and let $\mathbf{X} \sim \mathcal{N}(0, 1)^d$. Suppose $n \leq d$, then there exists a constant $c > 0$ such that for any $\epsilon > 0$ with probability at least $1 - 4\exp(-c\epsilon^2 n) - 4\exp(-c\epsilon^2(d-n))$ the population risk for the $L \to \infty$ end-to-end predictor $\boldsymbol{\beta}$ is bounded as follows:*

$$R(\boldsymbol{\beta}) \leq \frac{d - n}{d}(1 + \epsilon)^2 \|\boldsymbol{\theta}_T\|^2 \left\| \hat{\boldsymbol{\theta}}_T - \hat{\boldsymbol{\theta}}_S \right\|^2 + \frac{d - n}{d}\zeta(\|\boldsymbol{\theta}_T\|)^2, \tag{9}$$

*for $\zeta(\|\boldsymbol{\theta}_T\|) \approx \epsilon \|\boldsymbol{\theta}_T\|$. For the $L = 1$ linear regression solution $\boldsymbol{\gamma}$ this risk is bounded by*

$$R(\boldsymbol{\gamma}) \leq \frac{d - n}{d}(1 + \epsilon)^2 \|\boldsymbol{\theta}_T - \boldsymbol{\theta}_S\|^2. \tag{10}$$

The above result is a direct analysis of (7) when $\boldsymbol{\Sigma} = \mathbf{I}$ by using Lemma 5.3.2 from [37] to analyze the effects of $\mathbf{P}_{\|}, \mathbf{P}_{\perp}$. Comparing (9) and (10), we see that while (10) depends on the distance between the two un-normalized tasks, (9) depends on the norm of the target task and the alignment of the tasks, but not at all on the norm of the source task. The proofs of Theorem 5.3 and Theorem 5.4 are given in the supp.

## 5.2 Deep linear fine-tuning with fixing the first layer(s)

A common trick when performing fine-tuning is to fix, or "freeze" (i.e. not train), the first $k$ layers of a model during the optimization on the target task. This method reduces the risk of over-fitting these layers to the small training set.[2] The next theorem shows that for deep linear networks this method degenerates the training process.

---

[2] This over-fitting is sometimes referred to as "catastrophic forgetting" of the source task.

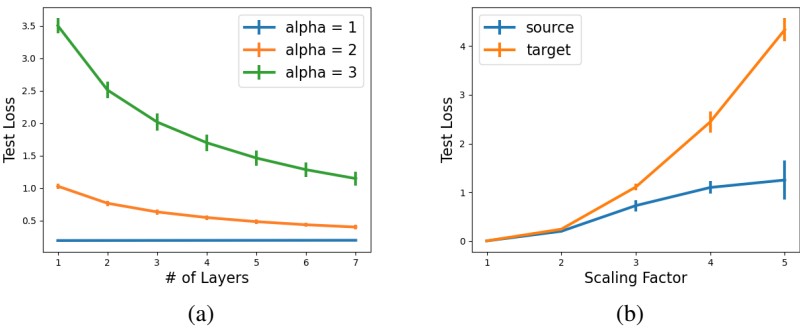

(a)                                         (b)

Figure 2: (a) The effect of depth on fine-tuning when $\boldsymbol{\theta}_T$ is a $\alpha$ scaled, $\epsilon$ noised version of $\boldsymbol{\theta}_S$ with $d/10$ samples. (b) The effect of changing the scale of either source weights or target weights in a 7-layers model.

**Theorem 5.5.** *Assume the setting of Theorem 5.2. Then, if we freeze the first layer (or any number $k$ of first layers) during fine-tuning, the fine-tuned model will be given by $\langle \boldsymbol{\beta}(t), \boldsymbol{x} \rangle = c \langle \mathbf{x}, \boldsymbol{\theta}_S \rangle$, for some constant $c$.*

The key idea in the proof is to show that the product of the $k$ first layers is equal to $\boldsymbol{\theta}_S$ up to a scaling factor, which is a result of [30]. The result implies that after fine-tuning the model is still equal to the source task, independently of the target task. Thus, fine-tuning essentially fails completely, and its error cannot be reduced with additional target data.

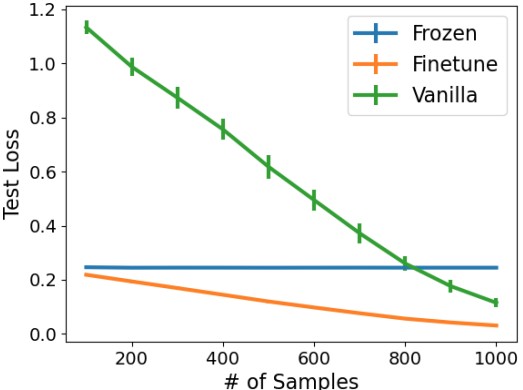

Figure 3: A network whose first layer is fixed has a constant generalization loss due to degeneration effect in Theorem 5.5.

This result is achieved under the assumption of $0$-balancedness prior to pretraining, which happens e.g. when initializing the weights with an infinitesimally small variance, as this property leads to the degeneracy of the output of the frozen k-layers. Though the proof of Theorem 5.5 depends on this $0$-balancedness property of the network, the experiments shown in Figure 3 were conducted with a small initialization scale, that is not guaranteed to result in $0$-balancedness, but rather in $\delta$-approximate balancedness [32] when $\delta$ is small. These experiments show empirically that the phenomenon of learning failure is observed even when $\delta > 0$. Intuitively, this is because the effective rank of the weight matrices is close to one, and thus learning the second layer is an ill-conditioned problem, which leads to slower convergence and can prevent the model from fine-tuning on the target data with a constant gradient step.

A possible workaround to this failure of learning would be to initialize the weights prior to pretraining with a larger scale of initialization (e.g. with Xavier [38]), thus increasing the rank of each layer and preventing degeneracy. Pre-training with multiple source tasks (as suggested in e.g. [22]) may also help the fine-tuning optimization.

## 5.3 Experiments

We next describe experiments that support the results in this section. Theorem 5.3 predicts that deeper nets will successfully learn a case where source and target vectors are aligned, but with different norms. This is demonstrated in Figure 2a where source and target tasks are related via $\boldsymbol{\theta}_T = \alpha \boldsymbol{\theta}_S + \boldsymbol{\epsilon}$, where $\boldsymbol{\epsilon}$ is a standard Gaussian vector whose norm is approximately $0.5 \|\boldsymbol{\theta}_S\|$. It can be seen that when $\alpha \approx 1$, there is no difference between models of different depth. However, as $\alpha$ increases, adding depth has a positive effect on fine-tuning accuracy. Theorem 5.4 predicts that the test loss for a deep linear model would depend only on the alignment of $\boldsymbol{\theta}_S$ and $\boldsymbol{\theta}_T$ (i.e. $\left\| \hat{\boldsymbol{\theta}}_T - \hat{\boldsymbol{\theta}}_S \right\|$) and on the $\|\boldsymbol{\theta}_T\|$, but not on $\|\boldsymbol{\theta}_S\|$. This is demonstrated in Figure 2b where source and target task are initialized s.t. $\left\| \hat{\boldsymbol{\theta}}_T - \hat{\boldsymbol{\theta}}_S \right\| \approx 0.1$. In each experiment, either $\boldsymbol{\theta}_T = \alpha \hat{\boldsymbol{\theta}}_T$ or $\boldsymbol{\theta}_S = \alpha \hat{\boldsymbol{\theta}}_S$, where $\alpha$ is the "Scaling Factor", and the other has norm of 1. It can be seen that increasing the norm of the target vector harms generalization much more than increasing the norm of the source vector, as the theorem predicts, even for a relatively shallow model.

Theorem 5.5 states that fixing the first layer in deep linear nets can result in failure to fine-tune. We illustrate this empirically in Figure 3, where we compare three two-layer linear models on the same target task: 1) A "Frozen" model that fixes the first layer after pretraining. 2) A "Vanilla" model that trains the network from scratch on the target, ignoring the source pre-training. 3) A "Finetune" model that first trains on source and fine-tunes to target. As predicted by theory, the "frozen" model's performance is poor, and fine-tuning has better sample complexity.

## 6 Analyzing fine-tuning in shallow ReLU networks

Analyzing optimization and generalization in non-linear networks is challenging. However, analysis in the Neural Tangent Kernel (NTK) regime is sometimes simpler [39, 31]. Thus, here we take a first step towards understanding fine-tuning in non-linear networks by analyzing this problem in the NTK regime. Specifically, we consider the setting of a two-layer ReLU network with $m$ neurons in the hidden layer. Hence, we consider $\boldsymbol{\Theta}(t) = \{\mathbf{W}(t), \boldsymbol{a}\}$ and $f(\mathbf{x}; \boldsymbol{\Theta}(t)) = \frac{1}{\sqrt{m}} \sum_{r=1}^{m} a_r \sigma(\mathbf{x}^\top \mathbf{w}_r(t))$ where $\sigma$ is the ReLU function, $\mathbf{w}_1(t), \ldots, \mathbf{w}_m(t) \in \mathbb{R}^d$, the rows of $\mathbf{W}(t)$, are vectors in the first layer, and $\boldsymbol{a} \in \{-1, 1\}^m$ is the vector of weights in the second layer. We initialize $\boldsymbol{a}$ uniformly and fix it during optimization as in [39]. Before pretraining, the first layer parameters are initialized from a standard Gaussian with variance $\kappa^2$. We also assume that $\|\mathbf{x}\| = 1$ for all $\mathbf{x}$ samples from $\mathcal{D}$. We let $f(\mathbf{X}, \boldsymbol{\Theta}) \in \mathbb{R}^n$ be the vector of predictions of $f$ on the data $\mathbf{X}$.

For the next theorem we do not assume linear teachers, and instead assume an arbitrary labeling function $g_S$ such that $\mathbf{y}_S = g_S(\mathbf{X}_S)$, for $\mathbf{X}_S \in \mathbb{R}^{n_S \times d}, \mathbf{y}_S \in \mathbb{R}^{n_S}$ the pretraining data and labels, respectively. We also assume that $\mathbf{y} = g_T(\mathbf{X})$ for some arbitrary function $g_T$. For simplicity, we assume $|y|_i \leq 1$ for $i \in [n]$. We consider a setting where the pretraining phase is done using a two-layer network in the NTK regime, under the assumptions of Theorem 4.1 from [31] with respect to the variables $m, \kappa, \eta$ and sufficiently many iterations.[3] Next, in the fine-tuning phase, we train a network initialized with the weights given by the pretraining phase. We use the same value of $m$ for the fine-tuning phase. We rely on the analysis given in [39, 31] and achieve an upper bound on the population risk of the fine-tuned model:

**Theorem 6.1.** *Fix a failure probability $\delta \in (0, 1)$. We assume that Assumption 3.1 holds. Suppose $\kappa = O\left(\frac{\lambda_0 \delta}{n}\right)$, $m \geq \kappa^{-2} \operatorname{poly}\left(n, n_S, \lambda_0^{-1}, \delta^{-1}\right)$. Consider any loss function $\ell : \mathbb{R} \times \mathbb{R} \to [0, 1]$ that is 1-Lipschitz in the first argument such that $\ell(y, y) = 0$. Then with probability at least $1 - \delta$,[4] the two-layer neural network $f(\cdot, \boldsymbol{\Theta}(t))$ fine-tuned by GD for $t \geq \Omega\left(\frac{1}{\eta \lambda_0} \log \|\tilde{\mathbf{y}}\|_2^{-1}\right)$ iterations has population loss:*

$$R(\boldsymbol{\Theta}(t)) \leq 2\sqrt{\frac{\tilde{\mathbf{y}}^\top (\mathbf{H}^\infty)^{-1} \tilde{\mathbf{y}}}{n}} + O\left(\sqrt{\frac{\log \frac{n}{\lambda_0 \delta}}{n}}\right), \tag{11}$$

*for $\tilde{\mathbf{y}} \equiv \mathbf{y} - f(\mathbf{X}, \boldsymbol{\Theta}(0))$.*

---

[3] See the supp for a bound on the number of iterations.

[4] Over the random initialization of the pretraining network.

The above result shows that the true risk of the fine-tuned model is related to the distance of learned outputs $\mathbf{y}$ from the outputs after pretraining $f(\mathbf{X}, \mathbf{\Theta}(0))$. The proof of Theorem 6.1 is given in the supp.

As in previous NTK regime analyses, this result holds when the weights of the fine-tuned model do not "move" too far away from the weights at random initialization. Thus, the proof approach is to bound the distance between the Gram matrix $\mathbf{H}(t)$ and the infinite-width gram matrix $\mathbf{H}^\infty$ with a decreasing function in $m$. The main challenge is that the weights $\mathbf{W}(0)$ are not initialized i.i.d as described above. To address this we provide a careful analysis of the dynamics and show that $\mathbf{H}(t)$ is close to $\mathbf{H}$ at random initialization, even when considering the pretraining phase, which in turn is close to $\mathbf{H}^\infty$.

We next apply our results to the case of linear source and target tasks. We thus assume that $g_S, g_T$ are linear functions with parameters $\boldsymbol{\theta}_S, \boldsymbol{\theta}_T$. For simplicity of exposition we assume $f(\mathbf{x}, \mathbf{\Theta}(0)) = \mathbf{x}^\top \boldsymbol{\theta}_S$ exactly (Assumption 3.2). Before bounding the risk of fine-tuning we bound the RHS of (11) in the linear case:

**Corollary 6.2.** *Suppose that* $g_S(\mathbf{X}) \triangleq \mathbf{X}^\top \boldsymbol{\theta}_S$, $g_T(\mathbf{X}) \triangleq \mathbf{X}^\top \boldsymbol{\theta}_T$, *and assume Assumption 3.2 holds. Then,* $\sqrt{\tilde{\boldsymbol{y}}^\top (\mathbf{H}^\infty)^{-1} \tilde{\boldsymbol{y}}} \leq 3 \|\boldsymbol{\theta}_T - \boldsymbol{\theta}_S\|_2$.

This is a direct corollary of Theorem 6.1 from [31] on $\tilde{\mathbf{y}}$ defined above. Theorem 6.1 and Corollary 6.2 result in the a bound on the risk of the fine-tuned model:

**Corollary 6.3.** *Under the conditions of Theorem 6.1 and Corollary 6.2, it holds that*

$$R(\mathbf{\Theta}(t)) \leq \frac{6 \|\boldsymbol{\theta}_T - \boldsymbol{\theta}_S\|_2}{\sqrt{n}} + O\left( \sqrt{\frac{\log \frac{n}{\lambda_0 \delta}}{n}} \right).$$

We note that fine-tuning is improved as the distance between source and target decreases. In our analysis of linear networks (Theorem 4.2 and Theorem 5.4) we obtained a more fine-grained result depending on the covariance structure. We conjecture that the non-linear case will have similar results, which will likely involve the covariance structure in the NTK feature space.

## 7 Discussion

This paper gives a fine-grained analysis of the process of fine-tuning with linear teachers in several different architectures. It offers insights into the inductive bias of gradient-descent and the implied relation between the source task, the target task and the target covariance that is needed for this process to succeed. We believe our conclusions pave a way towards understanding why some pretrained models work better than others and what biases are transferred from those models during fine-tuning.

A limitation of our work is the simplicity of the models analyzed, and it would certainly be interesting to extend these. Our setting deals only with linear teachers, and assumes the label noise to be zero. Furthermore, we only show upper bounds on the population risk, and not matching lower bounds. For deep linear networks we assume a certain initialization which is less standard than normalized initializers such as Xavier. For non-linear models, we analyze the simple model of a shallow ReLU network, and only in the NTK regime.

An interesting direction to explore is formulating a bound similar to Theorem 4.2 for regression in the RKHS space given by the NTK, where the covariance is now over the RKHS space and thus more challenging to analyze. Another interesting setting is classification with exponential losses. Since the classifier learned by GD in this case has diverging norm, it is not clear how fine-tuning is beneficial, although in practice it often is. We leave these questions for future work.

## Acknowledgments and Disclosure of Funding

This work has been supported by the Israeli Science Foundation research grant 1186/18 and the Yandex Initiative for Machine Learning. AB is supported by the Google Doctoral Fellowship in Machine Learning.

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
