# Supplementary Material: A Theoretical Analysis of Fine-tuning with Linear Teachers

**Gal Shachaf**
Blavatnik School of Computer Science,
Tel Aviv University, Israel

**Alon Brutzkus**
Blavatnik School of Computer Science,
Tel Aviv University, Israel

**Amir Globerson**
Blavatnik School of Computer Science,
Tel Aviv University, Israel
and Google Research

## 1 Proofs for linear regression

This appendix includes proofs for Section 4. It starts by analyzing the solution achieved by applying gradient descent on a linear regression problem with non-zero initialization, and shows its exact population risk. Then, this risk is bounded from above by using concentration bounds to bound various aspects of the difference between the true target covariance and the estimated target covariance.

Recall the assumptions:

**Assumption 3.1** (Main Text). $\mathbf{XX^T}$ *is non-singular. i.e. the rows of* $\mathbf{X}$ *are linearly-independent.*

**Assumption 3.2** (Main Text). *The pretraining optimization process learns the linear teacher perfectly, e.g. for linear regression we assume that* $f(\mathbf{x}, \mathbf{\Theta}_S) = \mathbf{x}^\top \boldsymbol{\theta}_S$, *for* $\mathbf{x} \sim \mathcal{D}$.

**Assumption 3.3** (Main Text). *The fine-tuning converges, i.e.* $\lim_{t\to\infty} L(\mathbf{\Theta}(t)) = 0$.

### 1.1 Proof of Theorem 4.1

As mentioned in the main text, both parts of the theorem have been proven before [1, 2, 3]. The proof is provided for completeness, and can be skipped.

**Lemma 1.1.** *Assume Assumption 3.3, and that there exists some vector* $\mathbf{w} \in \mathbb{R}^d$ *s.t.* $\mathbf{y} = \mathbf{Xw}$ *(i.e. the data is generated via a linear teacher), then the solution achieved by using GD with initialization* $\boldsymbol{\theta}_0$ *in order to minimize:*

$$\min_{\boldsymbol{\theta} \in \mathbb{R}^d} \tfrac{1}{2} \|\mathbf{X}\boldsymbol{\theta} - \mathbf{y}\|_2^2. \tag{1}$$

*is*

$$\boldsymbol{\theta}^\star = \mathbf{P}_\perp \boldsymbol{\theta}_0 + \mathbf{P}_\| \mathbf{w}. \tag{2}$$

*Proof.* First, observe that the gradient step for this problem is

$$\boldsymbol{\theta}_{t+1} = \boldsymbol{\theta}_t + \eta \mathbf{X}^T (\mathbf{y} - \mathbf{X}\boldsymbol{\theta}_T).$$

Hence, all of the steps are in the span of $\mathbf{X}^T$, and GD converges to a solution of the form:

$$\boldsymbol{\theta}^\star = \boldsymbol{\theta}_0 + \mathbf{X}^T \boldsymbol{a}$$

35th Conference on Neural Information Processing Systems (NeurIPS 2021).

for some $\boldsymbol{a} \in R^n$. The vector $\boldsymbol{\theta}^\star$ must also achieve a loss of zero in Equation (1) (because we know that $\mathbf{w}$ achieves a loss of zero, and GD minimizes this objective). Therefore:

$$\mathbf{X}\boldsymbol{\theta}^\star = \mathbf{y}$$
$$\mathbf{X}(\boldsymbol{\theta}_0 + \mathbf{X}^T\boldsymbol{a}) = \mathbf{y}$$
$$\mathbf{X}\mathbf{X}^T\boldsymbol{a} = \mathbf{y} - \mathbf{X}\boldsymbol{\theta}_0$$
$$\boldsymbol{a} \stackrel{1}{=} (\mathbf{X}\mathbf{X}^T)^{-1}(\mathbf{y} - \mathbf{X}\boldsymbol{\theta}_0)$$
$$\Rightarrow \boldsymbol{\theta}^\star = \boldsymbol{\theta}_0 + \mathbf{X}^T(\mathbf{X}\mathbf{X}^T)^{-1}(\mathbf{y} - \mathbf{X}\boldsymbol{\theta}_0),$$

with (1) due to Assumption 3.1.

Replacing $\mathbf{y}$ with $\mathbf{X}\mathbf{w}$, and by using the definitions of $\mathbf{P}_\parallel$ and $\mathbf{P}_\perp$ from Section 3, it follows that

$$\boldsymbol{\theta}_0 + \mathbf{X}^T(\mathbf{X}\mathbf{X}^T)^{-1}(\mathbf{y} - \mathbf{X}\boldsymbol{\theta}_0) = \boldsymbol{\theta}_0 + \mathbf{X}^T(\mathbf{X}\mathbf{X}^T)^{-1}(\mathbf{X}\mathbf{w} - \mathbf{X}\boldsymbol{\theta}_0)$$
$$= \left(\mathbf{I} - \mathbf{X}^T(\mathbf{X}\mathbf{X}^T)^{-1}\mathbf{X}\right)\boldsymbol{\theta}_0 + \mathbf{X}^T(\mathbf{X}\mathbf{X}^T)^{-1}\mathbf{X}\mathbf{w}$$
$$= \mathbf{P}_\perp\boldsymbol{\theta}_0 + \mathbf{P}_\parallel\mathbf{w}.$$

$\square$

We can now prove the theorem.

*Proof of Theorem 4.1 (Main Text).* The proof for Eq.1 in the main text is straightforward by using Lemma 1.1 with $\boldsymbol{\theta}_0 = \boldsymbol{\theta}_S$ and $\mathbf{w} = \boldsymbol{\theta}_T$.

As for Eq.2 in the main text, by Lemma 1.1 it follows that

$$\boldsymbol{\gamma} = \mathbf{P}_\perp\boldsymbol{\theta}_S + \mathbf{P}_\parallel\boldsymbol{\theta}_T.$$

Since $\mathbf{P}_\parallel + \mathbf{P}_\perp = \mathbf{I}$ it follows that

$$R(\boldsymbol{\gamma}) = \mathbb{E}_{\mathbf{x}\sim\mathcal{D}}\left[\left(\mathbf{x}^\top\boldsymbol{\theta}_T - f(\mathbf{x};\boldsymbol{\Theta}(t))\right)^2\right] = \mathbb{E}_{\mathbf{x}\sim\mathcal{D}}\left[\left(\mathbf{x}^\top\left(\boldsymbol{\theta}_T - \mathbf{P}_\perp\boldsymbol{\theta}_S - \mathbf{P}_\parallel\boldsymbol{\theta}_T\right)\right)^2\right]$$
$$= \mathbb{E}_{\mathbf{x}\sim\mathcal{D}}\left[\left(\mathbf{x}^\top\mathbf{P}_\perp\left(\boldsymbol{\theta}_T - \boldsymbol{\theta}_S\right)\right)^2\right] = \mathbb{E}_{\mathbf{x}\sim\mathcal{D}}\left[\left(\boldsymbol{\theta}_T - \boldsymbol{\theta}_S\right)^T\mathbf{P}_\perp\mathbf{x}\mathbf{x}^\top\mathbf{P}_\perp\left(\boldsymbol{\theta}_T - \boldsymbol{\theta}_S\right)\right]$$
$$= \left(\boldsymbol{\theta}_T - \boldsymbol{\theta}_S\right)^T\mathbf{P}_\perp\mathbb{E}_{\mathbf{x}\sim\mathcal{D}}\left[\mathbf{x}\mathbf{x}^\top\right]\mathbf{P}_\perp\left(\boldsymbol{\theta}_T - \boldsymbol{\theta}_S\right) = \left(\boldsymbol{\theta}_T - \boldsymbol{\theta}_S\right)^T\mathbf{P}_\perp^T\boldsymbol{\Sigma}\mathbf{P}_\perp\left(\boldsymbol{\theta}_T - \boldsymbol{\theta}_S\right)$$
$$= \left\|\boldsymbol{\Sigma}^{0.5}\mathbf{P}_\perp\left(\boldsymbol{\theta}_T - \boldsymbol{\theta}_S\right)\right\|^2.$$

thus concluding the proof. $\square$

## 1.2 Proof of Theorem 4.2: Upper bound of the population risk for linear regression

Recall the Davis-Kahan $sin(\Theta)$ theorem:

**Theorem 1.2** ([4])**.** *Let $A = E_0 A_0 E_0^T + E_1 A_1 E_1^T$ and $A + H = F_0 \Lambda_0 F_0^T + F_1 \Lambda_1 F_1^T$ be symmetric matrices with $[E_0, E_1]$ and $[F_0, F_1]$ orthogonal. If the eigenvalues of $A_0$ are contained in an interval $(a, b)$, and the eigenvalues of $\Lambda_1$ are excluded from the interval $(a - \delta, b + \delta)$ for some $\delta > 0$, then*

$$\|F_1^T E_0\| \leq \frac{\|F_1^T H E_0\|}{\delta} \tag{3}$$

*for any unitarily invariant norm $\|\cdot\|$.*

The following theorem is a concentration bound on the difference between the true and estimated covariance matrices: $\left\|\boldsymbol{\Sigma} - \tilde{\boldsymbol{\Sigma}}\right\|$:

**Theorem 1.3** (Theorem 9 from [5])**.** *Let $X, X_1, \ldots, X_n$ be i.i.d. weakly square integrable centered random vectors in $E$ with covariance operator $\boldsymbol{\Sigma}$. If $X$ is subgaussian and pregaussian, then there exists a constant $c > 0$ such that, for all $\delta \geq 1$, with probability at least $1 - e^{-\delta}$,*

$$\|\tilde{\boldsymbol{\Sigma}} - \boldsymbol{\Sigma}\| \leq c\|\boldsymbol{\Sigma}\| \max\left\{\sqrt{\frac{r(\boldsymbol{\Sigma})}{n}}, \frac{r(\boldsymbol{\Sigma})}{n}, \sqrt{\frac{\delta}{n}}, \frac{\delta}{n}\right\} \triangleq g(\boldsymbol{\lambda}, \delta, n),$$

*where*

$$r(\mathbf{\Sigma}) := \frac{(\mathbb{E}\|x\|)^2}{\|\mathbf{\Sigma}\|} \leq \frac{\mathrm{tr}(\mathbf{\Sigma})}{\|\mathbf{\Sigma}\|} = \frac{\sum_i \lambda_i}{\lambda_1}.$$

The following lemma uses Theorem 1.2 to upper bound the dot product between the $d - n$ bottom eigenvectors of the estimated covariance and the top $k$ eigenvectors of the target covariance:

**Lemma 1.4.** *For all $1 \leq k \leq d$ such that $\lambda_k > 0$ it holds that:*

$$\left\|\tilde{\mathbf{V}}_{>n}^T \mathbf{V}_{\leq k}\right\| \leq \frac{\|\tilde{\mathbf{\Sigma}} - \mathbf{\Sigma}\|}{\lambda_k}$$

*Proof.* In order to use Theorem 1.2 with $\delta = \lambda_k$ to bound $\|\tilde{\mathbf{V}}_{>n}^T \mathbf{V}_{\leq k}\|$, one must show that the conditions of Theorem 1.2 are met. Let $\mathbf{A} = \mathbf{\Sigma}$, $\mathbf{A} + \mathbf{H} = \tilde{\mathbf{\Sigma}}$, $\mathbf{E}_0 = \mathbf{V}_{\leq k}$, $\mathbf{A}_0 = \mathbf{\Lambda}_{\leq k}$, $\mathbf{F}_1 = \tilde{\mathbf{V}}_{>n}$, and $\mathbf{\Lambda}_1 = \tilde{\mathbf{\Lambda}}_{>n}$. Notice that $\mathbf{X}$ is a rank-$n$ matrix, and so is the estimated covariance $\tilde{\mathbf{\Sigma}}$, hence it bottom $d - n$ eigenvalues are zero. Thus, all of the $d - n$ eigenvalues of $\mathbf{\Lambda}_1$ equal zero. Also, recall that the eigenvalues of $\mathbf{\Sigma}$ are in descending order. Thus, all of the eigenvalues of $\mathbf{A}_0$ are in the interval $(\lambda_k, \lambda_1)$ and all of the eigenvalues of $\mathbf{\Lambda}_1$ (which equal 0) are excluded from the interval $(0, \lambda_1 + \lambda_k)$. Hence the conditions of Theorem 1.2 are met and for $\delta = \lambda_k$:

$$\|\tilde{\mathbf{V}}_{>n}^T \mathbf{V}_{\leq k}\| \leq \frac{\|\tilde{\mathbf{V}}_{>n}^T (\tilde{\mathbf{\Sigma}} - \mathbf{\Sigma}) \mathbf{V}_{\leq k}\|}{\lambda_k}$$

$$\overset{(1)}{\leq} \frac{\|\tilde{\mathbf{V}}_{>n}\| \|\tilde{\mathbf{\Sigma}} - \mathbf{\Sigma}\| \|\mathbf{V}_{\leq k}\|}{\lambda_k}$$

$$\overset{(2)}{=} \frac{\|\tilde{\mathbf{\Sigma}} - \mathbf{\Sigma}\|}{\lambda_k},$$

with (1) due to Cauchy-Schwartz inequality, (2) due to $\tilde{\mathbf{V}}_{>n}$, $\mathbf{V}_{\leq k}$ being orthonormal matrices, which concludes the proof. $\square$

We can now prove the theorem.

*Proof of Theorem 4.2 (Main Text).* Let $\tilde{\mathbf{U}}\tilde{\mathbf{\Gamma}}\tilde{\mathbf{V}}^T$ be the singular value decomposition of $\mathbf{X}$ such that $\tilde{\mathbf{U}} \in \mathbb{R}^{n \times n}$, $\tilde{\mathbf{V}} \in \mathbb{R}^{d \times d}$ are unitary matrices and let $\tilde{v}_i$ be the $i$-th column of $\tilde{\mathbf{V}}$.

First, notice that $\mathbf{P}_\| = \mathbf{X}^\top (\mathbf{X}\mathbf{X}^\top)^{-1}\mathbf{X}$ can be also written as $\mathbf{I} - \tilde{\mathbf{V}}_{>n}\tilde{\mathbf{V}}_{>n}^T$:

$$\mathbf{X}^\top (\mathbf{X}\mathbf{X}^\top)^{-1}\mathbf{X} = \tilde{\mathbf{V}}\tilde{\mathbf{\Gamma}}^\top\tilde{\mathbf{U}}^T (\tilde{\mathbf{U}}\tilde{\mathbf{\Gamma}}\tilde{\mathbf{V}}^T\tilde{\mathbf{V}}\tilde{\mathbf{\Gamma}}^\top\tilde{\mathbf{U}}^T)^{-1}\tilde{\mathbf{U}}\tilde{\mathbf{\Gamma}}\tilde{\mathbf{V}}^T$$

$$\overset{(1)}{=} \tilde{\mathbf{V}}\tilde{\mathbf{\Gamma}}^\top\tilde{\mathbf{U}}^T (\tilde{\mathbf{U}}(\tilde{\mathbf{\Gamma}}\tilde{\mathbf{\Gamma}}^\top)\tilde{\mathbf{U}}^T)^{-1}\tilde{\mathbf{U}}\tilde{\mathbf{\Gamma}}\tilde{\mathbf{V}}^T$$

$$= \tilde{\mathbf{V}}\tilde{\mathbf{\Gamma}}^\top\tilde{\mathbf{U}}^T (\tilde{\mathbf{U}}(\tilde{\mathbf{\Gamma}}\tilde{\mathbf{\Gamma}}^\top)\tilde{\mathbf{U}}^T)^{-1}\tilde{\mathbf{U}}\tilde{\mathbf{\Gamma}}\tilde{\mathbf{V}}^T$$

$$\overset{(2)}{=} \tilde{\mathbf{V}}\tilde{\mathbf{\Gamma}}^\top\tilde{\mathbf{U}}^T\tilde{\mathbf{U}}(\tilde{\mathbf{\Gamma}}\tilde{\mathbf{\Gamma}}^\top)^{-1}\tilde{\mathbf{U}}^T\tilde{\mathbf{U}}\tilde{\mathbf{\Gamma}}\tilde{\mathbf{V}}^T$$

$$\overset{(3)}{=} \tilde{\mathbf{V}}\tilde{\mathbf{\Gamma}}^\top (\tilde{\mathbf{\Gamma}}\tilde{\mathbf{\Gamma}}^\top)^{-1}\tilde{\mathbf{\Gamma}}\tilde{\mathbf{V}}^T = \tilde{\mathbf{V}} \cdot \mathbf{diag}(\mathbf{1_{1:n}}, \mathbf{0_{n+1:d}}) \cdot \tilde{\mathbf{V}}^T$$

$$= \sum_{i=1}^{n} \tilde{v}_i \cdot \tilde{v}_i^T = \sum_{i=1}^{d} \tilde{v}_i \cdot \tilde{v}_i^T - \sum_{i=n+1}^{d} \tilde{v}_i \cdot \tilde{v}_i^T$$

$$\overset{(4)}{=} \mathbf{I} - \sum_{i=n+1}^{d} \tilde{v}_i \cdot \tilde{v}_i^T = \mathbf{I} - \tilde{\mathbf{V}}_{>n}\tilde{\mathbf{V}}_{>n}^T.$$

Where (1),(3),(4) are due to $\tilde{\mathbf{U}}$, $\tilde{\mathbf{V}}$ being unitary, and (2) is due to $\tilde{\mathbf{U}}(\tilde{\mathbf{\Gamma}}\tilde{\mathbf{\Gamma}}^\top)\tilde{\mathbf{U}}^T (\tilde{\mathbf{U}}(\tilde{\mathbf{\Gamma}}\tilde{\mathbf{\Gamma}}^\top)^{-1}\tilde{\mathbf{U}}^T) = \mathbf{I}$.

From Eq.2 in the main text it follows that:

$$R(\boldsymbol{\gamma}) = \left\|\mathbf{\Sigma}^{0.5}\mathbf{P}_\perp (\boldsymbol{\theta}_T - \boldsymbol{\theta}_S)\right\|^2$$

$$= (\boldsymbol{\theta}_T - \boldsymbol{\theta}_S)^T \tilde{\mathbf{V}}_{>n}\tilde{\mathbf{V}}_{>n}^T \mathbf{\Sigma}\tilde{\mathbf{V}}_{>n}\tilde{\mathbf{V}}_{>n}^T(\boldsymbol{\theta}_T - \boldsymbol{\theta}_S)$$

$$= (\boldsymbol{\theta}_T - \boldsymbol{\theta}_S)^T \tilde{\mathbf{V}}_{>n}\tilde{\mathbf{V}}_{>n}^T \mathbf{V}\mathbf{\Lambda}\mathbf{V}^T\tilde{\mathbf{V}}_{>n}\tilde{\mathbf{V}}_{>n}^T(\boldsymbol{\theta}_T - \boldsymbol{\theta}_S),$$

Notice that $\mathbf{P}_\perp \tilde{\mathbf{\Sigma}} \mathbf{P}_\perp = 0$, as was shown in [2]:

$$\mathbf{P}_\perp \tilde{\mathbf{\Sigma}} = \mathbf{P}_\perp \tilde{\mathbf{V}} \tilde{\mathbf{\Lambda}} \tilde{\mathbf{V}}^T = \mathbf{P}_\perp \left( \tilde{\mathbf{V}}_{\leq n} \tilde{\mathbf{\Lambda}}_{\leq n} \tilde{\mathbf{V}}_{\leq n}^T + \tilde{\mathbf{V}}_{>n} \tilde{\mathbf{\Lambda}}_{>n} \tilde{\mathbf{V}}_{>n}^\top \right)$$

$$= \tilde{\mathbf{V}}_{>n} \tilde{\mathbf{V}}_{>n}^T \tilde{\mathbf{V}}_{\leq n} \tilde{\mathbf{\Lambda}}_{\leq n} \tilde{\mathbf{V}}_{\leq n}^T + \tilde{\mathbf{V}}_{>n} \tilde{\mathbf{V}}_{>n}^T \tilde{\mathbf{V}}_{>n} \tilde{\mathbf{\Lambda}}_{>n} \tilde{\mathbf{V}}_{>n}^T \stackrel{(1)}{=} 0$$

where (1) is due to $\tilde{\mathbf{V}}_{>n}, \tilde{\mathbf{V}}_{\leq n}$ being orthogonal and $\tilde{\lambda}_j = 0, \forall j > n$.

Then:

$$R(\boldsymbol{\gamma}) = (\boldsymbol{\theta}_S - \boldsymbol{\theta}_T)^\top \mathbf{P}_\perp \mathbf{\Sigma} \mathbf{P}_\perp (\boldsymbol{\theta}_S - \boldsymbol{\theta}_T)$$

$$= (\boldsymbol{\theta}_S - \boldsymbol{\theta}_T)^\top \mathbf{P}_\perp \left( \mathbf{\Sigma} - \tilde{\mathbf{\Sigma}} \right) \mathbf{P}_\perp (\boldsymbol{\theta}_S - \boldsymbol{\theta}_T)$$

$$= \left\| \left( \mathbf{\Sigma} - \tilde{\mathbf{\Sigma}} \right)^{0.5} \mathbf{P}_\perp (\boldsymbol{\theta}_S - \boldsymbol{\theta}_T) \right\|^2$$

$$\leq \left\| \mathbf{\Sigma} - \tilde{\mathbf{\Sigma}} \right\| \left\| \mathbf{P}_\perp (\boldsymbol{\theta}_S - \boldsymbol{\theta}_T) \right\|^2, \tag{4}$$

where the last inequality is due to the Cauchy-Schwartz inequality.

The next step in the proof is to bound $\|\mathbf{P}_\perp (\boldsymbol{\theta}_S - \boldsymbol{\theta}_T)\|^2$. We start by bounding $\|\mathbf{P}_\perp (\boldsymbol{\theta}_S - \boldsymbol{\theta}_T)\|$ by decomposing $(\boldsymbol{\theta}_T - \boldsymbol{\theta}_S)$ to its top-$k$ span component and bottom-$k$ span component. First notice that since $\mathbf{P}_\perp = \tilde{\mathbf{V}}_{>n} \tilde{\mathbf{V}}_{>n}^T$, $\|\mathbf{P}_\perp (\boldsymbol{\theta}_S - \boldsymbol{\theta}_T)\| = \left\| \tilde{\mathbf{V}}_{>n}^\top (\boldsymbol{\theta}_S - \boldsymbol{\theta}_T) \right\|$, we can write $\forall k \in [d]$:

$$\|\mathbf{P}_\perp (\boldsymbol{\theta}_S - \boldsymbol{\theta}_T)\| = \|\tilde{\mathbf{V}}_{>n}^T (\boldsymbol{\theta}_T - \boldsymbol{\theta}_0)\|$$

$$= \|\tilde{\mathbf{V}}_{>n}^T \mathbf{V} \mathbf{V}^T (\boldsymbol{\theta}_T - \boldsymbol{\theta}_0)\|$$

$$= \|\tilde{\mathbf{V}}_{>n}^T \mathbf{V}_{\leq k} \mathbf{V}_{\leq k}^T (\boldsymbol{\theta}_T - \boldsymbol{\theta}_0) + \tilde{\mathbf{V}}_{>n}^T \mathbf{V}_{>k} \mathbf{V}_{>k}^T (\boldsymbol{\theta}_T - \boldsymbol{\theta}_0)\|$$

$$\leq \|\tilde{\mathbf{V}}_{>n}^T \mathbf{V}_{\leq k}\| \|\mathbf{V}_{\leq k}^T (\boldsymbol{\theta}_T - \boldsymbol{\theta}_0)\| + \|\tilde{\mathbf{V}}_{>n}^T \mathbf{V}_{>k}\| \|\mathbf{V}_{>k}^T (\boldsymbol{\theta}_T - \boldsymbol{\theta}_0)\|, \tag{5}$$

Where the last inequality is due to Cauchy Schwartz for matrix-vector. The last step in the proof is to bound $\|\tilde{\mathbf{V}}_{>n}^T \mathbf{V}_{\leq k}\|$ by using Lemma 1.4 $\forall k \in [d] : \lambda_k > 0$, and bound $\|\tilde{\mathbf{V}}_{>n}^T \mathbf{V}_{>k}\|$ by 1 as follows:

$$\|\tilde{\mathbf{V}}_{>n}^T \mathbf{V}_{>k}\| \leq \|\tilde{\mathbf{V}}_{>n}\| \|\mathbf{V}_{>k}\| \leq 1,$$

due to $\tilde{\mathbf{V}}_{>n}$ and $\mathbf{V}_{>k}$ being orthonormal matrices and because spectral norm is sub-multiplicative. Plugging (5) into (4) gives the inequality:

$$R(\boldsymbol{\gamma}) \leq \left\| \frac{\left\| \mathbf{\Sigma} - \tilde{\mathbf{\Sigma}} \right\|^{3/2}}{\lambda_k} \|\mathbf{P}_{\leq k} (\boldsymbol{\theta}_S - \boldsymbol{\theta}_T)\| + \left\| \mathbf{\Sigma} - \tilde{\mathbf{\Sigma}} \right\|^{1/2} \|\mathbf{P}_{>k} (\boldsymbol{\theta}_S - \boldsymbol{\theta}_T)\| \right\|^2.$$

Since $2a^2 + 2b^2 \geq (a + b)^2$, it follows that:

$$R(\boldsymbol{\gamma}) \leq \frac{2 \left\| \mathbf{\Sigma} - \tilde{\mathbf{\Sigma}} \right\|^3}{\lambda_k^2} \|\mathbf{P}_{\leq k} (\boldsymbol{\theta}_S - \boldsymbol{\theta}_T)\|^2 + 2 \left\| \mathbf{\Sigma} - \tilde{\mathbf{\Sigma}} \right\| \|\mathbf{P}_{>k} (\boldsymbol{\theta}_S - \boldsymbol{\theta}_T)\|^2.$$

To conclude the proof we apply Theorem 1.3 from [5] to provide a high probability bound for $\left\| \mathbf{\Sigma} - \tilde{\mathbf{\Sigma}} \right\|$, as was done in [2]. $\qquad \square$

# 2 Proofs for deep linear networks

In this section we analyze the solution achieved by applying gradient flow optimization to fine-tuning a deep linear regression task (i.e. a regression task using a deep linear network as the regression model).

Our results show that the population risk of a fine-tuned deep linear model depends not only on the source and target tasks and the target covariance, as was shown in the previous section, but also on the depth of the model. We show that as the depth of the model goes to infinity, its population risk depends on the difference between the directions of the source and target task (i.e. the difference between their normalized vectors), instead on the difference between the un-normalized task vectors.

In Section 2.2 this is shown by analysing two settings where this effect is most pronounced: one where we make an assumption on the target task (but not on the target covariance), and one where we make an assumption on the target covariance (but not on the target task).

We conclude in Section 2.3 by showing that fine-tuning only some of the layers can lead to failure to learn.

We begin by recalling some definitions. An $L$-layer linear fully-connected network is defined as

$$\boldsymbol{\beta}(t) = \mathbf{W}_1(t) \cdots \mathbf{W}_{L-1}(t)\mathbf{W}_L(t),$$

where $\mathbf{W}_l \in \mathbb{R}^{d_l \times d_{l+1}}$ for $l \in [L-1]$ (we use $d_1 = d$) and $\mathbf{W}_L \in \mathbb{R}^{d_L}$. Thus, the linear network is equivalent to a linear function with weights $\boldsymbol{\beta}$.

The weights of a deep linear network are called 0-balanced (or perfectly balanced) at time $t$ if:

$$\mathbf{W}_j^\top(t)\mathbf{W}_j(t) = \mathbf{W}_{j+1}(t)\mathbf{W}_{j+1}^\top(t) \quad \text{for} \quad j \in [L-1]. \tag{6}$$

## 2.1 Proof of Theorem 5.2: The inductive bias of deep linear network fine-tuning

For this section, let $\boldsymbol{u}_l$, $\boldsymbol{v}_l$ and $s_l$ denote the top left singular vector, top right singular vector and top singular value of the weights $\mathbf{W}_l$, respectively. Define $t = 0$ as the end of pretraining.

Before proving the theorem, we state several useful lemmas.

**Lemma 2.1.** *Assume that at time $t$ the weights $\mathbf{W}_1(t), \dots, \mathbf{W}_L(t)$ are 0-balanced. Then $\mathbf{W}_l(t) = \boldsymbol{u}_l(t)s_l(t)\boldsymbol{v}_l^\top(t)$,*

$$\boldsymbol{v}_l(t) = \boldsymbol{u}_{l+1}(t), \tag{7}$$

*and:*

$$s_l(t) = \|\boldsymbol{\beta}(t)\|^{1/L} \text{ for } l \in [L]. \tag{8}$$

*Proof for Lemma 2.1.* This proof is a similar to the proof of Theorem 1 in [6]. Focusing on $j = L-1$ balancedness implies that:

$$\mathbf{W}_{L-1}(t)^\top \mathbf{W}_{L-1}(t) = \mathbf{W}_L(t)\mathbf{W}_L(t)^\top.$$

Hence, $\mathbf{W}_{L-1}^\top(t)\mathbf{W}_{L-1}(t)$ is (at most) rank-1 and so is $\mathbf{W}_{L-1}(t)$. By iterating $j$ from $L-2$ to 1, it follows that $\mathbf{W}_l(t)$ is rank-1 for $j \in [L]$.

Consider the SVD of the weights at time $t$. Since all weights are rank-1, they can be decomposed such that

$$\mathbf{W}_l(t) = \boldsymbol{u}_l(t)s_l(t)\boldsymbol{v}_l(t)^\top.$$

Plugging this into (6) it follows that

$$\boldsymbol{v}_j(t)s_j^2(t)\boldsymbol{v}_j^\top(t) = \boldsymbol{u}_{j+1}(t)s_{j+1}^2(t)\boldsymbol{u}_{j+1}^\top(t) \quad \text{for} \quad j \in [L-1],$$

Thus proving (7) and showing that the top singular values of all the layers in time $t$ are equal to each other.[1]

---

[1] maybe add in footnote that because the two matrices have the same SVD, their spectra are equal.

We now consider the norm of the end to end solution at time $t$, $\boldsymbol{\beta}(t)$:

$$\|\boldsymbol{\beta}(t)\| = \|\mathbf{W}_1(t)\cdots\mathbf{W}_L(t)\|$$
$$= \|\boldsymbol{u}_1(t)s_1(t)\boldsymbol{v}_1^\top s_2(t)\cdots s_L(t)\|$$
$$= \|\boldsymbol{u}_1(t)\prod_{i=1}^{L} s_l(t)\| = \prod_{i=1}^{L} s_l(t)\|\boldsymbol{u}_1(t)\| = \prod_{i=1}^{L} s_l(t).$$

Since all of the top singular values at time $t$ equal each other, and $\|u_1\| = 1$ by construction, the result follows. $\qquad\square$

The following Lemma is also used in the analysis:

**Lemma 2.2** (Theorem 1 from [6]). *Suppose a deep linear network is optimized using GF, starting from a 0-balanced initialization, i.e. initialization in which weights are 0-balanced. Then the weights stay balanced throughout optimization.*

We are now ready to prove the theorem.

*Proof of Theorem 5.2.* First consider the pretraining of the model under Assumption 3.2. Assume that before the pretraining, the model weights are perfectly balanced. From Lemma 2.2 it follows that after pretraining on the source task, i.e. at $t = 0$, the weights of the model are still balanced. From Lemma 2.1, this means they are also rank-1. From Assumption 3.2:

$$\mathbf{X}_S\beta(0) = \mathbf{y}_S,$$

and since $n_S > d$ this implies:

$$\boldsymbol{\beta}(0) = \boldsymbol{\theta}_S. \tag{9}$$

Lemma 2.1 gives us that:

$$\boldsymbol{\beta}(0) = \mathbf{W}_1(0)\cdots\mathbf{W}_L(0) = \boldsymbol{u}_1(0)\prod_{i=1}^{L} s_l(0) = \boldsymbol{u}_1(0)s_1^L(0),$$

Hence:

$$\boldsymbol{u}_1(0) = \frac{\boldsymbol{\theta}_S}{\|\boldsymbol{\theta}_S\|},$$

and

$$s_1(0) = \|\boldsymbol{\theta}_S\|^{1/L}, \tag{10}$$

Hence:

$$\mathbf{W}_1(0) = \boldsymbol{u}_1(0)s_1(0)\boldsymbol{v}_1^\top(0) = \frac{\boldsymbol{\theta}_S}{\|\boldsymbol{\theta}_S\|}\|\boldsymbol{\theta}_S\|^{1/L}\boldsymbol{v}_1^\top(0) = \frac{\boldsymbol{\theta}_S}{\|\boldsymbol{\theta}_S\|^{(L-1)/L}}\boldsymbol{v}_1^\top(0). \tag{11}$$

We next analyze the fine-tuning dynamics. Lemma 2.2 ensures that if the pretrained model has 0-balanced weights, then the weights will remain 0-balanced during finetune. This implies that Lemma 2.1 holds for all $t \geq 0$.

Observe the gradient flow dynamics of the layers during fine-tuning:

$$\dot{\mathbf{W}}_l(t) = -\mathbf{W}_{l-1}^T(t)\cdots\mathbf{W}_1^T(t)\mathbf{X}^T\boldsymbol{r}(t)\mathbf{W}_L^T(t)\cdots\mathbf{W}_{l+1}^T(t) \text{ for } l \in [L],$$

where $\boldsymbol{r}(t) \in \mathbb{R}^n$ is the residual vector satisfying $[\boldsymbol{r}]_i = \mathbf{x}_i^\top\boldsymbol{\beta}(t) - \mathbf{y}_i$.

From Lemma 2.1:

$$\dot{\mathbf{W}}_l(t) = - \boldsymbol{v}_{l-1}(t)s_{l-1}(t)\boldsymbol{u}_{l-1}^T(t)\boldsymbol{v}_{l-2}(t)s_{l-2}(t)\boldsymbol{u}_{l-2}^T(t)\cdots$$
$$\boldsymbol{v}_1(t)s_1(t)\boldsymbol{u}_1^T(t)\mathbf{X}^T\boldsymbol{r}(t)\boldsymbol{v}_L(t)s_{L-1}(t)\boldsymbol{u}_L^T(t)\cdots$$
$$\boldsymbol{v}_{l+1}(t)s_{l+1}(t)\boldsymbol{u}_{l+1}^T(t) \text{ for } l \in [L].$$

Using (7) and (8) it follows that $\forall t \geq 0$:

$$\dot{\mathbf{W}}_l(t) = -\boldsymbol{v}_{l-1}(t)\left(\prod_{i=1}^{l-1} s_i(t)\right)\boldsymbol{u}_1(t)^T\mathbf{X}^T\boldsymbol{r}(t)\left(\prod_{i=l+1}^{L} s_i(t)\right)\boldsymbol{u}_{l+1}^T(t) \text{ for } l \in [L]$$

$$= -\boldsymbol{v}_{l-1}(t)s^{l-1}(t)\boldsymbol{u}_1^T(t)\mathbf{X}^T\boldsymbol{r}(t)s^{L-l}(t)\boldsymbol{u}_{l+1}^T(t) \text{ for } l \in [L].$$

For $\mathbf{W}_1$,

$$\dot{\mathbf{W}}_1(t) = -\mathbf{X}^T\boldsymbol{r}(t)s^{L-1}(t)\boldsymbol{u}_2^T(t) = -\mathbf{X}^T\boldsymbol{r}(t)s^{L-1}(t)\boldsymbol{v}_1^T(t), \tag{12}$$

Where the last equality is due to (7). Hence $\dot{\mathbf{W}}_1$ is always a rank-1 matrix whose columns are in the row space of $\mathbf{X}$. This implies that the decomposition $\mathbf{W}_1$ into two orthogonal components $\mathbf{W}_1^\perp$ and $\mathbf{W}_1^\parallel$ so that $\mathbf{W}_1^\parallel = \mathbf{P}_\parallel\mathbf{W}_1$ and $\mathbf{W}_1^\perp = \mathbf{P}_\perp\mathbf{W}_1$ yields that $\forall t \geq 0$ it follows that

$$\dot{\mathbf{W}}_1^\perp(t) = \mathbf{0},$$

$$\dot{\mathbf{W}}_1^\parallel(t) = \dot{\mathbf{W}}_1(t) = \mathbf{X}^T\boldsymbol{r}(t)s^{L-1}(t)\boldsymbol{v}_1^T(t).$$

Hence, $\mathbf{W}_1^\perp(t)$ does not change for all $t \geq 0$. Using (11) it follows:

$$\mathbf{W}_1^\perp(t) = \mathbf{W}_1^\perp(0) \tag{13}$$

$$= \mathbf{P}_\perp\left(\frac{\boldsymbol{\theta}_S}{\|\boldsymbol{\theta}_S\|^{\frac{L-1}{L}}}\boldsymbol{v}_1^\top(0)\right)$$

$$= \frac{\mathbf{P}_\perp\boldsymbol{\theta}_S}{\|\boldsymbol{\theta}_S\|^{\frac{L-1}{L}}}\boldsymbol{v}_1^\top(0). \tag{14}$$

The next lemma states that $\boldsymbol{v}_1(t)$ does not change during optimization if $\|\mathbf{P}_\perp\mathbf{W}_1(0)\|_F > 0$.

**Lemma 2.3.** *Suppose we run GF over a deep linear network starting from 0-balanced initialization. Also assume that at initialization $\mathbf{W}_1(0)$ is rank-1 and:*

$$\|\mathbf{P}_\perp\mathbf{W}_1(0)\|_F > 0,$$

*Then for all $t > 0$:*

$$\boldsymbol{v}_1(t) = \boldsymbol{v}_1(0).$$

*Proof.* Assume towards contradiction that there exists $t > 0$ s.t. $\boldsymbol{v}_1(t) \neq \boldsymbol{v}_1(0)$.
From $\mathbf{W}_1(t)$ being rank-1 (Lemma 2.1), it follows that

$$\mathbf{P}_\perp\mathbf{W}_1(t) = \mathbf{P}_\perp\boldsymbol{u}_1(t)s(t)\boldsymbol{v}_1^\top(t) = \left(\mathbf{P}_\perp\boldsymbol{u}_1(t)s(t)\right)\boldsymbol{v}_1^\top(t),$$

And from the decomposition of $\mathbf{W}_1(t)$ to $\mathbf{W}_1^\parallel(t)$ and $\mathbf{W}_1^\perp(t)$, (13) and $\mathbf{W}_1(0)$ being rank-1 it follows that:

$$\mathbf{P}_\perp\mathbf{W}_1(t) = \mathbf{W}_1^\perp(t) = \mathbf{W}_1^\perp(0) = \mathbf{P}_\perp\boldsymbol{u}_1(0)s_1(0)\boldsymbol{v}_1^\top(0),$$

Hence:

$$\left(\mathbf{P}_\perp\boldsymbol{u}_1(t)s(t)\right)\boldsymbol{v}_1^\top(t) = \left(\mathbf{P}_\perp\boldsymbol{u}_1(0)s_1(0)\right)\boldsymbol{v}_1^\top(0).$$

From (12) we see that the orthogonal part of $\boldsymbol{u}_1(t)$ does not change during fine-tune:

$$\dot{\boldsymbol{u}}_1(t) = \dot{\mathbf{W}}_1(t) \cdot \frac{\partial\mathbf{W}_1(t)}{\partial\boldsymbol{u}_1(t)} = -\mathbf{X}^T\boldsymbol{r}(t)s^{L-1}(t)\boldsymbol{v}_1^T(t)\boldsymbol{v}_1(t)s(t) = -\mathbf{X}^T\boldsymbol{r}(t)s^L(t)$$

hence:

$$\mathbf{P}_\perp\dot{\boldsymbol{u}}_1(t) = 0 \Rightarrow \mathbf{P}_\perp\boldsymbol{u}_1(t) = \mathbf{P}_\perp\boldsymbol{u}_1(0). \tag{15}$$

Since $\boldsymbol{v}_1(t) \neq \boldsymbol{v}_1(0)$, and because non-degenerate singular values always have unique left and right singular vectors (up to a sign), $\mathbf{W}_1^\perp(t) = \mathbf{W}_1^\perp(0)$ only if:

$$s(t) = s_1(0) = 0,$$

by contradiction to the assumption that $s_1(0) = \|\mathbf{P}_\perp\mathbf{W}_1(0)\|_F > 0$, or if $\boldsymbol{v}_1(t) = -\boldsymbol{v}_1(0)$ and $\mathbf{P}_\perp\boldsymbol{u}_1(t) = -\mathbf{P}_\perp\boldsymbol{u}_1(0)$, which contradicts (15). $\qquad\square$

In the case where $\|\mathbf{P}_\perp\mathbf{W}_1(0)\|_F = 0$, since $\mathbf{P}_\perp\mathbf{W}_1(t) = \mathbf{P}_\perp\mathbf{W}_1(0)$, it follows that $\mathbf{W}_1(t) = \mathbf{P}_\|\mathbf{W}_1(t)$, which is similar to the case in [7], for which the solution is known to be $\mathbf{P}_\|\boldsymbol{\theta}_T$. Also, from (14), this implies $\mathbf{P}_\perp\boldsymbol{\theta}_S = 0$, and the expression for the end-to-end solution in Eq.5 in the main text holds.

The analysis continues for $\|\mathbf{P}_\perp\mathbf{W}_1(0)\|_F > 0$. By using Lemma 2.1 and Lemma 2.3 it follows that:

$$
\begin{aligned}
\mathbf{W}_1^\perp(t)\mathbf{W}_2(t)\cdots\mathbf{W}_L(t) &\overset{(1)}{=} \mathbf{W}_1^\perp(0)\mathbf{W}_2(t)\cdots\mathbf{W}_L(t) \\
&\overset{(2)}{=} \frac{\mathbf{P}_\perp\boldsymbol{\theta}_S}{\|\boldsymbol{\theta}_S\|^{\frac{L-1}{L}}}\boldsymbol{v}_1^\top(0)\mathbf{W}_2(t)\cdots\mathbf{W}_L(t) \\
&\overset{(3)}{=} \frac{\mathbf{P}_\perp\boldsymbol{\theta}_S}{\|\boldsymbol{\theta}_S\|^{\frac{L-1}{L}}}\boldsymbol{v}_1^\top(t)\mathbf{W}_2(t)\cdots\mathbf{W}_L(t) \\
&= \frac{\mathbf{P}_\perp\boldsymbol{\theta}_S}{\|\boldsymbol{\theta}_S\|^{\frac{L-1}{L}}}\boldsymbol{v}_1^\top(t)\boldsymbol{u}_2(t)\|\boldsymbol{\beta}(t)\|^{\frac{L-1}{L}} \\
&\overset{(4)}{=} \frac{\mathbf{P}_\perp\boldsymbol{\theta}_S}{\|\boldsymbol{\theta}_S\|^{\frac{L-1}{L}}}\boldsymbol{v}_1^\top(t)\boldsymbol{v}_1(t)\|\boldsymbol{\beta}(t)\|^{\frac{L-1}{L}} \\
&= \left(\frac{\|\boldsymbol{\beta}(t)\|}{\|\boldsymbol{\theta}_S\|}\right)^{\frac{L-1}{L}}\mathbf{P}_\perp\boldsymbol{\theta}_S.
\end{aligned}
\tag{16}
$$

With (1) due to (13), (2) due to (14), (3) due to Lemma 2.3 and (4) due to Lemma 2.1. From the requirement of Assumption 3.3 that $\lim_{t\to\infty}\mathbf{X}\boldsymbol{\beta}(t) = \mathbf{y}$, it follows that:

$$
\begin{aligned}
&\lim_{t\to\infty}\mathbf{X}\mathbf{W}_1(t)\cdots\mathbf{W}_L(t) = \mathbf{y} \\
\Rightarrow &\lim_{t\to\infty}\mathbf{X}\mathbf{W}_1^\|(t)\cdot\mathbf{W}_2(t)\cdots\mathbf{W}_L(t) = \mathbf{y} \\
\Rightarrow &\lim_{t\to\infty}\mathbf{W}_1^\|(t)\cdot\mathbf{W}_2(t)\cdots\mathbf{W}_L(t) = \mathbf{X}^T\left(\mathbf{X}\mathbf{X}^T\right)^{-1}\mathbf{y},
\end{aligned}
\tag{17}
$$

Which is the only solution for this equation in the span of $\mathbf{X}$, and due to Assumption 3.1.
Eq.5 in the main text follows from (16) and (17):

$$
\begin{aligned}
\lim_{t\to\infty}\boldsymbol{\beta}(t) &= \lim_{t\to\infty}\mathbf{W}_1(t)\cdot\mathbf{W}_2(t)\cdots\mathbf{W}_L(t) \\
&= \lim_{t\to\infty}\left(\mathbf{W}_1^\|(t) + \mathbf{W}_1^\perp(t)\right)\cdot\mathbf{W}_2(t)\cdots\mathbf{W}_L(t) \\
&= \lim_{t\to\infty}\mathbf{W}_1^\perp(t)\cdot\mathbf{W}_2(t)\cdots\mathbf{W}_L(t) + \mathbf{W}_1^\|(t)\cdot\mathbf{W}_2(t)\cdots\mathbf{W}_L(t) \\
&= \left(\frac{\|\lim_{t\to\infty}\boldsymbol{\beta}(t)\|}{\|\boldsymbol{\theta}_S\|}\right)^{\frac{L-1}{L}}\mathbf{P}_\perp\boldsymbol{\theta}_S + \mathbf{P}_\|\boldsymbol{\theta}_T.
\end{aligned}
\tag{18}
$$

To prove Eq.6 from the main text, consider the norm of $\lim_{t\to\infty}\boldsymbol{\beta}(t)$.

$$
\begin{aligned}
\|\lim_{t\to\infty}\boldsymbol{\beta}(t)\| &= \sqrt{\left(\frac{\|\lim_{t\to\infty}\boldsymbol{\beta}(t)\|}{\|\boldsymbol{\theta}_S\|}\right)^{\frac{2(L-1)}{L}}\|\mathbf{P}_\perp\boldsymbol{\theta}_S\|^2 + \|\mathbf{P}_\|\boldsymbol{\theta}_T\|^2} \\
\Rightarrow \|\lim_{t\to\infty}\boldsymbol{\beta}(t)\|^2 &= \left(\frac{\|\lim_{t\to\infty}\boldsymbol{\beta}(t)\|}{\|\boldsymbol{\theta}_S\|}\right)^{\frac{2(L-1)}{L}}\|\mathbf{P}_\perp\boldsymbol{\theta}_S\|^2 + \|\mathbf{P}_\|\boldsymbol{\theta}_T\|^2 \\
\Rightarrow \|\lim_{t\to\infty}\boldsymbol{\beta}(t)\|^2 &- \left(\frac{\|\lim_{t\to\infty}\boldsymbol{\beta}(t)\|}{\|\boldsymbol{\theta}_S\|}\right)^{\frac{2(L-1)}{L}}\|\mathbf{P}_\perp\boldsymbol{\theta}_S\|^2 - \|\mathbf{P}_\|\boldsymbol{\theta}_T\|^2 = 0.
\end{aligned}
$$

At the limit $L \to \infty$ we get:

$$\lim_{l \to \infty} \left( \| \lim_{t \to \infty} \boldsymbol{\beta}(t) \|^2 - \left( \frac{\| \lim_{t \to \infty} \boldsymbol{\beta}(t) \|}{\|\boldsymbol{\theta}_S\|} \right)^{\frac{2(L-1)}{L}} \|\mathbf{P}_\perp \boldsymbol{\theta}_S\|^2 - \|\mathbf{P}_\| \boldsymbol{\theta}_T\|^2 \right)$$

$$= \| \lim_{l \to \infty} \lim_{t \to \infty} \boldsymbol{\beta}(t) \|^2 - \left( \frac{\| \lim_{l \to \infty} \lim_{t \to \infty} \boldsymbol{\beta}(t) \|}{\|\boldsymbol{\theta}_S\|} \right)^2 \|\mathbf{P}_\perp \boldsymbol{\theta}_S\|^2 - \|\mathbf{P}_\| \boldsymbol{\theta}_T\|^2 = 0$$

$$\Rightarrow \frac{\| \lim_{l \to \infty} \lim_{t \to \infty} \boldsymbol{\beta}(t) \|^2}{\|\boldsymbol{\theta}_S\|^2} \left( \|\boldsymbol{\theta}_S\|^2 - \|\mathbf{P}_\perp \boldsymbol{\theta}_S\|^2 \right) = \|\mathbf{P}_\| \boldsymbol{\theta}_T\|^2,$$

Thus:

$$\frac{\| \lim_{l \to \infty} \lim_{t \to \infty} \boldsymbol{\beta}(t) \|}{\|\boldsymbol{\theta}_S\|} = \frac{\|\mathbf{P}_\| \boldsymbol{\theta}_T\|}{\sqrt{\|\boldsymbol{\theta}_S\|^2 - \|\mathbf{P}_\perp \boldsymbol{\theta}_S\|^2}} = \frac{\|\mathbf{P}_\| \boldsymbol{\theta}_T\|}{\|\mathbf{P}_\| \boldsymbol{\theta}_S\|}.$$

And it follows that at this limit:

$$\lim_{L \to \infty} \lim_{t \to \infty} \boldsymbol{\beta}(t) = \frac{\|\mathbf{P}_\| \boldsymbol{\theta}_T\|}{\|\mathbf{P}_\| \boldsymbol{\theta}_S\|} \mathbf{P}_\perp \boldsymbol{\theta}_S + \mathbf{P}_\| \boldsymbol{\theta}_T. \tag{19}$$

$\square$

From the same lines of proof as in Section 1.1 it follows that

**Corollary 2.4.** *For the conditions in Theorem 5.2 in the main text,*

$$R\left( \lim_{L \to \infty} \lim_{t \to \infty} \boldsymbol{\beta}(t) \right) = \left\| \boldsymbol{\Sigma}^{0.5} \left( \mathbf{P}_\perp \left( \boldsymbol{\theta}_T - \frac{\|\mathbf{P}_\| \boldsymbol{\theta}_T\|}{\|\mathbf{P}_\| \boldsymbol{\theta}_S\|} \boldsymbol{\theta}_S \right) \right) \right\|^2.$$

## 2.2 Proofs of Theorems 5.3 and 5.4: How does depth affect the population risk?

Corollary 2.4 above contains dependence on $\mathbf{P}_\|$ which is a random variable. We next provide high-probability risk bounds that can be derived from this result. The bounds are obtained under slightly different assumptions, either on the target task or on the target distribution, but both highlight the fact that fine-tuning in the $L \to \infty$ case will depend on $\hat{\boldsymbol{\theta}}_S - \hat{\boldsymbol{\theta}}_T$ rather than the un-normalized $\boldsymbol{\theta}_S - \boldsymbol{\theta}_T$.

Recall the definition of the fine-tuning solution as $L \to \infty$:

$$\boldsymbol{\beta} \triangleq \lim_{L \to \infty} \lim_{t \to \infty} \boldsymbol{\beta}(t).$$

In the first setting we will assume that $\boldsymbol{\theta}_T$ is a scaled version of $\boldsymbol{\theta}_S$, without any assumptions on $\mathcal{D}$. Theorem 5.3 from the main text demonstrates a gap between perfect fine-tuning for the $L \to \infty$ case and non-zero fine-tuning error for $L = 1$.

*Proof of Theorem 5.3 (Main Text).* First notice:

$$\frac{\|\mathbf{P}_\| \boldsymbol{\theta}_T\|}{\|\mathbf{P}_\| \boldsymbol{\theta}_S\|} = \frac{\|\mathbf{P}_\| \alpha \boldsymbol{\theta}_S\|}{\|\mathbf{P}_\| \boldsymbol{\theta}_S\|} = \alpha \frac{\|\mathbf{P}_\| \boldsymbol{\theta}_S\|}{\|\mathbf{P}_\| \boldsymbol{\theta}_S\|} = \alpha, \tag{20}$$

which from Eq.6 in the main text gives the solution

$$\boldsymbol{\beta} = \alpha \mathbf{P}_\perp \boldsymbol{\theta}_S + \mathbf{P}_\| \boldsymbol{\theta}_T = \mathbf{P}_\perp \boldsymbol{\theta}_T + \mathbf{P}_\| \boldsymbol{\theta}_T = \boldsymbol{\theta}_T.$$

On the other hand, for the $L = 1$ solution $\boldsymbol{\gamma}$ it follows from Eq.2 in the main text that

$$\left\| \boldsymbol{\Sigma}^{0.5} \mathbf{P}_\perp (\boldsymbol{\theta}_T - \boldsymbol{\theta}_S) \right\|^2 = \left\| \boldsymbol{\Sigma}^{0.5} \mathbf{P}_\perp \left( \boldsymbol{\theta}_T - \frac{\boldsymbol{\theta}_T}{\alpha} \right) \right\|^2$$

$$= \left( \frac{\alpha - 1}{\alpha} \right)^2 \left\| \boldsymbol{\Sigma}^{0.5} \mathbf{P}_\perp \boldsymbol{\theta}_T \right\|^2,$$

which is greater than zero for all $\alpha \neq 1$. $\square$

In the second setting we assume that $\mathcal{D} = \mathcal{N}(0,1)^d$, without any assumptions on $\boldsymbol{\theta}_T$. Here it shows that while the population risk of the $L = 1$ solution depends on $\|\boldsymbol{\theta}_T - \boldsymbol{\theta}_S\|$, the population risk of the infinitely-deep linear solution depends on the normalized $\left\|\hat{\boldsymbol{\theta}}_T - \hat{\boldsymbol{\theta}}_S\right\|$ and $\|\boldsymbol{\theta}_T\|$, i.e. on the alignment of $\boldsymbol{\theta}_T$ and $\boldsymbol{\theta}_S$ and the norm of $\boldsymbol{\theta}_T$.

**Theorem 5.4** (Main Text). *Assume that the conditions of Theorem 5.2 hold, and let $\mathbf{X} \sim \mathcal{N}(0,1)^d$. Suppose $n \leq d$, then there exists a constant $c > 0$ such that for an $\epsilon > 0$ it holds that with probability at least $1 - 4\exp(-c\epsilon^2 n) - 4\exp\left(-c\epsilon^2(d-n)\right)$ the population risk for the $L \to \infty$ end-to-end $\boldsymbol{\beta}$ is bounded:*

$$R(\boldsymbol{\beta}) \leq \frac{d-n}{d}(1+\epsilon)^2 \|\boldsymbol{\theta}_T\|^2 \left\|\hat{\boldsymbol{\theta}}_T - \hat{\boldsymbol{\theta}}_S\right\|^2 + \frac{d-n}{d}\zeta(\|\boldsymbol{\theta}_T\|)^2, \tag{21}$$

*for $\zeta(\|\boldsymbol{\theta}_T\|) \approx \epsilon \|\boldsymbol{\theta}_T\|$. For the $L = 1$ linear regression solution $\boldsymbol{\gamma}$ this risk is bounded by*

$$R(\boldsymbol{\gamma}) \leq \frac{d-n}{d}(1+\epsilon)^2 \|\boldsymbol{\theta}_T - \boldsymbol{\theta}_S\|^2 .$$

*Proof of Theorem 5.5 (Main Text).* We start by analyzing $R(\boldsymbol{\beta})$:

$$R(\boldsymbol{\beta}) = \left\| \boldsymbol{\Sigma}^{0.5} \mathbf{P}_\perp \left( \boldsymbol{\theta}_T - \frac{\|\mathbf{P}_\| \boldsymbol{\theta}_T\|}{\|\mathbf{P}_\| \boldsymbol{\theta}_S\|} \boldsymbol{\theta}_S \right) \right\|^2$$

$$\overset{(1)}{=} \left\| \mathbf{I}^{0.5} \mathbf{P}_\perp \left( \boldsymbol{\theta}_T - \frac{\|\mathbf{P}_\| \boldsymbol{\theta}_T\|}{\|\mathbf{P}_\| \boldsymbol{\theta}_S\|} \boldsymbol{\theta}_S \right) \right\|^2$$

$$= \left\| \mathbf{P}_\perp \left( \boldsymbol{\theta}_T - \frac{\|\mathbf{P}_\perp \boldsymbol{\theta}_T\|}{\|\mathbf{P}_\| \boldsymbol{\theta}_S\|} \boldsymbol{\theta}_S \right) \right\|^2,$$

where (1) is due to $\boldsymbol{\Sigma} = \mathbf{I}$ from the definition of the distribution of $\mathbf{X}$. We then bound the RHS with:

$$\left\| \mathbf{P}_\perp \left( \boldsymbol{\theta}_T - \frac{\|\mathbf{P}_\| \boldsymbol{\theta}_T\|}{\|\mathbf{P}_\| \boldsymbol{\theta}_S\|} \boldsymbol{\theta}_S \right) \right\|^2$$

$$\leq \left\| \mathbf{P}_\perp \left( \boldsymbol{\theta}_T - \frac{\|\mathbf{P}_\| \boldsymbol{\theta}_T\|}{\|\mathbf{P}_\| \boldsymbol{\theta}_S\|} \boldsymbol{\theta}_S \right) - \mathbf{P}_\perp \left( \|\boldsymbol{\theta}_T\| (\hat{\boldsymbol{\theta}}_T - \hat{\boldsymbol{\theta}}_S) \right) + \mathbf{P}_\perp \left( \|\boldsymbol{\theta}_T\| (\hat{\boldsymbol{\theta}}_T - \hat{\boldsymbol{\theta}}_S) \right) \right\|^2$$

$$\leq \left\| \mathbf{P}_\perp \left( \boldsymbol{\theta}_T - \frac{\|\mathbf{P}_\| \boldsymbol{\theta}_T\|}{\|\mathbf{P}_\| \boldsymbol{\theta}_S\|} \boldsymbol{\theta}_S \right) - \mathbf{P}_\perp \left( \|\boldsymbol{\theta}_T\| (\hat{\boldsymbol{\theta}}_T - \hat{\boldsymbol{\theta}}_S) \right) \right\|^2 + \left\| \mathbf{P}_\perp \left( \|\boldsymbol{\theta}_T\| (\hat{\boldsymbol{\theta}}_T - \hat{\boldsymbol{\theta}}_S) \right) \right\|^2 .$$

We see that we can bound the expression on the left:

$$\left\| \mathbf{P}_\perp \left( \boldsymbol{\theta}_T - \frac{\|\mathbf{P}_\| \boldsymbol{\theta}_T\|}{\|\mathbf{P}_\| \boldsymbol{\theta}_S\|} \boldsymbol{\theta}_S \right) - \mathbf{P}_\perp \left( \|\boldsymbol{\theta}_T\| (\hat{\boldsymbol{\theta}}_T - \hat{\boldsymbol{\theta}}_S) \right) \right\|^2$$

$$= \left\| \mathbf{P}_\perp \left( \boldsymbol{\theta}_T - \frac{\|\mathbf{P}_\| \boldsymbol{\theta}_T\|}{\|\mathbf{P}_\| \boldsymbol{\theta}_S\|} \boldsymbol{\theta}_S - \boldsymbol{\theta}_T + \|\boldsymbol{\theta}_T\| \hat{\boldsymbol{\theta}}_S \right) \right\|^2$$

$$= \left\| \mathbf{P}_\perp \left( \frac{\|\boldsymbol{\theta}_T\|}{\|\boldsymbol{\theta}_S\|} \boldsymbol{\theta}_S - \frac{\|\mathbf{P}_\| \boldsymbol{\theta}_T\|}{\|\mathbf{P}_\| \boldsymbol{\theta}_S\|} \boldsymbol{\theta}_S \right) \right\|^2$$

$$\leq \left\| \mathbf{P}_\perp \boldsymbol{\theta}_S \left( \frac{\|\boldsymbol{\theta}_T\|}{\|\boldsymbol{\theta}_S\|} - \frac{\|\mathbf{P}_\| \boldsymbol{\theta}_T\|}{\|\mathbf{P}_\| \boldsymbol{\theta}_S\|} \right) \right\|^2$$

$$\leq \|\mathbf{P}_\perp \boldsymbol{\theta}_S\|^2 \left\| \frac{\|\boldsymbol{\theta}_T\|}{\|\boldsymbol{\theta}_S\|} - \frac{\|\mathbf{P}_\| \boldsymbol{\theta}_T\|}{\|\mathbf{P}_\| \boldsymbol{\theta}_S\|} \right\|^2$$

Let $\mathbf{P}_\parallel$ be the projection matrix onto the row space of $\mathbf{X}$, then from [8], $\mathbf{P}_\parallel$ is a projection onto a random $n$-dimensional subspace uniformly distributed in the Grassmannian $\mathbf{G}_{d,n}$, and $\mathbf{P}_\perp$ is a projection onto a random $d - n$-dimensional subspace uniformly distributed in the Grassmannian $\mathbf{G}_{d,d-n}$.

According to Lemma 5.3.2 in [9], with probability at least $1 - 4\exp(-c\epsilon^2 n)$

$$\frac{1 - \epsilon}{1 + \epsilon} \frac{\|\boldsymbol{\theta}_T\|}{\|\boldsymbol{\theta}_S\|} \leq \frac{\|\mathbf{P}_\parallel \boldsymbol{\theta}_T\|}{\|\mathbf{P}_\parallel \boldsymbol{\theta}_S\|} \leq \frac{1 + \epsilon}{1 - \epsilon} \frac{\|\boldsymbol{\theta}_T\|}{\|\boldsymbol{\theta}_S\|},$$

which bounds:

$$\left\| \frac{\|\boldsymbol{\theta}_T\|}{\|\boldsymbol{\theta}_S\|} - \frac{\|\mathbf{P}_\parallel \boldsymbol{\theta}_T\|}{\|\mathbf{P}_\parallel \boldsymbol{\theta}_S\|} \right\|^2 \leq \left\| \frac{\|\boldsymbol{\theta}_T\|}{\|\boldsymbol{\theta}_S\|} - \frac{1 + \epsilon}{1 - \epsilon} \frac{\|\boldsymbol{\theta}_T\|}{\|\boldsymbol{\theta}_S\|} \right\|^2$$

$$= \left( \frac{\|\boldsymbol{\theta}_T\|}{\|\boldsymbol{\theta}_S\|} \right)^2 \frac{4\epsilon^2}{(1 - \epsilon)^2}.$$

Again, by applying Lemma 5.3.2 from [9], with probability at least $1 - 4\exp\left(-c\epsilon^2(d - n)\right) - 2\exp\left(-c\epsilon^2(d - n)\right)$ :

$$\|\mathbf{P}_\perp \boldsymbol{\theta}_S\|^2 \leq (1 + \epsilon)^2 \frac{d - n}{d} \|\boldsymbol{\theta}_S\|^2,$$

$$\left\| \mathbf{P}_\perp \|\boldsymbol{\theta}_T\| \left(\hat{\boldsymbol{\theta}}_T - \hat{\boldsymbol{\theta}}_S\right) \right\|^2 \leq (1 + \epsilon)^2 \frac{d - n}{d} \left\| \|\boldsymbol{\theta}_T\| \left(\hat{\boldsymbol{\theta}}_T - \hat{\boldsymbol{\theta}}_S\right) \right\|^2.$$

Thus the following bound is obtained:

$$R(\boldsymbol{\beta}) \leq \left\| \mathbf{P}_\perp \left( \boldsymbol{\theta}_T - \frac{\|\mathbf{P}_\parallel \boldsymbol{\theta}_T\|}{\|\mathbf{P}_\parallel \boldsymbol{\theta}_S\|} \boldsymbol{\theta}_S \right) - \mathbf{P}_\perp \left( \|\boldsymbol{\theta}_T\| (\hat{\boldsymbol{\theta}}_T - \hat{\boldsymbol{\theta}}_S) \right) \right\|^2 + \left\| \mathbf{P}_\perp \left( \|\boldsymbol{\theta}_T\| (\hat{\boldsymbol{\theta}}_T - \hat{\boldsymbol{\theta}}_S) \right) \right\|^2$$

$$\leq (1 + \epsilon)^2 \frac{d - n}{d} \left\| \|\boldsymbol{\theta}_T\| (\hat{\boldsymbol{\theta}}_T - \hat{\boldsymbol{\theta}}_S) \right\|^2 + \frac{4\epsilon^2(1 + \epsilon)^2}{(1 - \epsilon)^2} \frac{d - n}{d} \|\boldsymbol{\theta}_S\|^2 \frac{\|\boldsymbol{\theta}_T\|^2}{\|\boldsymbol{\theta}_S\|^2}$$

$$= (1 + \epsilon)^2 \frac{d - n}{d} \left\| \|\boldsymbol{\theta}_T\| (\hat{\boldsymbol{\theta}}_T - \hat{\boldsymbol{\theta}}_S) \right\|^2 + \frac{4\epsilon^2(1 + \epsilon)^2}{(1 - \epsilon)^2} \frac{d - n}{d} \|\boldsymbol{\theta}_T\|^2.$$

Define $\zeta(\|\boldsymbol{\theta}_T\|) = \frac{2\epsilon(1+\epsilon)}{(1-\epsilon)} \|\boldsymbol{\theta}_T\|$, which concludes the proof for the infinite depth case.

Now for the upper bound of the population risk of the $L = 1$ solution $\boldsymbol{\gamma}$. Look at Eq.2, and from $\mathbf{P}_\perp$ being a random projection, it follows that with probability at least $1 - 2\exp\left(-c\epsilon^2(d - n)\right)$:

$$R(\boldsymbol{\gamma}) \leq \left\| \boldsymbol{\Sigma}^{0.5} \mathbf{P}_\perp (\boldsymbol{\theta}_T - \boldsymbol{\theta}_S) \right\|^2$$

$$= \left\| \mathbf{I} \mathbf{P}_\perp (\boldsymbol{\theta}_T - \boldsymbol{\theta}_S) \right\|^2$$

$$\leq (1 + \epsilon)^2 \frac{d - n}{d} \|\boldsymbol{\theta}_T - \boldsymbol{\theta}_S\|^2.$$

$\square$

## 2.3 Proof of Theorem 5.5: The effect of fixing layers during fine-tuning

*Proof.* Since we assume that the weights before pretraining are 0-balanced, it follows from Lemma 2.1 and Lemma 2.2 that all layers $\mathbf{W}_1(t), \ldots \mathbf{W}_k(t)$ are rank-1. From Assumption 3.2 it follows that at the end of pretraining $\boldsymbol{\beta}(0) = \boldsymbol{\theta}_S$, and from (11) it follows that $u_1(0) = \hat{\boldsymbol{\theta}}_S$.

Consider the setting where the first $k$ layers are fixed. It follows that

$$\mathbf{W}_i(t) = \mathbf{W}_i(0) \quad \forall t \geq 0, \quad 0 \leq i \leq k.$$

Then from Lemma 2.1 it follows that for $t \geq 0$ and for any $\mathbf{x} \in \mathbb{R}^d$:

$$\mathbf{x}^\top \mathbf{W}_1(t) \cdots \mathbf{W}_k(t) = \mathbf{x}^\top \mathbf{W}_1(0) \cdots \mathbf{W}_k(0) = \mathbf{x}^\top \boldsymbol{u}_1(0) \prod_{i=1}^k s_i \boldsymbol{v}_k^\top(0)$$

$$= \mathbf{x}^\top \|\boldsymbol{\theta}_S\|^{k/L} \boldsymbol{u}_1(0) \|\boldsymbol{\theta}_S\|^{k/L} \boldsymbol{v}_k^\top(0)$$

$$= \mathbf{x}^\top \boldsymbol{\theta}_S \|\boldsymbol{\theta}_S\|^{k-L/L} \boldsymbol{v}_k^\top(0) = \|\boldsymbol{\theta}_S\|^{k-L/L} \langle \mathbf{x}, \boldsymbol{\theta}_S \rangle \boldsymbol{v}_k^\top(0).$$

Let's define

$$\boldsymbol{b}(t) \triangleq \mathbf{W}_{k+1}(t) \cdots \mathbf{W}_L(t),$$

then for any constant $c_1(t) \triangleq \langle \boldsymbol{v}_k, \boldsymbol{b}(t) \rangle$ it follows :

$$\mathbf{x}^\top \boldsymbol{\beta}(t) = \mathbf{x}^\top \mathbf{W}_1(t) \cdots \mathbf{W}_k(t) \cdot \mathbf{W}_{k+1}(t) \cdots \mathbf{W}_L(t)$$

$$= \|\boldsymbol{\theta}_S\|^{k-L/L} \langle \mathbf{x}, \boldsymbol{\theta}_S \rangle \boldsymbol{v}_k^\top(0) \boldsymbol{b}(t)$$

$$= c_1(t) \|\boldsymbol{\theta}_S\|^{k-L/L} \langle \mathbf{x}, \boldsymbol{\theta}_S \rangle.$$

By setting $c(t) = c_1(t) \|\boldsymbol{\theta}_S\|^{k-L/L}$ we conclude the proof. $\quad\square$

# 3 Proofs for the shallow ReLU section

This section shows that fine-tuning from a shallow ReLU model pretrained on $\boldsymbol{\theta}_S$ has sample complexity depending on $\|\boldsymbol{\theta}_T - \boldsymbol{\theta}_S\|$, compared to training from a random initialization which depends on $\|\boldsymbol{\theta}_T\|$.

We would like to adapt the results from [10] to the case of fine-tuning in the NTK regime, where we can take better advantage of the fact that the bound in Theorem 4.1 in [10] fundamentally depends on $\|\tilde{\mathbf{y}}\|$, thus enabling us to bound the distance of each weight from $t = 0$ by using $\tilde{\mathbf{y}}$ instead of $\mathbf{y}$ for our case, where $\boldsymbol{u}(0)$ is known.

The proof scheme is as follows:

1. First we show that $\|\mathbf{H}(t) - \mathbf{H}^\infty\| = O(\frac{1}{\sqrt{m}})$, thus ensuring we are indeed in the NTK regime for $m$ bounded from bellow as in Theorem 6.1 from the main text.

2. Then, we can use an adaption of Theorem 4.1 from [10] to bound the distance of each weight $\|\mathbf{w}_r(t) - \mathbf{w}_r(0)\| \, \forall r \in [m]$.

3. Since $\mathbf{W}(0)$ is fixed, we can use the Rademacher bound in Theorem 5.1 from [10] with $\mathbf{W}(0)$ instead of $\mathbf{W}(init)$ to obtain a bound that depends on $\tilde{\mathbf{y}}^\top \mathbf{H}^\infty \tilde{\mathbf{y}}$ instead of $\mathbf{y}^\top \mathbf{H}^\infty \mathbf{y}$.

4. For $\tilde{\mathbf{y}} = \mathbf{X}(\boldsymbol{\theta}_T - \boldsymbol{\theta}_S)$, we can use Corollary 6.2 from [10] with $\boldsymbol{\beta} = (\boldsymbol{\theta}_T - \boldsymbol{\theta}_S)$ to obtain the generalization error using the Rademacher bound above.

## 3.1 Staying in the NTK regime

Start with the first item: showing that $\|\mathbf{H}(t) - \mathbf{H}^\infty\| = O(\frac{1}{\sqrt{m}})$. This is done by bounding the distance each $\mathbf{w}_r \forall r \in [m]$ travels during both the pretraining and fine-tuning optimization, which is achievable by using Theorem 4.1 from [11] "as is" for the pretraining part, and adapting it to the fine-tuning part.

**Assumptions** For brevity, we assume for the pretraining data that $|\mathbf{x}_{S_i}| \leq 1, |y_{S_i}| \leq 1$ for all $i \in [n_S]$. Also assume the following for all results:

**Assumption 3.1.** *We assume that* $\mathbf{W}(init)$*, i.e. the weights at* $t = init$*, were i.i.d. initialized* $\mathbf{w}_r \sim \mathcal{N}(\mathbf{0}, \mathbf{I})$*,* $a_r \sim \mathrm{unif}\,[\{-1, 1\}]$ *for* $r \in [m]$*.*

Also assume for $\mathbf{X}, \mathbf{X}_s$:

**Assumption 3.2.** *Define matrix* $\mathbf{H}^\infty \in \mathbb{R}^{n \times n}$ *with*

$$\mathbf{H}_{ij}^\infty = \mathbb{E}_{\mathbf{w} \sim N(\mathbf{0}, \mathbf{I})} \left[ \mathbf{x}_i^\top \mathbf{x}_j \mathbb{I} \left\{ \mathbf{w}^\top \mathbf{x}_i \geq 0, \mathbf{w}^\top \mathbf{x}_j \geq 0 \right\} \right].$$

*We assume* $\lambda_0 \triangleq \lambda_{\min}(\mathbf{H}^\infty) > 0$*, and* $\lambda_{0_S} \triangleq \lambda_{\min}(\mathbf{H}_S^\infty) > 0$ *for* $\mathbf{H}_S$ *being the NTK gram matrix of the pretraining data* $\mathbf{X}_S$*.*

The assumption that $\lambda_0 > 0$ is justified by combining Assumption 3.1 and Theorem 3.1 from [11]. The assumption that $\lambda_{0_S} > 0$, which is actually the assumption for Theorem 3.4, holds for most real-data data-sets and w.h.p for most real-life distributions, as discussed in [11].

**Assumption 3.3.** *We assume that* $m = \Omega\left(\frac{n_s^6}{\lambda_{0_S}^4 \kappa^2 \delta^3}\right)$*,* $\kappa = O\left(\frac{\epsilon\delta}{\sqrt{n_S}} + \frac{\epsilon\delta}{\sqrt{n}}\right)$ *and* $\eta_T = O\left(\frac{\lambda_0}{n^2}\right)$*,* $\eta_S = O\left(\frac{\lambda_{0_S}}{n_S^2}\right)$*.*

We now restate a few results from [11] which are applied directly for the part of pretraining:

**Theorem 3.4** (Theorem 3.1 from [11]). *If for any* $i \neq j$*,* $\mathbf{x}_i \not\parallel \mathbf{x}_j$*, then* $\lambda_0 > 0$*.*

**Theorem 3.5** (Theorem 3.3 from [11] for pretraining). *Assume Assumption 3.1, Assumption 3.2 and Assumption 3.3 hold, then with probability at least* $1 - \delta$ *over the random initialization at time* $t = init$*, we have:*

$$\frac{1}{2} \|\mathbf{y}_s - \boldsymbol{u}(init)\| = O(n_S/\delta).$$

**Lemma 3.6** (Lemma C.1 from [10]). *Assume Assumption 3.1, Assumption 3.2 and Assumption 3.3 hold, then there exists $C > 0$ such that with probability at least $1 - \delta$ over the random initialization at time $t = init$ we have*

$$\|\mathbf{w}_r(0) - \mathbf{w}_r(init)\|_2 \leq \frac{4\sqrt{n_s}\,\|\mathbf{y}_s - \boldsymbol{u}(init)\|}{\sqrt{m}\lambda_{0_S}} \quad \forall r \in [m].$$

Plugging Theorem 3.5 into Lemma 3.6 we get:

**Corollary 3.7.** *Assume Assumption 3.1, Assumption 3.2 and Assumption 3.3 hold, then there exists $C > 0$ s.t. with probability at least $1 - 2\delta$ over the random initialization at time $t = init$ we have*

$$\|\mathbf{w}_r(0) - \mathbf{w}_r(init)\|_2 \leq \frac{Cn_S}{\sqrt{m}\delta\lambda_{0_S}} \quad \forall r \in [m].$$

**Lemma 3.8** (Lemma 3.2 from [11]). *If $\mathbf{w}_1, \ldots, \mathbf{w}_m$ at $t = init$ are i.i.d. generated from $\mathcal{N}(\mathbf{0}, \mathbf{I})$, then with probability at least $1 - \delta$, the following holds. For any set of weight vectors $\mathbf{w}_1, \ldots, \mathbf{w}_m \in \mathbb{R}^d$ that satisfy for any $r \in [m]$, $\|\mathbf{w}_r(init) - \mathbf{w}_r\|_2 \leq \frac{c\delta\kappa\lambda_0}{n^2}$ for some small positive constants $c$, then the matrix $\mathbf{H} \in \mathbb{R}^{n \times n}$ defined by*

$$\mathbf{H}_{ij} = \frac{1}{m}\mathbf{x}_i^\top \mathbf{x}_j \sum_{r=1}^{m} \mathbb{I}\left\{\mathbf{w}_r^\top \mathbf{x}_i \geq 0, \mathbf{w}_r^\top \mathbf{x}_j \geq 0\right\}$$

*satisfies $\|\mathbf{H} - \mathbf{H}(init)\|_2 < \frac{\lambda_0}{4}$ and $\lambda_{\min}(\mathbf{H}) > \frac{\lambda_0}{2}$.*

We state the following lemmas that is used in the analysis:

**Lemma 3.9** (Similar to Lemma C.2 from [10]). *Assume Assumption 3.1 holds. For some $R > 0$ we define:*

$$\mathbf{A}_{r,i} \triangleq \left\{|\mathbf{x}_i^\top \mathbf{w}_r(init)| \leq R\right\}, \tag{22}$$

*then with probability at least $1 - \delta$ on the initialization of $\mathbf{W}(init)$ we get:*

$$\mathbb{E}[\mathbb{I}\{\mathbf{A}_{r,i}\}] \leq \frac{2R}{\sqrt{2\pi}\kappa},$$

*and:*

$$\sum_{i=1}^{n}\sum_{r=1}^{m} \mathbb{I}\{\mathbf{A}_{r,i}\} = O\left(\frac{mnR}{\kappa\delta}\right).$$

where the expectation is with respect to $\mathbf{W}(init)$.

*Proof.* Since $\mathbf{w}_r(init)$ has the same distribution as $\mathcal{N}(0, \kappa^2)$ we have

$$\mathbb{E}[\mathbb{I}\{\mathbf{A}_{r,i}\}] \leq \mathbb{E}[\mathbb{I}\left\{|\mathbf{x}_i^\top \mathbf{w}_r(init)| \leq R\right\}]$$

$$= \Pr_{z \sim \mathcal{N}(0, \kappa^2)}[|z| \leq R] = \int_{-R}^{R} \frac{1}{\sqrt{2\pi}\kappa} e^{-x^2/2\kappa^2} dx$$

$$\leq \frac{2R}{\sqrt{2\pi}\kappa}.$$

Then we know $\mathbb{E}\left[\sum_{i=1}^{n}\sum_{r=1}^{m} \mathbb{I}\{\mathbf{A}_{r,i}\}\right] \leq \frac{2mnR}{\sqrt{2\pi}\kappa}$. Due to Markov, with probability at least $1 - \delta$ we have:

$$\sum_{i=1}^{n}\sum_{r=1}^{m} \mathbb{I}\{\mathbf{A}_{r,i}\} = O\left(\frac{mnR}{\kappa\delta}\right).$$

$\square$

We now state our equivalent for Theorem 4.1 from [11] :

**Theorem 3.10** (Adaption of Theorem 4.1 from [11]). *Suppose Assumption 3.1 and Assumption 3.2 hold and for all $i \in [n]$, $\|\mathbf{x}_i\|_2 = 1$ and $|\mathbf{y}_i| \leq C$ for some constant $C$. if we set the number of hidden nodes*

$$m = \Omega\left(\frac{n^5 \|\tilde{\mathbf{y}}\|_2}{\lambda_0^4 \delta^2} + \frac{n_s^6}{\lambda_{0_s}^4 \kappa^2 \delta^3}\right),$$

*and we set the step sizes $\eta_T = O\left(\frac{\lambda_0}{n^2}\right)$, $\eta_S = O\left(\frac{\lambda_{0_S}}{n_S^2}\right)$ then with probability at least $1 - 2\delta$ over the random initialization we have for $t = 0, 1, 2, \ldots$*

$$\|\mathbf{y} - \boldsymbol{u}(t)\|_2^2 \leq \left(1 - \frac{\eta \lambda_0}{2}\right)^t \|\tilde{\mathbf{y}}\|_2^2 ; \tag{23}$$

$$\|\mathbf{w}_r(t) - \mathbf{w}_r(0)\| \leq \frac{4\sqrt{n} \|\tilde{\mathbf{y}}\|}{\sqrt{m} \lambda_0}, \quad \forall r \in [m].$$

*Proof of Theorem 3.10.* We follow the exact proof as in [11], with the exception of using Lemma 3.9 instead of Lemma 4.1, and Lemma 3.8 instead of Lemma 3.2.

The lower bound for $m$ is derived from the requirement on the constant $R$ that bounds the distance of $\mathbf{w}_r(t)$ from the random initialization at $t = \text{init}$. Notice that:

$$\|\mathbf{w}_r(t) - \mathbf{w}_r(\text{init})\| \leq \|\mathbf{w}_r(0) - \mathbf{w}_r(\text{init})\| + \|\mathbf{w}_r(t) - \mathbf{w}_r(0)\|, \quad \forall r \in [m],$$

where the bound for the left expression on the R.H.S is given by with probability $1 - \delta$ by Corollary 3.7.

The bound for the right expression on the R.H.S is given as a corollary of (23):

$$\|\mathbf{w}_r(t) - \mathbf{w}_r(0)\| \leq \eta \sum_{s=0}^{t-1} \left\|\frac{\partial L(\mathbf{X}, \boldsymbol{\Theta}(s))}{\partial \mathbf{w}_r(s)}\right\| \leq \eta \sum_{s=0}^{t} \frac{\sqrt{n} \|\mathbf{y} - \boldsymbol{u}(s)\|}{\sqrt{m}}$$

$$\leq \eta \sum_{s=0}^{t} \frac{\sqrt{n} \left(1 - \frac{n\lambda_0}{2}\right)^{s/2}}{\sqrt{m}} \|\mathbf{y} - \boldsymbol{u}(s)\|$$

$$\leq \eta \sum_{s=0}^{\infty} \frac{\sqrt{n} \left(1 - \frac{n\lambda_0}{2}\right)^{s/2}}{\sqrt{m}} \|\mathbf{y} - \boldsymbol{u}(s)\| = \frac{4\sqrt{n} \|\tilde{\mathbf{y}}\|}{\sqrt{m} \lambda_0}.$$

Hence we require $R = \frac{C n_S}{\sqrt{m \delta} \lambda_{0_S}} + \frac{4\sqrt{n} \|\tilde{\mathbf{y}}\|}{\sqrt{m} \lambda_0}$. From this requirement we derive the lower bound for $m$. $\square$

Using Corollary 3.7 and Theorem 3.10 we obtain a the following corollary:

**Corollary 3.11.** *Assume Assumption 3.1, Assumption 3.2 and Assumption 3.3 hold, exists $C > 0$ s.t. with probability at least $1 - 2\delta$ over the random initialization at time $t = \text{init}$ we have*

$$\|\mathbf{w}_r(t) - \mathbf{w}_r(init)\|_2 \leq \|\mathbf{w}_r(0) - \mathbf{w}_r(init)\|_2 + \|\mathbf{w}_r(t) - \mathbf{w}_r(0)\|_2$$

$$\leq \frac{C n_S}{\sqrt{m \delta} \lambda_{0_S}} + \frac{4\sqrt{n} \|\tilde{\mathbf{y}}\|}{\sqrt{m} \lambda_0} \quad \forall r \in [m].$$

Restate Lemma C.2 and Lemma C.3 from [10]:

**Lemma 3.12** (Adaption of Lemma C.2 from [10]). *Under the same setting as Theorem 3.10, with probability at least $1 - 8\delta$ over the random initialization, for all $t \geq 0$ we have:*

$$\|\mathbf{H}(0) - \mathbf{H}(init)\|_F = O\left(\frac{n^2 n_S}{\sqrt{m} \delta^{3/2} \lambda_{0_S} \kappa}\right),$$

$$\|\mathbf{H}(t) - \mathbf{H}(init)\|_F = O\left(\frac{n^2 n_S}{\sqrt{m} \delta^{3/2} \lambda_{0_S} \kappa} + \frac{n^{5/2} \|\tilde{\mathbf{y}}\|}{\sqrt{m} \lambda_0 \kappa \delta}\right),$$

$$\|\mathbf{Z}(t) - \mathbf{Z}(0)\|_F = O\left(\sqrt{\frac{n n_S}{\sqrt{m} \delta^{3/2} \kappa \lambda_{0_S}}} + \frac{n^{3/2} \|\tilde{\mathbf{y}}\|}{\sqrt{m} \lambda_0 \kappa \delta}\right),$$

*for $\mathbf{Z}(t) \triangleq \frac{1}{m} \sum_{i=1}^{n} \sum_{r=1}^{m} \mathbb{I}\left\{\mathbf{w}_r^\top(t)\mathbf{x}_i > 0\right\}.$*

*Proof.* For the first and seconds equality we use the exact proof of Lemma C.2 from [10], replacing the value of $R$ with $\frac{Cn_S}{\sqrt{m}\delta\lambda_{0_S}}$ and $\frac{Cn_S}{\sqrt{m}\delta\lambda_{0_S}} + \frac{4\sqrt{n}\|\tilde{\mathbf{y}}\|}{\sqrt{m}\lambda_0}$ respectively (by using Corollary 3.7 and Corollary 3.11 to bound the norm of the distance of each weight from initialization). The third equality also follows the same lines, with the difference being in:

$$\mathbb{E}\left[\|\mathbf{Z}(t) - \mathbf{Z}(0)\|_F^2\right] \leq \frac{1}{m}\sum_{i=1}^{n}\sum_{r=1}^{m}\mathbb{E}\left[\mathbb{I}\{A_{r,i}\} + \mathbb{I}\{\|\mathbf{w}_r(t) - \mathbf{w}_r(0)\| > \frac{4\sqrt{n}\,\|\tilde{\mathbf{y}}\|}{\sqrt{m}\lambda_0}\}\right]$$

$$\leq \frac{1}{m}\cdot mn\cdot\frac{2R}{\sqrt{2\pi}\kappa} + \frac{n}{m}\delta.$$

The last pass is justified due to the bound on $\|\mathbf{w}_r(t) - \mathbf{w}_r(0)\|$ for all $r \in [m]$ with probability $1 - \delta$ from Theorem 3.10. The wanted result is obtained, again, by plugging the R.H.S of Corollary 3.11 instead of $R$. $\qquad\square$

**Lemma 3.13** (Lemma C.3 from [10])**.** *with probability at least $1 - \delta$, we have $\|\mathbf{H}(init) - \mathbf{H}^\infty\| = O\left(\frac{n\sqrt{\log\frac{n}{\delta}}}{\sqrt{m}}\right)$.*

Using the results above, the wanted results of this section follows:

**Corollary 3.14.** *Under the same setting as Theorem 3.10, with probability at least $1 - 9\delta$ over the random initialization we have have*

$$\|\mathbf{H}(t) - \mathbf{H}^\infty\| = O\left(\frac{n^2 n_S}{\sqrt{m}\delta^{3/2}\lambda_{0_S}\kappa} + \frac{n^{5/2}\,\|\tilde{\mathbf{y}}\|}{\sqrt{m}\lambda_0\kappa\delta}\right),$$

$$\|\mathbf{H}(0) - \mathbf{H}^\infty\| = O\left(\frac{n^2 n_S}{\sqrt{m}\delta^{3/2}\lambda_{0_S}\kappa}\right).$$

*Proof.* This corollary is direct by bounding $\|\mathbf{H}(t) - \mathbf{H}^\infty\| \leq \|\mathbf{H}(init) - \mathbf{H}^\infty\| + \|\mathbf{H}(t) - \mathbf{H}(init)\|$ and using Lemma 3.13 and Lemma 3.12 to bound the R.H.S for the general $t > 0$ case and for $t = 0$. $\qquad\square$

## 3.2 Bound the distance from initialization

Write the eigen-decomposition

$$\mathbf{H}^\infty = \sum_{i=1}^{n}\lambda_i\boldsymbol{v}_i\boldsymbol{v}_i^\top,$$

where $\boldsymbol{v}_1,\ldots,\boldsymbol{v}_n \in \mathbb{R}^n$ are orthonormal eigenvectors of $\mathbf{H}^\infty$ and $\lambda_1,\ldots,\lambda_n$ are corresponding eigenvalues. also define

$$\mathbb{I}_{i,r}(t) \triangleq \mathbb{I}\left\{\mathbf{w}_r^\top(t)\mathbf{x}_i \geq 0\right\}.$$

**Theorem 3.15** (Adaption of Theorem 4.1 from [10])**.** *Assume Assumption 3.2, and suppose $m = \Omega\left(\frac{n^5\|\tilde{\mathbf{y}}\|_2^4}{\epsilon^2\kappa^2\delta^2\lambda_0^4} + \frac{n^4 n_s^2\|\tilde{\mathbf{y}}\|_2^2}{\epsilon^2\lambda_{0_s}^2\lambda_0^2\kappa^2\delta^3}\right)$. Then with probability at least $1 - \delta$ over the random initialization before pretraining ($t = init$), for all $t = 0,1,2,\ldots$ we have:*

$$\|\mathbf{y} - \boldsymbol{u}(t)\|_2 = \sqrt{\sum_{i=1}^{n}(1 - \eta\lambda_i)^{2t}\left(\boldsymbol{v}_i^\top\tilde{\mathbf{y}}\right)^2} \pm \epsilon. \tag{24}$$

We first note the important difference between this result and the original theorem is in the treatment of $\boldsymbol{u}(0)$, the predictions of the model at $t = 0$. While the original theorem shows that these predictions could be treated as negligible noise (for large enough $m$), we instead use them as part of the bound to the convergence of the training loss.

*Proof.* The core of our proof is to show that when $m$ is sufficiently large, the sequence $\{u(t)\}_{t=0}^{\infty}$ stays close to another sequence $\{\tilde{u}(t)\}_{t=0}^{\infty}$ which has a *linear* update rule:

$$\tilde{u}(0) = u(0),$$
$$\tilde{u}(t+1) = \tilde{u}(t) - \eta \mathbf{H}^{\infty}(\tilde{u}(t) - \mathbf{y}). \tag{25}$$

From (25) we have

$$\tilde{u}(t+1) - \mathbf{y} = (\mathbf{I} - \eta \mathbf{H}^{\infty})(\tilde{u}(t) - \mathbf{y}),$$

which implies

$$\tilde{u}(t) - \mathbf{y} = (\mathbf{I} - \eta \mathbf{H}^{\infty})^{t}(\tilde{u}(0) - \mathbf{y}) = -(\mathbf{I} - \eta \mathbf{H}^{\infty})^{t}\tilde{\mathbf{y}}.$$

Note that $(\mathbf{I} - \eta \mathbf{H}^{\infty})^{t}$ has eigen-decomposition

$$(\mathbf{I} - \eta \mathbf{H}^{\infty})^{t} = \sum_{i=1}^{n}(1 - \eta \lambda_i)^{t} \boldsymbol{v}_i \boldsymbol{v}_i^{\top}$$

and that $\tilde{\mathbf{y}}$ can be decomposed as

$$\tilde{\mathbf{y}} = \sum_{i=1}^{n}(\boldsymbol{v}_i^{\top}\tilde{\mathbf{y}})\boldsymbol{v}_i.$$

Then we have

$$\tilde{u}(t) - \mathbf{y} = -\sum_{i=1}^{n}(1 - \eta \lambda_i)^{t}(\boldsymbol{v}_i^{\top}\tilde{\mathbf{y}})\boldsymbol{v}_i,$$

which implies

$$\|\tilde{u}(t) - \mathbf{y}\|_2^2 = \sum_{i=1}^{n}(1 - \eta \lambda_i)^{2t}(\boldsymbol{v}_i^{\top}\tilde{\mathbf{y}})^2. \tag{26}$$

To prove that the two sequences stay close, we follow the exact proof of Theorem 4.1 in Appendix C of [10]. We start by observing the difference between the predictions at two successive steps:

$$\boldsymbol{u}_i(t+1) - \boldsymbol{u}_i(t) = \frac{1}{\sqrt{m}}\sum_{r=1}^{m} a_r \left[\sigma\left(\mathbf{w}_r(t+1)^{\top}\mathbf{x}_i\right) - \sigma\left(\mathbf{w}_r(t)^{\top}\mathbf{x}_i\right)\right]. \tag{27}$$

For each $i \in [n]$, divide the $m$ neurons into two parts: the neurons that can change their activation pattern of data-point $\mathbf{x}_i$ during optimization and those which can't. Since $|\mathbf{x}_i| \leq 1$, a neuron cannot change its activation pattern with respect to $\mathbf{x}_i$ if $|\mathbf{x}_i^{\top}\mathbf{w}_r(\text{init})| > R$ and $|\mathbf{w}_r(t) - \mathbf{w}_r(\text{init})| \leq R$ for the value of $R$ in Corollary 3.11. Define the indices of the neurons in this group (i.e. cannot change their activation pattern...) as as $\bar{S}_i$, and the indices of the complementary group as $S_i$.

From Lemma 3.9 we know that with probability $1 - \delta$, for $R = \left(\frac{n_S}{\sqrt{m\bar{\delta}\lambda_{0_S}}} + \frac{\sqrt{n}\|\tilde{\mathbf{y}}\|}{\sqrt{m}\lambda_0}\right)$

$$|\bar{S}_i| \leq O\left(\frac{mn}{\kappa\delta}\left(\frac{n_S}{\sqrt{m\bar{\delta}\lambda_{0_S}}} + \frac{\sqrt{n}\|\tilde{\mathbf{y}}\|}{\sqrt{m}\lambda_0}\right)\right). \tag{28}$$

Following the same steps as in [10] and notice that (27) can be treated as:

$$\boldsymbol{u}(t+1) - \boldsymbol{u}(t) = -\eta \mathbf{H}(t)(\boldsymbol{u}(t) - \mathbf{y}) + \boldsymbol{\epsilon}(t), \tag{29}$$

where:

$$\boldsymbol{\epsilon}_i(t) \triangleq \frac{1}{\sqrt{m}}\sum_{r\in\bar{S}_i}\left[\sigma\left(\mathbf{w}_r(t+1)^{\top}\mathbf{x}_i\right) - \sigma\left(\mathbf{w}_r(t)^{\top}\mathbf{x}_i\right)\right]$$
$$+ \frac{\eta}{m}\sum_{j=1}^{n}(u_j(t) - y_j)\mathbf{x}_j^{\top}\mathbf{x}_i\sum_{r\in\bar{S}_i}\mathbb{I}_{r,i}(t)\mathbb{I}_{r,j}(t).$$

Next use (28) to bound $\|\boldsymbol{\epsilon}(t)\|$:

$$\|\boldsymbol{\epsilon}(t)\|_2 \le \|\boldsymbol{\epsilon}(t)\|_1 \le \sum_{i=1}^{n} \frac{2\eta\sqrt{n}|\bar{S}_i|}{m} \|\boldsymbol{u}(t) - \mathbf{y}\|_2$$

$$= O\left(\frac{\sqrt{m}n^{3/2}}{\kappa\delta^{3/2}}\left(\frac{\sqrt{\delta}\|\tilde{\mathbf{y}}\|_2}{\lambda_0} + \frac{n_s}{\sqrt{n}\lambda_{0_s}}\right)\right) \frac{2\eta\sqrt{n}}{m} \|\boldsymbol{u}(t) - \mathbf{y}\|_2$$

$$= O\left(\frac{\eta n^2}{\sqrt{m}\kappa\delta^{3/2}}\left(\frac{\sqrt{\delta}\|\tilde{\mathbf{y}}\|_2}{\lambda_0} + \frac{n_s}{\sqrt{n}\lambda_{0_s}}\right)\right) \|\boldsymbol{u}(t) - \mathbf{y}\|_2.$$

Notice from Corollary 3.14 that $\mathbf{H}(t)$ stays close to $\mathbf{H}^\infty$. Then it is possible to rewrite Equation (29) as

$$\boldsymbol{u}(t+1) - \boldsymbol{u}(t) = -\eta\mathbf{H}^\infty\left(\boldsymbol{u}(k) - \mathbf{y}\right) + \boldsymbol{\zeta}(t), \tag{30}$$

where $\boldsymbol{\zeta}(t) = -\eta\left(\mathbf{H}^\infty - \mathbf{H}(t)\right)\left(\boldsymbol{u}(k) - \mathbf{y}\right) + \boldsymbol{\epsilon}(t)$. Using Corollary 3.14 it follows that

$$\|\boldsymbol{\zeta}(t)\|_2 \le \eta\|\mathbf{H}^\infty - \mathbf{H}(t)\|_2 \|\boldsymbol{u}(t) - \mathbf{y}\|_2 + \|\boldsymbol{\epsilon}(t)\|_2$$

$$= O\left(\frac{\eta n^{5/2}\|\tilde{\mathbf{y}}\|_2}{\sqrt{m}\kappa\delta\lambda_0} + \frac{\eta n^2 n_s}{\sqrt{m}\lambda_{0_s}\kappa\delta^{3/2}}\right) \|\boldsymbol{u}(t) - \mathbf{y}\|_2$$

$$+ O\left(\frac{\eta n^2}{\sqrt{m}\kappa\delta^{3/2}}\left(\frac{\sqrt{\delta}\|\tilde{\mathbf{y}}\|_2}{\lambda_0} + \frac{n_s}{\sqrt{n}\lambda_{0_s}}\right)\right) \|\boldsymbol{u}(t) - \mathbf{y}\|_2$$

$$= O\left(\frac{\eta n^{5/2}\|\tilde{\mathbf{y}}\|_2}{\sqrt{m}\kappa\delta\lambda_0} + \frac{\eta n^2 n_s}{\sqrt{m}\lambda_{0_s}\kappa\delta^{3/2}}\right) \|\boldsymbol{u}(t) - \mathbf{y}\|_2. \tag{31}$$

Apply (30) recursively and get:

$$\boldsymbol{u}(t) - \mathbf{y} = -\left(\mathbf{I} - \eta\mathbf{H}^\infty\right)^t \tilde{\mathbf{y}} + \sum_{s=0}^{t-1} \left(\mathbf{I} - \eta\mathbf{H}^\infty\right)^t \boldsymbol{\zeta}(t-1-s). \tag{32}$$

For the left term in (32) we've shown in (26) that:

$$\left\|-\left(\mathbf{I} - \eta\mathbf{H}^\infty\right)^t(\tilde{\mathbf{y}})\right\|_2 = \sqrt{\sum_{i=1}^{n}(1 - \eta\lambda_i)^{2t}(\boldsymbol{v}_i^\top\tilde{\mathbf{y}})^2}.$$

The right term in (32) can be bounded using (31):

$$\left\|\sum_{s=0}^{t-1}(\mathbf{I} - \eta\mathbf{H}^\infty)^s \boldsymbol{\zeta}(t-1-s)\right\|_2 \le \sum_{s=0}^{t-1}\|\mathbf{I} - \eta\mathbf{H}^\infty\|_2^s \|\boldsymbol{\zeta}(t-1-s)\|_2$$

$$\le \sum_{s=0}^{t-1}(1 - \eta\lambda_0)^s O\left(\frac{\eta n^{5/2}\|\tilde{\mathbf{y}}\|_2}{\sqrt{m}\kappa\delta\lambda_0} + \frac{\eta n^2 n_s}{\sqrt{m}\lambda_{0_s}\kappa\delta^{3/2}}\right) \|\boldsymbol{u}(t-1-s) - \mathbf{y}\|_2$$

$$\le \sum_{s=0}^{t-1}(1 - \eta\lambda_0)^s O\left(\frac{\eta n^{5/2}\|\tilde{\mathbf{y}}\|_2}{\sqrt{m}\kappa\delta\lambda_0} + \frac{\eta n^2 n_s}{\sqrt{m}\lambda_{0_s}\kappa\delta^{3/2}}\right)\left(1 - \frac{\eta\lambda_0}{4}\right)^{t-1-s}\|\tilde{\mathbf{y}}\|_2$$

$$\le t\left(1 - \frac{\eta\lambda_0}{4}\right)^{t-1} O\left(\frac{\eta n^{5/2}\|\tilde{\mathbf{y}}\|_2^2}{\sqrt{m}\kappa\delta\lambda_0} + \frac{\eta n^2 n_s\|\tilde{\mathbf{y}}\|_2}{\sqrt{m}\lambda_{0_s}\kappa\delta^{3/2}}\right).$$

Combining all of the above it follows:

$$\|\boldsymbol{u}(t) - \mathbf{y}\|_2 = \sqrt{\sum_{i=1}^{n}(1 - \eta\lambda_i)^{2t}(\boldsymbol{v}_i^\top\tilde{\mathbf{y}})^2} \pm O\left(t\left(1 - \frac{\eta\lambda_0}{4}\right)^{t-1}\left(\frac{\eta n^{5/2}\|\tilde{\mathbf{y}}\|_2^2}{\sqrt{m}\kappa\delta\lambda_0} + \frac{\eta n^2 n_s\|\tilde{\mathbf{y}}\|_2}{\sqrt{m}\lambda_{0_s}\kappa\delta^{3/2}}\right)\right)$$

$$= \sqrt{\sum_{i=1}^{n}(1 - \eta\lambda_i)^{2t}(\boldsymbol{v}_i^\top\tilde{\mathbf{y}})^2} \pm O\left(\frac{n^{5/2}\|\tilde{\mathbf{y}}\|_2^2}{\sqrt{m}\kappa\delta\lambda_0^2} + \frac{n^2 n_s\|\tilde{\mathbf{y}}\|_2}{\sqrt{m}\lambda_{0_s}\lambda_0\kappa\delta^{3/2}}\right).$$

where we used $\max_{t \geq 0} \{t(1 - \eta\lambda_0/4)^{t-1}\} = O(1/(\eta\lambda_0))$. From the choices of $\kappa$ and $m$, the above error term is at most $\epsilon$. This completes the proof of Theorem 3.15. $\qquad \square$

## 3.3 Deriving a population risk bound

Before proving Theorem 6.1 from the main text, we start by stating and proving some Lemmas:

**Lemma 3.16.** *Suppose $m \geq \kappa^{-2} \operatorname{poly}\left(\|\tilde{\mathbf{y}}\|_2, n, n_s, \lambda_0^{-1}, \lambda_{0_s}^{-1}, \delta^{-1}\right)$ and $\eta = O\left(\frac{\lambda_0}{n^2}\right)$. Then with probability at least $1 - \delta$ over the random initialization at $t = init$, we have for all $t \geq 0$:*

- $\|\mathbf{w}_r(t) - \mathbf{w}_r(0)\|_2 = O\left(\frac{\sqrt{n}\|\tilde{\mathbf{y}}\|_2}{\sqrt{m}\lambda_0}\right)$ $(\forall r \in [m])$, *and*

- $\|\mathbf{W}(t) - \mathbf{W}(0)\|_F \leq \sqrt{\tilde{\mathbf{y}}^\top (\mathbf{H}^\infty)^{-1} \tilde{\mathbf{y}}} + \dfrac{\operatorname{poly}\left(\|\tilde{\mathbf{y}}\|_2, n, n_s, \frac{1}{\lambda_0}, \frac{1}{\lambda_{0_s}}, \frac{1}{\delta}\right)}{m^{1/4}\kappa^{1/2}}$.

*Proof.* The bound on the movement of each $\mathbf{w}_r$ is proven in Theorem 3.10. The second bound is achieved by coupling the trajectory of $\{\mathbf{W}(t)\}_{k=0}^\infty$ with another simpler trajectory $\left\{\widetilde{\mathbf{W}}(t)\right\}_{k=0}^\infty$ defined as:

$$
\begin{aligned}
\widetilde{\mathbf{W}}(0) &= \mathbf{W}(0), \\
\operatorname{vec}\left(\widetilde{\mathbf{W}}(t+1)\right) &= \operatorname{vec}\left(\widetilde{\mathbf{W}}(t)\right) \\
&\quad - \eta\mathbf{Z}(0)\left(\mathbf{Z}(0)^\top \operatorname{vec}\left(\widetilde{\mathbf{W}}(t)\right) - \mathbf{y}\right).
\end{aligned}
\tag{33}
$$

First we give a proof of $\left\|\widetilde{\mathbf{W}}(\infty) - \widetilde{\mathbf{W}}(0)\right\|_F = \sqrt{\tilde{\mathbf{y}}^\top \mathbf{H}(0)^{-1}\tilde{\mathbf{y}}}$ as an illustration for the proof of Lemma 3.16. Define $\boldsymbol{v}(t) = \mathbf{Z}(0)^\top \operatorname{vec}\left(\widetilde{\mathbf{W}}(t)\right) \in \mathbb{R}^n$. Then from (33) we have $\boldsymbol{v}(0) = \mathbf{Z}(0)^\top \operatorname{vec}(\mathbf{W}(0))$ and $\boldsymbol{v}(k+1) = \boldsymbol{v}(t) - \eta\mathbf{H}(0)(\boldsymbol{v}(t) - \mathbf{y})$, yielding $\boldsymbol{v}(t) - \mathbf{y} = -(\mathbf{I} - \eta\mathbf{H}(0))^t\tilde{\mathbf{y}}$. Plugging this back to (33) we get $\operatorname{vec}\left(\widetilde{\mathbf{W}}(t+1)\right) - \operatorname{vec}\left(\widetilde{\mathbf{W}}(t)\right) = \eta\mathbf{Z}(0)(\mathbf{I} - \eta\mathbf{H}(0))^t\tilde{\mathbf{y}}$. Then taking a sum over $k = 0, 1, \ldots$ we have

$$
\begin{aligned}
\operatorname{vec}\left(\widetilde{\mathbf{W}}(\infty)\right) - \operatorname{vec}\left(\widetilde{\mathbf{W}}(0)\right) &= \sum_{k=0}^\infty \eta\mathbf{Z}(0)(\mathbf{I} - \eta\mathbf{H}(0))^k\tilde{\mathbf{y}} \\
&= \mathbf{Z}(0)\mathbf{H}(0)^{-1}\tilde{\mathbf{y}}.
\end{aligned}
$$

The desired result thus follows:

$$
\begin{aligned}
\left\|\widetilde{\mathbf{W}}(\infty) - \widetilde{\mathbf{W}}(0)\right\|_F^2 &= \tilde{\mathbf{y}}^\top \mathbf{H}(0)^{-1}\mathbf{Z}(0)^\top \mathbf{Z}(0)\mathbf{H}(0)^{-1}\tilde{\mathbf{y}} \\
&= \tilde{\mathbf{y}}^\top \mathbf{H}(0)^{-1}\tilde{\mathbf{y}}.
\end{aligned}
$$

Now we bound the difference between the trajectories. Recall the update rule for $\mathbf{W}$:

$$
\operatorname{vec}(\mathbf{W}(t+1)) = \operatorname{vec}(\mathbf{W}(t)) - \eta\mathbf{Z}(t)(\boldsymbol{u}(t) - \mathbf{y}).
\tag{34}
$$

Follow the same steps from Lemma 5.3 from [10], using the results from Theorem 3.15 when needed to obtain the proof for this lemma. According to the proof of Theorem 3.15 we can write

$$
\boldsymbol{u}(t) - \mathbf{y} = -(\mathbf{I} - \eta\mathbf{H}^\infty)^t\tilde{\mathbf{y}} + \boldsymbol{e}(t),
\tag{35}
$$

where

$$
\|\boldsymbol{e}(t)\| = O\left(t\left(1 - \frac{\eta\lambda_0}{4}\right)^{t-1} \cdot \left(\frac{\eta n^{5/2}\|\tilde{\mathbf{y}}\|_2^2}{\sqrt{m}\kappa\delta\lambda_0} + \frac{\eta n^2 n_s\|\tilde{\mathbf{y}}\|_2}{\sqrt{m}\lambda_{0_s}\kappa\delta^{3/2}}\right)\right).
\tag{36}
$$

Plugging (35) into (34) and taking a sum over $t = 0, 1, \ldots, T - 1$, we get:

$$
\begin{aligned}
&\mathrm{vec}\,(\mathbf{W}(T)) - \mathrm{vec}\,(\mathbf{W}(0)) \\
&= \sum_{t=0}^{T-1} (\mathrm{vec}\,(\mathbf{W}(t+1)) - \mathrm{vec}\,(\mathbf{W}(t))) \\
&= -\sum_{t=0}^{T-1} \eta \mathbf{Z}(t)(\boldsymbol{u}(t) - \mathbf{y}) \\
&= \sum_{t=0}^{T-1} \eta \mathbf{Z}(t) \left( (\mathbf{I} - \eta \mathbf{H}^\infty)^t \tilde{\mathbf{y}} - \boldsymbol{e}(t) \right) \\
&= \sum_{t=0}^{T-1} \eta \mathbf{Z}(t)(\mathbf{I} - \eta \mathbf{H}^\infty)^t \tilde{\mathbf{y}} - \sum_{t=0}^{T-1} \eta \mathbf{Z}(t)\boldsymbol{e}(t) \\
&= \sum_{t=0}^{T-1} \eta \mathbf{Z}(0)(\mathbf{I} - \eta \mathbf{H}^\infty)^t \tilde{\mathbf{y}} + \sum_{t=0}^{T-1} \eta (\mathbf{Z}(t) - \mathbf{Z}(0))(\mathbf{I} - \eta \mathbf{H}^\infty)^t \tilde{\mathbf{y}} - \sum_{t=0}^{T-1} \eta \mathbf{Z}(t)\boldsymbol{e}(t). \qquad (37)
\end{aligned}
$$

The second and the third terms in (37) are considered perturbations, and we can upper bound their norms easily. For the second term, from Lemma 3.8 we get:

$$
\begin{aligned}
&\left\| \sum_{t=0}^{T-1} \eta (\mathbf{Z}(t) - \mathbf{Z}(0))(\mathbf{I} - \eta \mathbf{H}^\infty)^t \mathbf{y} \right\|_2 \\
&\leq \sum_{t=0}^{T-1} \eta \cdot O\left( \sqrt{\frac{n^{3/2} \|\tilde{\mathbf{y}}\|_2}{\sqrt{m}\kappa\delta\lambda_0} + \frac{nn_s}{\sqrt{m}\kappa\lambda_{0_s}\delta^{3/2}}} \right) \|\mathbf{I} - \eta \mathbf{H}^\infty\|_2^t \|\tilde{\mathbf{y}}\|_2 \\
&\leq O\left( \eta \sqrt{\frac{n^{3/2} \|\tilde{\mathbf{y}}\|_2}{\sqrt{m}\kappa\delta\lambda_0} + \frac{nn_s}{\sqrt{m}\kappa\lambda_{0_s}\delta^{3/2}}} \right) \sum_{t=0}^{T-1} (1 - \eta\lambda_0)^t \|\tilde{\mathbf{y}}\|_2 \\
&= O\left( \sqrt{\frac{n^{3/2} \|\tilde{\mathbf{y}}\|_2^3}{\sqrt{m}\kappa\delta\lambda_0^3} + \frac{nn_s \|\tilde{\mathbf{y}}\|_2^2}{\sqrt{m}\kappa\lambda_{0_s}\lambda_0^2\delta^{3/2}}} \right). \qquad (38)
\end{aligned}
$$

For the third term we get:

$$
\begin{aligned}
&\left\| \sum_{t=0}^{T-1} \eta \mathbf{Z}(t)\boldsymbol{e}(t) \right\|_2 \\
&\leq \sum_{t=0}^{T-1} \eta\sqrt{n} \cdot O\left( t\left(1 - \frac{\eta\lambda_0}{4}\right)^{t-1} \cdot \left( \frac{\eta n^{5/2} \|\tilde{\mathbf{y}}\|_2^2}{\sqrt{m}\kappa\delta\lambda_0} + \frac{\eta n^2 n_s \|\tilde{\mathbf{y}}\|_2}{\sqrt{m}\lambda_{0_s}\kappa\delta^{3/2}} \right) \right) \\
&= O\left( \left( \frac{\eta^2 n^3 \|\tilde{\mathbf{y}}\|_2^2}{\sqrt{m}\kappa\delta\lambda_0} + \frac{\eta^2 n^{5/2} n_s \|\tilde{\mathbf{y}}\|_2}{\sqrt{m}\lambda_{0_s}\kappa\delta^{3/2}} \right) \sum_{t=0}^{T-1} t\left(1 - \frac{\eta\lambda_0}{4}\right)^{t-1} \right) \\
&= O\left( \left( \frac{\eta^2 n^3 \|\tilde{\mathbf{y}}\|_2^2}{\sqrt{m}\kappa\delta\lambda_0} + \frac{\eta^2 n^{5/2} n_s \|\tilde{\mathbf{y}}\|_2}{\sqrt{m}\lambda_{0_s}\kappa\delta^{3/2}} \right) \cdot \frac{1}{\eta\lambda_0} \right) \\
&= O\left( \frac{\eta n^3 \|\tilde{\mathbf{y}}\|_2^2}{\sqrt{m}\kappa\delta\lambda_0^2} + \frac{\eta n^{5/2} n_s \|\tilde{\mathbf{y}}\|_2}{\sqrt{m}\lambda_{0_s}\lambda_0\kappa\delta^{3/2}} \right). \qquad (39)
\end{aligned}
$$

Define $\mathbf{K} = \eta \sum_{t=0}^{T-1} (\mathbf{I} - \eta \mathbf{H}^\infty)^t$. using $\|\mathbf{H}(0) - \mathbf{H}^\infty\|_F = O\left( \frac{n^2 n_s}{\sqrt{m}\lambda_{0_s}\kappa\delta^{3/2}} \right)$ (Corollary 3.14) we have

$$\left\| \sum_{t=0}^{T-1} \eta \mathbf{Z}(0)(\mathbf{I} - \eta \mathbf{H}^{\infty})^{t} \tilde{\mathbf{y}} \right\|_{2}^{2} \tag{40}$$

$$= \|\mathbf{Z}(0)\mathbf{K}\tilde{\mathbf{y}}\|_{2}^{2} \tag{41}$$

$$= \tilde{\mathbf{y}}^{\top} \mathbf{K} \mathbf{Z}(0)^{\top} \mathbf{Z}(0) \mathbf{K} \tilde{\mathbf{y}} \tag{42}$$

$$= \tilde{\mathbf{y}}^{\top} \mathbf{K} \mathbf{H}(0) \mathbf{K} \tilde{\mathbf{y}} \tag{43}$$

$$\leq \tilde{\mathbf{y}}^{\top} \mathbf{K} \mathbf{H}^{\infty} \mathbf{K} \tilde{\mathbf{y}} + \|\mathbf{H}(0) - \mathbf{H}^{\infty}\|_{2} \|\mathbf{K}\|_{2}^{2} \|\tilde{\mathbf{y}}\|_{2}^{2} \tag{44}$$

$$\leq \tilde{\mathbf{y}}^{\top} \mathbf{K} \mathbf{H}^{\infty} \mathbf{K} \tilde{\mathbf{y}} + O\left( \frac{n^{2} n_{s}}{\sqrt{m} \lambda_{0_{s}} \kappa \delta^{3/2}} \right) \cdot \left( \eta \sum_{t=0}^{T-1} (\mathbf{I} - \eta \lambda_{0})^{t} \right)^{2} \|\tilde{\mathbf{y}}\|_{2}^{2} \tag{45}$$

$$= \tilde{\mathbf{y}}^{\top} \mathbf{K} \mathbf{H}^{\infty} \mathbf{K} \tilde{\mathbf{y}} + O\left( \frac{n^{2} n_{s} \|\tilde{\mathbf{y}}\|_{2}^{2}}{\sqrt{m} \lambda_{0_{s}} \lambda_{0}^{2} \kappa \delta^{3/2}} \right). \tag{46}$$

Let the eigen-decomposition of $\mathbf{H}^{\infty}$ be $\mathbf{H}^{\infty} = \sum_{i=1}^{n} \lambda_{i} \boldsymbol{v}_{i} \boldsymbol{v}_{i}^{\top}$. Since $\mathbf{K}$ is a polynomial of $\mathbf{H}^{\infty}$, it has the same set of eigenvectors as $\mathbf{H}^{\infty}$, and we have

$$\mathbf{K} = \sum_{i=1}^{n} \eta \sum_{t=0}^{T-1} (1 - \eta \lambda_{i})^{t} \boldsymbol{v}_{i} \boldsymbol{v}_{i}^{\top} = \sum_{i=1}^{n} \frac{1 - (1 - \eta \lambda_{i})^{T}}{\lambda_{i}} \boldsymbol{v}_{i} \boldsymbol{v}_{i}^{\top}.$$

It follows that

$$\mathbf{K} \mathbf{H}^{\infty} \mathbf{K} = \sum_{i=1}^{n} \left( \frac{1 - (1 - \eta \lambda_{i})^{T}}{\lambda_{i}} \right)^{2} \lambda_{i} \boldsymbol{v}_{i} \boldsymbol{v}_{i}^{\top} \preceq \sum_{i=1}^{n} \frac{1}{\lambda_{i}} \boldsymbol{v}_{i} \boldsymbol{v}_{i}^{\top} = (\mathbf{H}^{\infty})^{-1}.$$

Plugging this into (40), we get

$$\left\| \sum_{t=0}^{T-1} \eta \mathbf{Z}(0)(\mathbf{I} - \eta \mathbf{H}^{\infty})^{t} \tilde{\mathbf{y}}_{2} \right\| \leq \sqrt{ \tilde{\mathbf{y}}^{\top} (\mathbf{H}^{\infty})^{-1} \tilde{\mathbf{y}} + O\left( \frac{n^{2} n_{s} \|\tilde{\mathbf{y}}\|_{2}^{2}}{\sqrt{m} \lambda_{0_{s}} \lambda_{0}^{2} \kappa \delta^{3/2}} \right)} \tag{47}$$

$$\leq \sqrt{ \tilde{\mathbf{y}}^{\top} (\mathbf{H}^{\infty})^{-1} \tilde{\mathbf{y}} } + O\left( \sqrt{ \frac{n^{2} n_{s} \|\tilde{\mathbf{y}}\|_{2}^{2}}{\sqrt{m} \lambda_{0_{s}} \lambda_{0}^{2} \kappa \delta^{3/2}} } \right). \tag{48}$$

Finally, plugging the three bounds (38), (39) and (47) into (37), we have

$$\|\mathbf{W}(T) - \mathbf{W}(0)\|_{F}$$
$$= \|\text{vec}(\mathbf{W}(T)) - \text{vec}(\mathbf{W}(0))\|_{2}$$
$$\leq \sqrt{ \tilde{\mathbf{y}}^{\top} (\mathbf{H}^{\infty})^{-1} \tilde{\mathbf{y}} } + O\left( \sqrt{ \frac{n^{2} n_{s} \|\tilde{\mathbf{y}}\|_{2}^{2}}{\sqrt{m} \lambda_{0_{s}} \lambda_{0}^{2} \kappa \delta^{3/2}} } \right) + O\left( \sqrt{ \frac{n^{3/2} \|\tilde{\mathbf{y}}\|_{2}^{3}}{\sqrt{m} \kappa \delta \lambda_{0}^{3}} + \frac{n n_{s} \|\tilde{\mathbf{y}}\|_{2}^{2}}{\sqrt{m} \kappa \lambda_{0_{s}} \lambda_{0}^{2} \delta^{3/2}} } \right)$$
$$+ O\left( \frac{\eta n^{3} \|\tilde{\mathbf{y}}\|_{2}^{2}}{\sqrt{m} \kappa \delta \lambda_{0}^{2}} + \frac{\eta n^{5/2} n_{s} \|\tilde{\mathbf{y}}\|_{2}}{\sqrt{m} \lambda_{0_{s}} \lambda_{0} \kappa \delta^{3/2}} \right)$$
$$= \sqrt{ \tilde{\mathbf{y}}^{\top} (\mathbf{H}^{\infty})^{-1} \tilde{\mathbf{y}} } + \frac{\text{poly}\left( \|\tilde{\mathbf{y}}\|_{2}, n, n_{s}, \frac{1}{\lambda_{0}}, \frac{1}{\lambda_{0_{s}}}, \frac{1}{\delta} \right)}{m^{1/4} \kappa^{1/2}}.$$

This finishes the proof of Lemma 3.16. $\qquad\square$

**Lemma 3.17.** *Given $R > 0$, with probability at least $1 - \delta$ over the random initialization $(\mathbf{W}(init), \boldsymbol{a})$, simultaneously for every $B > 0$, the following function class*

$$\mathcal{F}_{R,B}^{\mathbf{W}(0), \boldsymbol{a}} = \{ f_{\mathbf{W}} : \|\boldsymbol{w}_{r} - \boldsymbol{w}_{r}(0)\|_{2} \leq R \, (\forall r \in [m]),$$
$$\|\mathbf{W} - \mathbf{W}(0)\|_{F} \leq B \}$$

*has empirical Rademacher complexity bounded as:*

$$\mathcal{R}_S\left(\mathcal{F}_{R,B}^{\mathbf{W}(0),\boldsymbol{a}}\right) = \frac{1}{n}\mathbb{E}_{\boldsymbol{\varepsilon}\in\{\pm1\}^n}\left[\sup_{f\in\mathcal{F}_{R,B}^{\mathbf{W}(0),\boldsymbol{a}}}\sum_{i=1}^n \varepsilon_i f(\mathbf{x}_i)\right]$$

$$\leq \frac{B}{\sqrt{n}} + \frac{2R(R+\frac{Cn_s}{\sqrt{m\delta}\lambda_{0_S}})\sqrt{m}}{\kappa} + R\sqrt{2\log\frac{2}{\delta}}.$$

*Proof.* We need to upper bound

$$\mathcal{R}_S\left(\mathcal{F}_{R,B}^{\mathbf{W}(0),\boldsymbol{a}}\right) = \frac{1}{n}\mathbb{E}_{\boldsymbol{\varepsilon}\sim\{\pm1\}^n}\left[\sup_{f\in\mathcal{F}_{R,B}^{\mathbf{W}(0),\boldsymbol{a}}}\sum_{i=1}^n \varepsilon_i f(\mathbf{x}_i)\right]$$

$$= \frac{1}{n}\mathbb{E}_{\boldsymbol{\varepsilon}\sim\{\pm1\}^n}\left[\sup_{\substack{\mathbf{W}:\|\mathbf{W}-\mathbf{W}(0)\|_{2,\infty}\leq R \\ \|\mathbf{W}-\mathbf{W}(0)\|_F\leq B}}\sum_{i=1}^n \varepsilon_i \sum_{r=1}^m \frac{1}{\sqrt{m}}a_r\sigma(\boldsymbol{w}_r^\top\boldsymbol{x}_i)\right],$$

where $\|\mathbf{W}-\mathbf{W}(0)\|_{2,\infty} = \max_{r\in[m]}\|\mathbf{w}_r - \mathbf{w}_r(0)\|_2$.

Similar to the proof of Lemma 3.9, we define events:

$$\tilde{A}_{r,i} \triangleq \left\{\left|\mathbf{w}_r(0)^\top\mathbf{x}_i\right| \leq R\right\}, \quad i\in[n], r\in[m].$$

Since we only look at $\mathbf{W}$ such that $\|\mathbf{w}_r - \mathbf{w}_r(0)\|_2 \leq R$ for all $r\in[m]$, if $\mathbb{I}\{\tilde{A}_{r,i}\} = 0$ we must have $\mathbb{I}\{\mathbf{w}_r^\top\mathbf{x}_i > 0\} = \mathbb{I}\{\mathbf{w}_r(0)\mathbf{x}_i \geq 0\} = \mathbb{I}_{r,i}(0)$. Thus we have:

$$\mathbb{I}\left\{\neg\tilde{A}_{r,i}\right\}\sigma\left(\mathbf{w}_r^\top\mathbf{x}_i\right) = \mathbb{I}\left\{\neg\tilde{A}_{r,i}\right\}\mathbb{I}_{r,i}(0)\mathbf{w}_r^\top\mathbf{x}_i,$$

It follows that:

$$\sum_{i=1}^n \varepsilon_i \sum_{r=1}^m a_r\sigma\left(\mathbf{w}_r^\top\mathbf{x}_i\right) - \sum_{i=1}^n \varepsilon_i \sum_{r=1}^m a_r\mathbb{I}_{r,i}(0)\mathbf{w}_r^\top\mathbf{x}_i$$

$$= \sum_{r=1}^m\sum_{i=1}^n \left(\mathbb{I}\left\{\tilde{A}_{r,i}\right\} + \mathbb{I}\left\{\neg\tilde{A}_{r,i}\right\}\right)\varepsilon_i a_r\left(\sigma\left(\mathbf{w}_r^\top\mathbf{x}_i\right) - \mathbb{I}_{r,i}(0)\mathbf{w}_r^\top\mathbf{x}_i\right)$$

$$= \sum_{r=1}^m\sum_{i=1}^n \mathbb{I}\left\{\tilde{A}_{r,i}\right\}\varepsilon_i a_r\left(\sigma\left(\mathbf{w}_r^\top\mathbf{x}_i\right) - \mathbb{I}_{r,i}(0)\mathbf{w}_r^\top\mathbf{x}_i\right)$$

$$= \sum_{r=1}^m\sum_{i=1}^n \mathbb{I}\left\{\tilde{A}_{r,i}\right\}\varepsilon_i a_r\left(\sigma\left(\mathbf{w}_r^\top\mathbf{x}_i\right) - \mathbb{I}_{r,i}(0)\mathbf{w}_r(0)^\top\mathbf{x}_i - \mathbb{I}_{r,i}(0)(\mathbf{w}_r - \mathbf{w}_r(0))^\top\mathbf{x}_i\right)$$

$$= \sum_{r=1}^m\sum_{i=1}^n \mathbb{I}\left\{\tilde{A}_{r,i}\right\}\varepsilon_i a_r\left(\sigma\left(\mathbf{w}_r^\top\mathbf{x}_i\right) - \sigma\left(\mathbf{w}_r(0)^\top\mathbf{x}_i\right) - \mathbb{I}_{r,i}(0)(\mathbf{w}_r - \mathbf{w}_r(0))^\top\mathbf{x}_i\right)$$

$$\leq \sum_{r=1}^m\sum_{i=1}^n \mathbb{I}\left\{\tilde{A}_{r,i}\right\}\cdot 2R.$$

Thus we can bound the Rademacher complexity as:

$$\mathcal{R}_S\left(\mathcal{F}_{R,B}^{\mathbf{W}(0),\boldsymbol{a}}\right) = \frac{1}{n}\mathbb{E}_{\boldsymbol{\varepsilon}\sim\{\pm1\}^n}\left[\sup_{\substack{\mathbf{W}:\|\mathbf{W}-\mathbf{W}(0)\|_{2,\infty}\leq R \\ \|\mathbf{W}-\mathbf{W}(0)\|_F\leq B}}\sum_{i=1}^n\varepsilon_i\sum_{r=1}^m\frac{a_r}{\sqrt{m}}\sigma\left(\mathbf{w}_r^\top\mathbf{x}\right)\right]$$

$$\leq \frac{1}{n}\mathbb{E}_{\boldsymbol{\varepsilon}\sim\{\pm1\}^n}\left[\sup_{\substack{\mathbf{W}:\|\mathbf{W}-\mathbf{W}(0)\|_{2,\infty}\leq R \\ \|\mathbf{W}-\mathbf{W}(0)\|_F\leq B}}\sum_{i=1}^n\varepsilon_i\sum_{r=1}^m\frac{a_r}{\sqrt{m}}\mathbb{I}_{r,i}(0)\mathbf{w}_r^\top\mathbf{x}_i\right] + \frac{2R}{n\sqrt{m}}\sum_{r=1}^m\sum_{i=1}^n\mathbb{I}\left\{\tilde{A}_{r,i}\right\}$$

$$\leq \frac{1}{n}\mathbb{E}_{\boldsymbol{\varepsilon}\sim\{\pm1\}^n}\left[\sup_{\mathbf{W}:\|\mathbf{W}-\mathbf{W}(0)\|_F\leq B}\sum_{i=1}^n\varepsilon_i\sum_{r=1}^m\frac{a_r}{\sqrt{m}}\mathbb{I}_{r,i}(0)\mathbf{w}_r^\top\mathbf{x}_i\right] + \frac{2R}{n\sqrt{m}}\sum_{r=1}^m\sum_{i=1}^n\mathbb{I}\left\{\tilde{A}_{r,i}\right\}$$

$$= \frac{1}{n}\mathbb{E}_{\boldsymbol{\varepsilon}\sim\{\pm1\}^n}\left[\sup_{\mathbf{W}:\|\mathbf{W}-\mathbf{W}(0)\|_F\leq B}\text{vec}\left(\mathbf{W}\right)^\top\mathbf{Z}(0)\boldsymbol{\varepsilon}\right] + \frac{2R}{n\sqrt{m}}\sum_{r=1}^m\sum_{i=1}^n\mathbb{I}\left\{\tilde{A}_{r,i}\right\}$$

$$= \frac{1}{n}\mathbb{E}_{\boldsymbol{\varepsilon}\sim\{\pm1\}^n}\left[\sup_{\mathbf{W}:\|\mathbf{W}-\mathbf{W}(0)\|_F\leq B}\text{vec}\left(\mathbf{W}-\mathbf{W}(0)\right)^\top\mathbf{Z}(0)\boldsymbol{\varepsilon}\right] + \frac{2R}{n\sqrt{m}}\sum_{r=1}^m\sum_{i=1}^n\mathbb{I}\left\{\tilde{A}_{r,i}\right\}$$

$$\leq \frac{1}{n}\mathbb{E}_{\boldsymbol{\varepsilon}\sim\{\pm1\}^n}\left[B\cdot\|\mathbf{Z}(0)\boldsymbol{\varepsilon}\|_2\right] + \frac{2R}{n\sqrt{m}}\sum_{r=1}^m\sum_{i=1}^n\mathbb{I}\left\{\tilde{A}_{r,i}\right\}$$

$$\leq \frac{B}{n}\sqrt{\mathbb{E}_{\boldsymbol{\varepsilon}\sim\{\pm1\}^n}\left[\|\mathbf{Z}(0)\boldsymbol{\varepsilon}\|_2^2\right]} + \frac{2R}{n\sqrt{m}}\sum_{r=1}^m\sum_{i=1}^n\mathbb{I}\left\{\tilde{A}_{r,i}\right\}$$

$$= \frac{B}{n}\|\mathbf{Z}(0)\|_F + \frac{2R}{n\sqrt{m}}\sum_{r=1}^m\sum_{i=1}^n\mathbb{I}\left\{\tilde{A}_{r,i}\right\}.$$

Next we bound $\|\mathbf{Z}(0)\|_F$ and $\sum_{r=1}^m\sum_{i=1}^n\mathbb{I}\left\{\tilde{A}_{r,i}\right\}$.

For $\|\mathbf{Z}(0)\|_F$, notice that

$$\|\mathbf{Z}(0)\|_F^2 = \frac{1}{m}\sum_{r=1}^m\left(\sum_{i=1}^n\mathbb{I}_{r,i}(0)\right) \leq n.$$

Now observe the following lemma:

**Lemma 3.18.** *With probability $1-\delta$, if $\left|\mathbf{w}_r(init)^\top\mathbf{x}_i\right| > R + \frac{Cn_s}{\sqrt{m\delta}\lambda_{0_S}}$ then $\mathbb{I}\{\tilde{A}_{r,i}\} = 0$.*

*Proof.* From Corollary 3.7 exists $C > 0$ s.t. with probability $1-\delta$, for all $r \in [m]$ : $\|\mathbf{w}_r(0) - \mathbf{w}_r(init)\| \leq \frac{Cn_s}{\sqrt{m\delta}\lambda_{0_S}}$. From the triangle inequality:

$$\left|\mathbf{w}_r(0)^\top\mathbf{x}_i\right| \geq \left\|\mathbf{w}_r(0)^\top\mathbf{x}_i\right\|$$
$$= \left\|\mathbf{w}_r(init)^\top\mathbf{x}_i - (\mathbf{w}_r(init) - \mathbf{w}_r(0))^\top\mathbf{x}_i\right\|$$
$$\geq \left\|\mathbf{w}_r(init)^\top\mathbf{x}_i\right\| - \left\|(\mathbf{w}_r(init) - \mathbf{w}_r(0))^\top\mathbf{x}_i\right\|.$$

Since $\|\mathbf{x}\| = 1$, and with the same probability above:

$$\left\|(\mathbf{w}_r(init) - \mathbf{w}_r(0))^\top\mathbf{x}_i\right\| \leq \frac{Cn_s}{\sqrt{m\delta}\lambda_{0_S}},$$

thus

$$\left|\mathbf{w}_r(0)^\top \mathbf{x}_i\right| \geq \left\|\mathbf{w}_r(\text{init})^\top \mathbf{x}_i\right\| - \left\|(\mathbf{w}_r(\text{init}) - \mathbf{w}_r(0))^\top \mathbf{x}_i\right\|$$

$$\geq \left\|\mathbf{w}_r(\text{init})^\top \mathbf{x}_i\right\| - \frac{Cn_s}{\sqrt{m}\bar{\delta}\lambda_{0_S}}$$

$$> R + \frac{Cn_s}{\sqrt{m}\bar{\delta}\lambda_{0_S}} - \frac{Cn_s}{\sqrt{m}\bar{\delta}\lambda_{0_S}} = R.$$

$\square$

For $\sum_{r=1}^m \sum_{i=1}^n \mathbb{I}\left\{\tilde{A}_{r,i}\right\}$, from Lemma 3.18 we notice that

$$\sum_{r=1}^m \sum_{i=1}^n \mathbb{I}\left\{\tilde{A}_{r,i}\right\} \leq \sum_{r=1}^m \sum_{i=1}^n \mathbb{I}\left\{A_{r,i}\right\},$$

for $A_{r,i}$ being defined as in Lemma 3.9. Since all $m$ neurons are independent at $t = \text{init}$ and from Lemma 3.9 and Corollary 3.7 we know $\mathbb{E}\left[\sum_{i=1}^n \mathbb{I}\left\{A_{r,i}\right\}\right] \leq \frac{\sqrt{2}n(R + \frac{Cn_s}{\sqrt{m}\bar{\delta}\lambda_{0_S}})}{\sqrt{\pi}\kappa}$. Then by Hoeffding's inequality, with probability at least $1 - \delta/2$ we have

$$\sum_{r=1}^m \sum_{i=1}^n \mathbb{I}\left\{\tilde{A}_{r,i}\right\} \leq \sum_{r=1}^m \sum_{i=1}^n \mathbb{I}\left\{A_{r,i}\right\} \leq mn \left(\frac{\sqrt{2}(R + \frac{Cn_s}{\sqrt{m}\bar{\delta}\lambda_{0_S}})}{\sqrt{\pi}\kappa} + \sqrt{\frac{\log\frac{2}{\delta}}{2m}}\right).$$

Therefore, with probability at least $1 - \delta$, the Rademacher complexity is bounded as:

$$\mathcal{R}_S\left(\mathcal{F}_{R,B}^{\mathbf{W}(0),\boldsymbol{a}}\right) \leq \frac{B}{n}\left(\sqrt{n}\right) + \frac{2R}{n\sqrt{m}}mn\left(\frac{\sqrt{2}(R + \frac{Cn_s}{\sqrt{m}\bar{\delta}\lambda_{0_S}})}{\sqrt{\pi}\kappa} + \sqrt{\frac{\log\frac{2}{\delta}}{2m}}\right)$$

$$= \frac{B}{\sqrt{n}} + \frac{2\sqrt{2}R(R + \frac{Cn_s}{\sqrt{m}\bar{\delta}\lambda_{0_S}})\sqrt{m}}{\sqrt{\pi}\kappa} + R\sqrt{2\log\frac{2}{\delta}},$$

completing the proof of Lemma 3.17. (Note that the high probability events used in the proof do not depend on the value of $B$, so the above bound holds simultaneously for every $B$.) $\square$

### 3.4 Proof of Theorem 6.1 (Main Text)

*Proof of Theorem 6.1 (Main Text).* First of all, from Assumption 3.1 we have $\lambda_{\min}(\mathbf{H}^\infty) \geq \lambda_0$. The rest of the proof is conditioned on this happening. We follow exactly the same steps as in [10] with minor changes.

From Theorem 3.10, Lemma 3.16 and Lemma 3.17, we know that for any sample $S$, with probability at least $1 - \delta/3$ over the random initialization, the followings hold simultaneously:

(i) Optimization succeeds (Theorem 3.10):

$$\frac{1}{2}\left\|\tilde{\mathbf{y}} - \boldsymbol{u}(t)\right\| \leq \left(1 - \frac{\eta\lambda_0}{2}\right)^t \cdot \left\|\tilde{\mathbf{y}}\right\|_2 \leq \frac{1}{2}.$$

This implies an upper bound on the training error $L(\mathbf{X}; \boldsymbol{\Theta}(t)) = \frac{1}{n}\sum_{i=1}^n \ell(f_{\mathbf{W}(t)}(\boldsymbol{x}_i), y_i) = \frac{1}{n}\sum_{i=1}^n \ell(u_i(t), y_i)$:

$$L(\mathbf{X}; \boldsymbol{\Theta}(t)) = \frac{1}{n}\sum_{i=1}^n [\ell(u_i(t), y_i) - \ell(y_i, y_i)] \leq \frac{1}{n}\sum_{i=1}^n |u_i(t) - y_i|$$

$$\leq \frac{1}{\sqrt{n}}\left\|\boldsymbol{u}(t) - \mathbf{y}\right\|_2 = \sqrt{\frac{2\frac{1}{2}\left\|\tilde{\mathbf{y}} - \boldsymbol{u}(t)\right\|}{n}} \leq \frac{1}{\sqrt{n}}.$$

(ii) $\|\mathbf{w}_r(t) - \mathbf{w}_r(0)\|_2 \leq R$ ($\forall r \in [m]$) and $\|\mathbf{W}(t) - \mathbf{W}(0)\|_F \leq B$, where $R = O\left(\frac{\sqrt{n}\|\tilde{\mathbf{y}}\|_2}{\sqrt{m}\lambda_0}\right)$

and $B = \sqrt{\tilde{\mathbf{y}}^\top (\mathbf{H}^\infty)^{-1} \tilde{\mathbf{y}}} + \frac{\text{poly}\left(\|\tilde{\mathbf{y}}\|_2, n, n_s, \frac{1}{\lambda_0}, \frac{1}{\lambda_{0_s}}, \frac{1}{\delta}\right)}{m^{1/4}\kappa^{1/2}}$. Note that $B \leq O\left(\sqrt{\frac{n}{\lambda_0}}\right)$.

(iii) Let $B_i = i$ ($i = 1, 2, \ldots$). Simultaneously for all $i$, the function class $\mathcal{F}_{R,B_i}^{\mathbf{W}(0),\boldsymbol{a}}$ has Rademacher complexity bounded as

$$\mathcal{R}_S\left(\mathcal{F}_{R,B_i}^{\mathbf{W}(0),\boldsymbol{a}}\right) \leq \frac{B_i}{\sqrt{n}} + \frac{2R(R + \frac{Cn_s}{\sqrt{m}\delta\lambda_{0_S}})\sqrt{m}}{\kappa} + R\sqrt{2\log\frac{10}{\delta}}.$$

Let $i^*$ be the smallest integer such that $B \leq B_{i^*}$. Then we have $i^* \leq O\left(\sqrt{\frac{n}{\lambda_0}}\right)$ and $B_{i^*} \leq B + 1$.

From above we know $f_{\mathbf{W}(t)} \in \mathcal{F}_{R,B_{i^*}}^{\mathbf{W}(0),\boldsymbol{a}}$, and

$$\mathcal{R}_S\left(\mathcal{F}_{R,B_{i^*}}^{\mathbf{W}(0),\boldsymbol{a}}\right) \leq \frac{B+1}{\sqrt{n}} + \frac{2R(R + \frac{Cn_s}{\sqrt{m}\delta\lambda_{0_S}})\sqrt{m}}{\kappa} + R\sqrt{2\log\frac{10}{\delta}}$$

$$= \frac{\sqrt{\tilde{\mathbf{y}}^\top (\mathbf{H}^\infty)^{-1} \tilde{\mathbf{y}}}}{\sqrt{n}} + \frac{1}{\sqrt{n}} + \frac{\text{poly}\left(\|\tilde{\mathbf{y}}\|_2, n, n_s, \frac{1}{\lambda_0}, \frac{1}{\lambda_{0_s}}, \frac{1}{\delta}\right)}{m^{1/4}\kappa^{1/2}} + \frac{2R(R + \frac{Cn_s}{\sqrt{m}\delta\lambda_{0_S}})\sqrt{m}}{\kappa} + R\sqrt{2\log\frac{10}{\delta}}$$

$$\leq \sqrt{\frac{\tilde{\mathbf{y}}^\top (\mathbf{H}^\infty)^{-1} \tilde{\mathbf{y}}}{n}} + \frac{1}{\sqrt{n}} + \frac{\text{poly}\left(\|\tilde{\mathbf{y}}\|_2, n, n_s, \frac{1}{\lambda_0}, \frac{1}{\lambda_{0_s}}, \frac{1}{\delta}\right)}{m^{1/4}\kappa^{1/2}} \leq \sqrt{\frac{\tilde{\mathbf{y}}^\top (\mathbf{H}^\infty)^{-1} \tilde{\mathbf{y}}}{n}} + \frac{2}{\sqrt{n}}.$$

Next, from the theory of Rademacher complexity and a union bound over a finite set of different $i$'s, for any random initialization $(\mathbf{W}(\text{init}), \boldsymbol{a})$, with probability at least $1 - \delta/3$ over the sample $S$, we have

$$\sup_{f \in \mathcal{F}_{R,B_i}^{\mathbf{W}(0),\boldsymbol{a}}} \{R(f) - L(f)\} \leq 2\mathcal{R}_S\left(\mathcal{F}_{R,B_i}^{\mathbf{W}(0),\boldsymbol{a}}\right) + O\left(\sqrt{\frac{\log\frac{n}{\lambda_0\delta}}{n}}\right), \qquad \forall i \in \left\{1, 2, \ldots, O\left(\sqrt{\frac{n}{\lambda_0}}\right)\right\}.$$

Finally, taking a union bound, we know that with probability at least $1 - \frac{2}{3}\delta$ over the sample $S$ and the random initialization $(\mathbf{W}(\text{init}), \boldsymbol{a})$, the followings are all satisfied (for some $i^*$):

$$L(\mathbf{X}, \boldsymbol{\Theta}(t)) \leq \frac{1}{\sqrt{n}},$$

$$f\left(\cdot, \boldsymbol{\Theta}(t)\right) \in \mathcal{F}_{R,B_{i^*}}^{\mathbf{W}(0),\boldsymbol{a}},$$

$$\mathcal{R}_S\left(\mathcal{F}_{R,B_{i^*}}^{\mathbf{W}(0),\boldsymbol{a}}\right) \leq \sqrt{\frac{\tilde{\mathbf{y}}^\top (\mathbf{H}^\infty)^{-1} \tilde{\mathbf{y}}}{n}} + \frac{2}{\sqrt{n}},$$

$$\sup_{f \in \mathcal{F}_{R,B_{i^*}}^{\mathbf{W}(0),\boldsymbol{a}}} \{R(f) - L(f)\} \leq 2\mathcal{R}_S\left(\mathcal{F}_{R,B_{i^*}}^{\mathbf{W}(0),\boldsymbol{a}}\right) + O\left(\sqrt{\frac{\log\frac{n}{\lambda_0\delta}}{n}}\right).$$

These together can imply:

$$R(\boldsymbol{\Theta}(t)) \leq \frac{1}{\sqrt{n}} + 2\mathcal{R}_S\left(\mathcal{F}_{R,B_{i^*}}^{\mathbf{W}(0),\boldsymbol{a}}\right) + O\left(\sqrt{\frac{\log\frac{n}{\lambda_0\delta}}{n}}\right)$$

$$\leq \frac{1}{\sqrt{n}} + 2\left(\sqrt{\frac{\tilde{\mathbf{y}}^\top (\mathbf{H}^\infty)^{-1} \tilde{\mathbf{y}}}{n}} + \frac{2}{\sqrt{n}}\right) + O\left(\sqrt{\frac{\log\frac{n}{\lambda_0\delta}}{n}}\right)$$

$$= 2\sqrt{\frac{\tilde{\mathbf{y}}^\top (\mathbf{H}^\infty)^{-1} \tilde{\mathbf{y}}}{n}} + O\left(\sqrt{\frac{\log\frac{n}{\lambda_0\delta}}{n}}\right).$$

This completes the proof. $\qquad\square$

## 3.5 Linear teachers: Proof of corollary 6.3

We now consider the case where

$$g_S(\mathbf{x}) = \mathbf{x}^\top \boldsymbol{\theta}_S, \quad g_T(\mathbf{x}) = \mathbf{x}^\top \boldsymbol{\theta}_T,$$

which is the case in Corollary 6.3.

We will start with stating the random initialization population risk bound for this case, which we will compare our result to:

**Corollary 3.19** (Population risk bound for random initialization from [10]). *Assume that the random initialized model with weights $\boldsymbol{\Theta}(t)$ was trained according to Theorem 5.1 from [10] and that $\mathbf{y} = \mathbf{X}\boldsymbol{\theta}_T$, then with probability $1 - \delta$*

$$R(\boldsymbol{\Theta}(t)) \leq \frac{3\sqrt{2}\,\|\boldsymbol{\theta}_T\|_2}{\sqrt{n}} + O\left(\sqrt{\frac{\log \frac{n}{\lambda_0 \delta}}{n}}\right). \tag{49}$$

This corollary is a direct result of plugging $\mathbf{y} = \mathbf{X}\boldsymbol{\theta}_T$ into Corollary 6.2 from [10], and plugging the result into Theorem 5.1 from [10].

As discussed in Section 6.1, we will assume that $f(\mathbf{X}; \boldsymbol{\Theta}(0)) = \mathbf{X}\boldsymbol{\theta}_S$. Since our model is non-linear, this assumption is not trivial, and requires some clarification. For infinite width, Lemma 1 from [12] tells us that $n_S = 2d$ can suffice to achieve this, if the samples are chosen according to some conditions. For the case of finite width $m$, like is assumed in Theorem 6.1, no such equivalent exist. However, we can use Corollary 3.19 for the pretraining, and achieve an $\epsilon$ bound on the pretraining population risk, for sufficiently large $n_S = \Omega\left(\frac{\|\boldsymbol{\theta}_S\|^2}{\epsilon^2}\right)$. Then, approximate relaxations can be derived when we assume the two functions are $\epsilon$ close (i.e. $f(\mathbf{x}, \boldsymbol{\Theta}(0)) = \mathbf{x}^\top \boldsymbol{\theta}_S + \epsilon$).

We now restate our two corollaries from the main text:

**Corollary 6.2** (Main Text). *Suppose that $g_S(\mathbf{X}) \triangleq \mathbf{X}^\top \boldsymbol{\theta}_S$, $g_T(\mathbf{X}) \triangleq \mathbf{X}^\top \boldsymbol{\theta}_T$, and assume Assumption 3.2 holds. Then, $\sqrt{\tilde{\mathbf{y}}^\top (\mathbf{H}^\infty)^{-1} \tilde{\mathbf{y}}} \leq 3\,\|\boldsymbol{\theta}_T - \boldsymbol{\theta}_S\|_2$.*

This is a direct corollary of Theorem 6.1 from [10] on $\tilde{\mathbf{y}}$ defined above.

**Corollary 6.3** (Main Text). *Under the conditions of Theorem 6.1 and Corollary 6.2, it holds that*

$$R(\boldsymbol{\Theta}(t)) \leq \frac{6\,\|\boldsymbol{\theta}_T - \boldsymbol{\theta}_S\|_2}{\sqrt{n}} + O\left(\sqrt{\frac{\log \frac{n}{\lambda_0 \delta}}{n}}\right).$$

Comparing this to Corollary 3.19 gives us the exact condition for when it is better to use fine-tuning instead of random initialization, which is

$$\|\boldsymbol{\theta}_T - \boldsymbol{\theta}_S\| < \frac{\|\boldsymbol{\theta}_T\|}{\sqrt{2}}.$$

We will now provide a proof for this results:

*Proof of Corollary 6.3.* In order to achieve this bound, we use the assumption on $f(\mathbf{X}; \boldsymbol{\Theta}(0))$, which gives us:

$$\tilde{\mathbf{y}} = \mathbf{X}\boldsymbol{\theta}_T - \mathbf{X}\boldsymbol{\theta}_S = \mathbf{X}(\boldsymbol{\theta}_T - \boldsymbol{\theta}_S).$$

Hence, we can treat $\tilde{\mathbf{y}}$ as if it was created by a linear label generation function $\boldsymbol{\theta}_T - \boldsymbol{\theta}_S$. Hence, by using Theorem 6.1 from [10] we can bound

$$\sqrt{\tilde{\mathbf{y}}(\mathbf{H}^\infty)^{-1}\tilde{\mathbf{y}}} \leq 3\,\|\boldsymbol{\theta}_T - \boldsymbol{\theta}_S\|.$$

Plugging this into Theorem 6.1 concludes the proof. $\qquad\square$

**Code** In the code used for the experiments we used Pytorch [13], Numpy [14], SciPy [15], and Matplotlib [16].