# OpenReview forum: "A Theoretical Analysis of Fine-tuning with Linear Teachers"
_NeurIPS.cc/2021/Conference — NeurIPS 2021 Poster_

### Official Review · Reviewer_J96B · 2021-07-15

**Rating:** 7
**Confidence:** 3

**Summary:**

Taking linear regression as a base model, this paper provides some interesting insights about model fine-tuning by theoretical analyses. The main findings are demonstrating more factors such as covariance structure of target data, and network depth, have impact to the process of fine-tuning.

**Limitations And Societal Impact:**

The authors have stated their main limitation, that is, working on linear regression model.

**Main Review:**

Originality: Providing theoretical support to fine-tuning is sort of new in machine learning, although there are some pioneer works which are also mentioned in the paper. As I am not knowledgeable enough to check the correctness of theory details, I cannot justify its technical novelty.

Quality: This paper focuses on theory analysis to fine-tuning and the techniques are complete. Some conclusions are interesting and I only have some minor concerns.
First, I feel confused when I try to read the theorems. For example, it seems theta_T, theta have the different meanings in Section 3 and 4 from line105 and Eq. 1.
Second, it is strange to put Experiments as Section 4.1 with two the main conclusion Theorem 4.2 before it. Is it clearer to make them as two sequential subsections?
Third, it seems that Section 5.2 tries to say freezing the former k layers and updating the remaining ones fails. But any workaround?

Clarity: The submission might be clear but it is not easy to follow the details for me. But overall I feel it is good.

Significance: I felt the results are important and also interesting. This work tries to answer some challenging questions for fine-tuning, which may inspire the future research.

**Time Spent Reviewing:**

5h

---

> ### Author Response · Authors · 2021-08-10
> **Clarifications and Freezing layers workarounds**
>
> Thank you for your review, for your time and for your valuable feedback on the quality and clarity of the paper. We will aim to make the work clearer.
>
> >Quality.a: First, I feel confused when I try to read the theorems. For example, it seems theta_T, theta have the different meanings in Section 3 and 4 from line105 and Eq. 1.
>
> In both sections 3 and 4, $\theta_T$ represents the ground truth of the target task. This is not the case for $\theta$: whereas in section 3, $\theta$ is used to describe some vector or matrix in a general model, in section 4 it is used to describe the linear predictor achieved after the fine-tuning process. Assume that the weights of the linear predictor at time $t$ to be $\mathbf{w}(t)$, then $\mathbf{\theta} = \lim_{t \rightarrow \infty} \mathbf{w}(t)$, as is also described at line 134.
> This overload of $\theta$ is definitely confusing, and we appreciate you spotting it. Future versions will address this in order to make them clearer.
>
> >Quality.b: Second, it is strange to put Experiments as Section 4.1 with two the main conclusion Theorem 4.2 before it. Is it clearer to make them as two sequential subsections?
>
> We believe it is clearer, and we will address it in future versions.
>
> >Quality.c: Third, it seems that Section 5.2 tries to say freezing the former k layers and updating the remaining ones fails. But any workaround?
>
> Thanks for bringing up this important point. This result is achieved under the assumption of 0-balancedness prior to pretraining, which happens e.g. when initializing the weights with an infinitesimally small variance (further techniques on initializing balanced networks can be seen in [29]). This property leads to the degeneracy of the output of the frozen k-layers. A possible workaround would be to initialize the weights prior to pretraining with a larger scale of initialization (e.g. xavier), raising the rank of each layer and preventing degeneracy. Pretraining with multiple source tasks (as suggested in e.g. Du, Simon Shaolei, et al. "Few-Shot Learning via Learning the Representation, Provably." International Conference on Learning Representations. 2020) may also help prevent this degeneracy. A future version will address this subtlety of Theorem 5.4.

---

### Official Review · Reviewer_e71p · 2021-07-19

**Rating:** 4
**Confidence:** 4

**Summary:**

This paper presents a theoretical analysis of fine-tuning for linear regression problems. The main results in the paper are as follows.

1. Theorem 4.2 develops an upper bound on the population risk on the target task in terms of the number of samples of n, the training time t, the data matrix of the target task and the difference between the true weights of the target task (the teacher) and the pertained weights on the source. There is technical novelty in how this bound is derived (matrix perturbation methods). The conclusion of this theorem is that if the teacher weights and pertained weights are close, as measured by the top few eigenvectors of the target data, then one gets a better population risk by training for fewer iterations.
2. This analysis is repeated for deep linear networks in Theorem 5.1 with similar conclusions.
3. Theorems 5.2-5.3 show an interesting phenomenon where if the student is a deep linear network, then the weights of the student need only be aligned with the teacher (as opposed to being close in ell-2 norm).
4. Theorem 6.1 is an analogue of the above result in the NTK regime. The technical novelty in the proof here is that one has to analyze the NTK dynamics in two stages.

Some experimental results are provided on synthetic data and MNIST.

**Main Review:**

This is a well-written paper on a practically relevant problem, namely that of understanding when fine-tuning is effective for transfer learning. The technical development of the paper is however quite underwhelming because it heavily relies on linear regression setup.

1. I have concerns about the intellectual novelty of this paper. The paper should address (a) how these results can be extended to more practically relevant setups beyond linear models, (b) why they do not follow from existing results on linear/two-layer networks [27-29] or the NTK regime of [39], and (c) how these are results practically relevant. In particular, it is not sufficient to measure the distance between the source and target tasks using the difference/alignment of their weights because it is unlikely to extend to nonlinear teachers. A more useful way to think about this problem seems to be assume a similarity measure on the tasks (and line 348 in the discussion section is not far from this) and study this setting. Distances between even simple tasks can be quite non-trivial, e.g., https://arxiv.org/abs/2002.04747.

2. Theorem 5.4 which claims that freezing the bottom few layers of the student does not work for target tasks. This certainly does not hold for nonlinear networks where freezing weights is a commonly used strategy to reduce the sample complexity of learning the new task.

3. The development in section 6 should come with the following caveat that the authors are encouraged to make prominent. The NTK regime holds when the pertained model does not move very far away from its initialization. Similarly fine-tuning in the NTK regime holds when the fine-tuned model does not travel very far away from the pertained one.

**Time Spent Reviewing:**

2

---

> ### Author Response · Authors · 2021-08-10
> **Addressing concerns regarding using linear models and the intellectual novelty of our work**
>
> Thank you for your review and for taking the time to evaluate our work and provide valuable feedback on it. Taking into account your concerns and caveats will improve the quality of our work.
>
> >1.(a). how these results can be extended to more practically relevant setups beyond linear models
>
> Thank you for this question. Deep non-linear models, especially those with more than two or three layers, are difficult to analyze theoretically, which can only be done under very stringent constraints or under strong assumptions. This analysis is especially challenging when attempting to understand the inductive bias of gradient-based methods used to train these models. Such analysis can benefit from understanding deep linear networks, as there are already many challenges at this level. In addition, we believe that insights from the analysis of the gradient-based training procedure of deep linear networks can be applied to the training procedure of deep non-linear networks, as has been demonstrated in numerous papers. For example, [29] theoretically analyze deep linear networks, show that balanced initialization leads to faster convergence in deep linear models, and also empirically demonstrate the same for a fully-connected ReLU network in figure 1.(d). In section 5, we consider deep linear models that are balanced, a characteristic observed in deep non-linear homogeneous networks as well (e.g. Du et al. "Algorithmic regularization in learning deep homogeneous models: layers are automatically balanced." NeurIPS. 2018). Thus, our results might also be relevant to the non-linear regime, since these consist of linear domains (e.g. ​​Phuong and Lampert. "The inductive bias of ReLU networks on orthogonally separable data." ICLR. 2020).
>
> >1.(b). why they do not follow from existing results on linear/two-layer networks [27-29] or the NTK regime of [39]
>
> The purpose of this work is to analyze the generalization of fine-tuning in multiple settings and provide appropriate risk bounds. In order to achieve this, we analyze models whose weights prior to training on the target data were the outcome of a pretraining phase and provide upper risk bounds for this setting. None of the works mentioned deal with fine-tuning, and most don’t provide generalization bounds: [29] only analyze convergence of deep linear networks, not generalization; [27] examine deep linear models that were initialized with near-zero initialization; [28, 39] analyze shallow-ReLU models that were initialized randomly, i.i.d Gaussian.
> Our work also shows the effect of depth on fine-tuning and the risk of fine-tuned models, which to our knowledge has never been done before. This effect is shown by providing the explicit expression of the end-to-end equivalent of a deep linear model. Moreover, in Theorems 5.2 and 5.3 we provide novel theoretical and empirical analyses that illustrate two scenarios where the inductive bias of deep linear networks is prominent and beneficial.
>
> > 1.(c). how are these results practically relevant. In particular, it is not sufficient to measure the distance between the source and target tasks using the difference/alignment of their weights because it is unlikely to extend to nonlinear teachers. A more useful way to think about this problem seems to be assume a similarity measure on the tasks (and line 348 in the discussion section is not far from this) and study this setting. Distances between even simple tasks can be quite non-trivial, e.g., https://arxiv.org/abs/2002.04747.
>
> Thank you for the interesting reference and the important question. The question of the right source-target metric is key to choosing a good source task, as you wrote. Here we show the factors that affect this choice for linear networks. We do believe these factors will come into play for the non-linear case (and in particular NTK analyzes which strongly rely on understanding the linear case). Things may be more subtle. e.g., in the NTK regime (as indeed mentioned in the discussion section), where one may need to consider the covariance matrix in the NTK representation rather than in the original feature space.
>
> >2. Theorem 5.4 which claims that freezing the bottom few layers of the student does not work for target tasks. This certainly does not hold for nonlinear networks where freezing weights is a commonly used strategy to reduce the sample complexity of learning the new task.
>
> Freezing the bottom few layers of the student, sometimes referred to as "representation learning", is indeed a commonly used strategy. In our work we expose a weakness of this strategy even for a very simple model. Interestingly, our results agree with previous theoretical results on representation learning (Du et al. "Few-Shot Learning via Learning the Representation, Provably." ICLR. 2020), where it is shown that lower sample complexity is achievable when the target task resides in the subspace spanned by the source (pretraining) tasks. This result makes it interesting to understand in which cases this degerency happens, and when does representation learning work, e.g. by analyzing fine-tuning with multiple source tasks in a similar manner to our work.
>
> >3. The development in section 6 should come with the following caveat that the authors are encouraged to make prominent. The NTK regime holds when the pertained model does not move very far away from its initialization. Similarly fine-tuning in the NTK regime holds when the fine-tuned model does not travel very far away from the pertained one.
>
> We will better explain lines 313-318 in future versions of this work in order to make this caveat more prominent.

---

### Official Review · Reviewer_PQB2 · 2021-07-30

**Rating:** 7
**Confidence:** 3

**Summary:**

Fine-tuning is a common technique in training large networks on relatively small datasets. It achieves excellent generalization results. However, it is not well understood theoretically. This paper attempts to analyze the sample complexity of fine-tuning for regression with linear teachers in several settings. Intutitvely, the success of fine-tuning depends on the similarity between the source tasks and the target tasks. This paper shows that the relevant measure has to do with the relation between the source task, the target task and the covariance structure of the target data. The analysis offers the insights into the inductive bias of gradient-descent optimization and the implied relation between the source task, the target task and the target covariance that is needed for this process to succeed.

This work focuses on the case where both source and target regression tasks are linear functions of the input. It first considers the case where the model architecture is one layer linear networks, and derive sample complexity results for fine-tuning. It shows that under certain assumptions there can be substantial sample complexity reduction when certain measure of source task and target task is low.

Further, it analyzes deep linear networks. There it derives a novel result providing the learned model as a function of initialization, and uses it to derive corresponding sample complexity bounds. The results provide several surprising insights. The first is that the covariance structure of the target data has a significant effect on the success of fine-tuning. In particular the relation between the vector of the source-target difference and the eigenvectors of the target covariance will affect how well fine-tuning will work. Second, it finds that the depth of the network can dramatically affect the results of the fine-tuning process, since deeper networks will serve to cancel the effect of scale differences between source and target tasks. Further, it corroborates the results by empirical evaluations.

A limitation of the work is the simplicity of the models analysed. Their setting deals only with linear teachers, and assumes the label noise to be zero. Furthermore, it only shows upper bounds on the population loss, and not matching lower bounds. For deep linear networks it assumes a certain initialization which is less standard than normalized initializers like Xavier. For non-linear models, it analyses the simple model of a Shallow-ReLU network, and only in the NTK regime.

**Limitations And Societal Impact:**

Yes.

**Main Review:**

The main strength of the paper is in the theoretical results, the analysis of the fine-tuning for regression with linear teachers in several settings. It gives an insight into the right measure of similarity between the source tasks and the target tasks, for fine-tuning to succeed. Particularly, it shows that the relevant measure has to do with the relation between the source task, the target task, and teh covariance structure of the target data. This is an interesting find. The theoretical results are non-trivial and did require some effort.

The main weakness is in the fact that the theoretical results are for the restrictive setting of a linear teacher. Further, the theoretical results does not lead to any insight into improving fine-tuning methodology for any setting. It only gives some insights into the method of fine-tuning.

**Time Spent Reviewing:**

6

---

> ### Author Response · Authors · 2021-08-10
> **Justification for the setting of linear teachers and practical implications of our work**
>
> We appreciate your time and effort spent on writing your review and your valuable comments and notes.
>
> >1. The main weakness is in the fact that the theoretical results are for the restrictive setting of a linear teacher.
>
> We thank you for this comment. The inductive bias of optimizing deep models with gradient based methods is still not well understood in many cases. The setting of linear teachers and linearly separable data is very often used in the literature to prove properties of this inductive bias and has provided numerous insights on non-linear models [1, 2,  3 , 4]. We obtained surprising results for this setting, e.g. in section 5. Furthermore, this setting is considered a common choice when analyzing linear and deep linear models, as it allows the assumption of realizability. We believe this is a good test-bed for analysis and sets the ground for analyzing more complex nonlinear teachers.
>
> >2. Further, the theoretical results does not lead to any insight into improving fine-tuning methodology for any setting. It only gives some insights into the method of fine-tuning.
>
> We agree that a key goal of theoretical work is both to explain why current methods work, and also how they can be improved. We do believe our work suggests interesting practical implications. For example, a key challenge in fine-tuning is how to choose the right source task (e.g., see the wealth of recent work on variants of BERT self-supervision tasks). Our results suggest that fine-tuning is beneficial when $\theta_T$ and $\theta_S$ are close in the span of the bottom eigenvectors of the target distribution. Despite not having access to the target ground truth, one might apply this observation when choosing the source (pretraining) task; for example, for image classification, a source task which focuses on rarer, “finer” attributes of the images, which are believed to be shared with target task (e.g. by domain-specific experts), might be beneficial. It is only suggested by our theory, and should be verified empirically, perhaps as a direction for future research.
>
> A second example implication of our work is that depth can have an impact on the quality of transfer and thus different depths should be explored when fine-tuning. Moreover, if one notices that adding more layers to a deep linear network improves accuracy on the validation set, one could use the direct expression for the infinite number of layers given by Eq.6.
>
> [1] Kaifeng Lyu and Jian Li. Gradient descent maximizes the margin of homogeneous neural networks. In International Conference on Learning Representations, 2019.
>
> [2] Sarussi, Roei, Alon Brutzkus, and Amir Globerson. "Towards Understanding Learning in Neural Networks with Linear Teachers." arXiv preprint arXiv:2101.02533 (2021).
>
> [3] ​​Phuong, Mary, and Christoph H. Lampert. "The inductive bias of ReLU networks on orthogonally separable data." International Conference on Learning Representations. 2020.
>
> [4] Wang, Gang, Georgios B. Giannakis, and Jie Chen. "Learning ReLU networks on linearly separable data: Algorithm, optimality, and generalization." IEEE Transactions on Signal Processing 67.9 (2019): 2357-2370.

---

### Official Review · Reviewer_ynu3 · 2021-08-02

**Rating:** 6
**Confidence:** 4

**Summary:**

The authors perform an analysis of fine-tuning a parameter learned from a source task for a target task in the few-shot setting (i.e. $n \ll d$). They provide:


1. A fine-grained analysis of the linear setting, where both source and target tasks involve linear functions of the input $x$, with risk bounds that incorporate the data covariance of the target task.
2. A similar analysis for the overparameterized linear setting, providing risk bounds that are not a function of $\lVert \theta_S - \theta_T\rVert$ in the infinite depth limit. Furthermore, in the same limit, they show that fixing the initial layers provably forces the predictor to remain in the span of $\theta_S$.
3. An analysis of fine-tuning for shallow ReLU networks in the NTK regime.

The theoretical results are corroborated by simulations.


**Limitations And Societal Impact:**

The authors have adequately addressed the limitations of their work. I particularly appreciate the thoroughness with which the authors addressed the shortcomings of the settings they considered in their discussion in Section 7.

**Main Review:**

**Originality:**
The authors provide a novel analysis of fine-tuning in the deep linear setting, as well as in the NTK setting. The NTK analysis in particular incorporates a new argument allowing for the analysis of gradient descent in this regime from a non-random initialization obtained from pre-training.

**Quality:**
The submission seems technically sound.  I thought that the experimental evaluation was very thorough and illustrated the predictions of the theory well. It was particularly interesting to see the tighter analysis incorporating the covariance in the linear setting, as well as the accompanying characterization of when fine-tuning helps.

Questions:
1. How are the results in Section 5 impacted by weakening the $0$-balancedness assumption on the weights to a $\delta$-approximate balancedness? According to [1], initializing with small variance only guarantees approximate balance, not $0$-balance. How does $\delta$ enter into the risk bounds, and does approximate balance affect the negative result on freezing earlier layers in Theorem 5.4?
2. Can Theorem 5.2 be experimentally verified more directly? For example, a simple experiment would be to fix $\langle \hat\theta_S, \hat\theta_T\rangle$ and verify that scaling does not affect the risk (and vice versa).

**Clarity:**
I thought that the results were clearly presented for the most part; however, I did find the following issues:
1. The deep linear setting was not particularly well-described in the main text. What are the dimensions on the matrices $W_1, \dots, W_L$? What are the assumptions made on these dimensions?
2. Line 168-171: I did not quite follow the discussion of the tradeoff for the choice of $m$. Is this assuming that $g(\lambda, t, n)^{3/2} < 1$, and so it is advantageous to push more of the norm of $\theta_T - \theta_S$ into the first term since it decays more quickly?

Minor Typos:
1. Line 211: Missing parenthesis around “Theorem 4.1”
2. $\beta$ is not defined in Equation 5, which I assume is supposed to be $\beta(\infty)$.
3. Line 222: “$\lim_t{t\to\infty}\beta(t)$” should be “$\lim_{t \to \infty}\beta(t)$”.
4. Line 347: Fix theorem reference

**Significance:**
The analysis in the paper suggests future directions in obtaining tighter bounds in the few-shot setting, as well as better characterizations of when pretraining helps, by explicitly incorporating the data covariance in the analysis.

I thought the NTK-regime analysis of fine-tuning was particularly interesting, especially since prior work (to my knowledge) relies heavily on the random initialization throughout the analysis, and thus it is unclear how one would analyze meta-learning/few-shot learning (where one starts from a fixed initialization) within the NTK regime.

[1] Sanjeev Arora, Nadav Cohen, Noah Golowich, and Wei Hu. A convergence analysis of gradient descent for deep linear neural networks. In International Conference on Learning Representations, 2018.


**Time Spent Reviewing:**

5

---

> ### Author Response · Authors · 2021-08-10
> **$\delta$-approximate balancedness, additional experiments and clarifications**
>
> We are grateful for the time and effort you put into this review, and for the helpful comments and suggestions.
>
> > Quality.1. How are the results in Section 5 impacted by weakening the 0-balancedness assumption on the weights to a $\delta$-approximate balancedness? According to [1], initializing with small variance only guarantees approximate balance, not 0-balance. How does $\delta$ enter into the risk bounds, and does approximate balance affect the negative result on freezing earlier layers in Theorem 5.4?
>
> This is a very good question. The experiments in section 5 were actually conducted with small initialization scale, that is not guaranteed to result in 0-balancedness, but rather in
> $\delta$-approximate balancedness where $\delta$ is small. The empirical results we obtained (Figure 2) are very much in agreement with our theoretical predictions. As for a theoretical analysis, we presented an analysis for $\delta=0$ for simplicity. However, the analysis can be performed for non-zero $\delta$, and this will introduce “error” terms depending on $\delta$ into the results (e.g., see Section H.2 of [2]).
>
> As for the effect of $\delta$-approximate balancedness on Theorem 5.4: the experiments in Figure 2 were done with non zero $\delta$ and the phenomenon of learning failure is observed there. Intuitively, this is because the effective rank of the weight matrices is close to one, and thus learning the second layer is an ill-conditioned problem, which leads to slower convergence and can prevent the model from fine-tuning on the target data with a constant gradient step. We will add a discussion of this point.
>
> >Quality.2. Can Theorem 5.2 be experimentally verified more directly? For example, a simple experiment would be to fix $\langle \hat{\theta}_S, \hat{\theta}_T \rangle$ and verify that scaling does not affect the risk (and vice versa).
>
> Theorem 5.2 predicts that for fixed directions of the source and target tasks, the test loss of a deep linear model does not depend on the norm of the source task, but only on the norm of the target weight. Indeed, the experiment you suggest is a better test-bed for this result. Following your suggestion, we carried out the experiment (described next) and will include it in future versions. In this experiment, we first fixed $\hat{\theta}_S, \hat{\theta}_T$ such that $||\hat{\theta}_S - \hat{\theta}_T|| \approx 0.1$. We then tested what happens when a rescaling is only to the source weights, or only to the target weights (before rescaling both weights have a norm of 1). We shall report here only the test loss results for a seven-layer deep linear model for the sake of brevity:
>
> Scale | 2 | 3 | 4 | 5 |
> --------|--- |---|---|----|
> Source is rescaled | 0.008 ± 5e-4 | 0.019 ± 0.001 | 0.033 ± 0.003 | 0.044 ± 0.005
> Target is rescaled| 0.031 ± 0.004 | 0.103 ± 0.012 | 0.221 ± 0.024 | 0.425 ± 0.047
>
> We find that, as Theorem 5.2 predicts, when the source weights are rescaled (and the norm of the target weights is 1) we see only a small difference in the test loss between the different scales. However, when the target weights are rescaled (and the norm of the source weight is 1), the test loss changes significantly between a small scale and a large scale.
>
> >Clarity.1. The deep linear setting was not particularly well-described in the main text. What are the dimensions on the matrices $W_1,\ldots ,W_L$ ? What are the assumptions made on these dimensions?
>
> Indeed, it would be clearer to define these dimensions properly, as follows. Let the dimensions of $W_i$ be $d_{i-1} \times d_{i}$. We assume that $d_0 = d$ (where $d$ is the dimension of the data), $d_{L} = 1$ and $d_i \ge d$ for $i \in [1, L-1]$ (this is not strictly necessary. We just use it to avoid rank reduction effects in the network itself, so that the implicit bias of GD is the only source of rank reduction). This will be clarified in future versions of the paper.
>
> >Clarity.2. ​​Line 168-171: I did not quite follow the discussion of the tradeoff for the choice of $m$. Is this assuming that $g(\lambda ,t,n)^{3/2}<1$  and so it is advantageous to push more of the norm of $θ_T−θ_S$ into the first term since it decays more quickly?
>
> Indeed, this paragraph was written under the assumption that $n$ is large enough such that $g(\lambda,t,n)^{3/2}<1$, hence causing the first term to decay more quickly. This will be clarified in future versions of the paper.
>
> [2] Yun, Chulhee, Shankar Krishnan, and Hossein Mobahi. "A unifying view on implicit bias in training linear neural networks." International Conference on Learning Representations. 2021.

---

### Decision · Program_Chairs · 2021-09-27

**Decision:**

Accept (Poster)

**Comment:**

This paper presents new results for fine-tuning in the linear regime (linear parameterization and linearized NN). While there is still a gap between the linear regime and practice, the reviewers and the AC believe this paper gives an important initial step towards building a comprehensive theory for fine-tuning.